# RESIDUAL DIFFUSION IMPLICIT MODELS

## ABSTRACT

Diffusion models achieve state-of-the-art results across multiple tasks. However, in inverse problems, standard initialization from pure Gaussian noise misaligns the generative process with real-world degradations. More recent methods such as diffusion bridges impose strict endpoint constraints and often require long reverse processes that are prone to hallucinations. Alternative consistency models provide noise-invariant, one-step mappings but lack inherent variance modeling and can degrade under severe corruption. Hence, residual diffusion implicit models (RDIMs) are proposed, constituting a generalized framework that explicitly models the residuals between high-quality (HQ) and low-quality (LQ) images, aligning the forward process with the actual degradation. A non-Markovian implicit reverse sampler is derived, which can skip intermediate timesteps, enabling accurate few-step or even single-step reconstruction, while mitigating the hallucinations inherent to long diffusion chains. RDIM also introduces a controllable variance mechanism that interpolates between deterministic and stochastic sampling, balancing fidelity and diversity. Furthermore, it enables the straightforward use of perceptual losses, when needed. Experiments on denoising and super-resolution benchmarks demonstrate that RDIMs consistently outperforms the state of the art, including bridge and consistency models, in terms of PSNR, SSIM, and LPIPS, reducing halucinations while requiring only a few sampling steps (often just one). The results position RDIMs as an efficient solution for a broad range of image restoration tasks.

## 1 INTRODUCTION

Image reconstruction is a fundamental problem in computer vision and signal processing, aiming to recover high-quality (HQ) images from corrupted observations. Tasks such as image denoising and super-resolution (SR) are crucial for numerous real-world applications, including medical and biological imaging, satellite imagery, and consumer photo enhancement (Sagheer & George, 2020; Wang et al., 2022; Delbracio et al., 2021).

Denoising diffusion probabilistic models (DDPMs) (Ho et al., 2020) have emerged as a powerful class of models for image synthesis and have been successfully adapted for image reconstruction. Their probabilistic formulation and iterative refinement enable them to handle challenging tasks by progressively improving predictions through small corrective updates (Saharia et al., 2022b). Moreover, their stochasticity enables the exploration of multiple plausible paths, promoting output diversity and often leading to better solutions (Lugmayr et al., 2022; Whang et al., 2022). These properties make diffusion models well-suited to deal with severe noise and information loss (Chung et al., 2023).

However, these strengths also introduce practical challenges. Although stochasticity is beneficial for capturing diversity and avoiding poor generalization (Lugmayr et al., 2022; Whang et al., 2022; Dhariwal & Nichol, 2021), excessive and uncontrolled variability can hinder convergence in inverse problems, destabilizing the reconstruction process and leading to inconsistent outputs. Therefore, balancing stochasticity is crucial (Chung et al., 2022). More critically, the standard DDPM formulation initializes the reverse process from pure noise, which is misaligned with reconstruction tasks where a degraded input already provides valuable information (Chung et al., 2022; Yue et al., 2023; Wu et al., 2024). Additionally, the recursive formulation of diffusion models leads to an inefficient reverse process requiring to traverse all diffusion timesteps, often hundreds (Shih et al., 2023; Liu et al., 2024), making them computationally expensive and impractical in latency-sensitive settings. Notably, techniques based on denoising diffusion bridge models (DDBMs) (Zhou et al., 2024) alleviate the

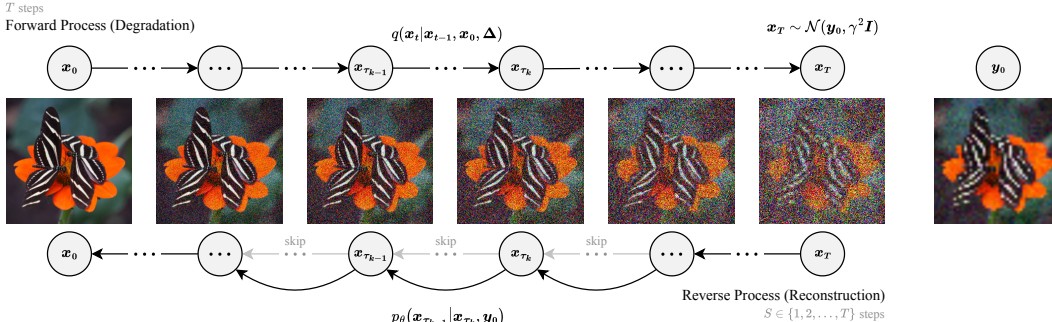

Figure 1: Overview of RDIM, a diffusion framework tailored for inverse problems, such as image reconstruction. The reverse process can accurately reconstruct back the data in $S \leq T$ steps.

noise–data mismatch by explicitly conditioning on degradation endpoints, but they also typically require iterating through all timesteps. While recent works tried to address this issue (Pan et al., 2025), we aim to further improve the reconstruction quality using a minimal number of diffusion steps.

To tackle these challenges, we revisit ResShift (Yue et al., 2023) and introduce a principled theoretical generalization, which we refer to as residual diffusion implicit model (RDIM). Our framework can be interpreted as a bridge-like approach as it constructs a process connecting a starting-point (low-quality (LQ) image) to an ending-point (HQ image) while preserving sample-level correspondences between the two domains, an essential property for SR and denoising tasks. A key feature of RDIM is its ability to introduce controlled stochasticity into the transportation between the two domains by relaxing the terminal constraint. This can be achieved by controllable variance mechanism that interpolates between deterministic and stochastic reconstructions. This provides greater modeling flexibility compared to the fixed terminal states typically imposed in diffusion bridge methods, allowing to obtain state-of-the-art results in SR and denoising applications. Moreover, an implicit sampling mechanism in the style of denoising diffusion implicit models (DDIMs) (Song et al., 2021) is introduced to allow skipping intermediate steps and improving the efficiency of the reconstruction process through few-step or even single-step HQ reconstructions. In summary, the main contributions of this paper are:

- A novel diffusion framework for inverse problems that generalizes ResShift, offering a bridge-like alignment, and provides an implicit formulation with efficient sampling, enabling reconstructions in a few or even on a single step.

- A controllable variance mechanism that interpolates between deterministic and stochastic reconstructions, balancing fidelity and diversity depending on degradation severity.

- Evidence that reducing the number of reverse steps accelerates inference and yields more faithful reconstructions by limiting hallucinations that arise in long diffusion chains.

- State-of-the-art results on denoising and SR benchmarks, showing that RDIMs outperforms existing methods while reducing the number of inference steps by up to $100\times$.

## 2 METHODOLOGY

RDIM is a diffusion framework tailored for inverse problems (herein focused on image reconstruction) where the forward process gradually degrades the original data into an informed corrupted version. The reverse process is efficient, allowing for a minimal number of steps (see Figure 1).

### 2.1 PROBLEM DEFINITION

Inverse problems are concerned with the recovery of a signal, $x_0 \in \mathcal{X}$, from a corrupted observation, $y_0 \in \mathcal{Y}$. Particularly, the forward model that degrades the original signal can be expressed as:

$$y_0 = \mathcal{F}(x_0),$$ (1)

where $\mathcal{F} : \mathcal{X} \rightarrow \mathcal{Y}$ is a known or unknown forward operator that often entails information loss (e.g., blurring, downsampling, masking, or noise). Accordingly, such problems are often ill-posed.

Meanwhile, deep learning (DL) techniques can be leveraged to learn a parametric reconstruction model $\mathcal{R} : \mathcal{Y} \rightarrow \mathcal{X}$, with trainable parameters $\Theta$, that invert the forward model:

$$\boldsymbol{x_0} \approx \mathcal{R}(\boldsymbol{y_0}; \Theta). \tag{2}$$

Traditional diffusion models reconstruct the signal $\boldsymbol{x_0}$ through a parameterized Markov chain with length $T$, which starts from pure noise and progressively denoises latent variables, $\boldsymbol{x_t}$, at each step $t \in \{1, 2, \ldots, T\}$. Hence, they first derive a diffusion process that transforms $\boldsymbol{x_0}$ into pure noise. Subsequently, they learn to reverse this process by training a parametric model, $p_\theta$, which can reconstruct $\boldsymbol{x_0}$ back from pure noise, $\boldsymbol{x_T} \sim \mathcal{N}(0, \boldsymbol{I})$, while conditioning on the corresponding degraded observation, $\boldsymbol{y_0}$. However, this diffusion process is fundamentally misaligned with the degradation model in Equation 1, since it maps $\boldsymbol{x_0}$ to pure noise rather than to the corrupted observation $\boldsymbol{y_0}$. In contrast, the RDIM forward process is explicitly designed to align with the degradation mechanism by progressively removing the residuals between the clean and corrupted signals while optionally injecting a controllable amount of noise. This stochastic component introduces variability that improves generalization, enabling the model to balance fidelity and diversity during reconstruction and better capture the uncertainty inherent in inverse problems.

## 2.2 Markovian Forward Process

Considering that $\boldsymbol{x_0}$ and $\boldsymbol{y_0}$ denote the original data and its corrupted version[1], respectively, the RDIM forward process (degradation) intends to gradually remove fractions of the residual, $\boldsymbol{\Delta} = \boldsymbol{x_0} - \boldsymbol{y_0}$, from $\boldsymbol{x_0}$ over a series of timesteps $t \in \{1, 2, \ldots, T\}$. For that purpose, a forward process fixed to a Markov chain is first defined, which converts the distribution of the original data, $q(\boldsymbol{x_0})$, into the last latent variable distribution. Following, the whole Markovian forward process is defined as:

$$q(\boldsymbol{x_{1:T}}|\boldsymbol{x_0}, \boldsymbol{\Delta}) = \prod_{t=1}^{T} q(\boldsymbol{x_t}|\boldsymbol{x_{t-1}}, \boldsymbol{\Delta}), \tag{3}$$

where all latent variables $\boldsymbol{x_1}, \ldots, \boldsymbol{x_T}$ have the same dimensionality as the original data, $\boldsymbol{x_0} \sim q(\boldsymbol{x_0})$.

The residual is removed from $\boldsymbol{x_0}$ according to a fixed weighting schedule $\lambda_1, \lambda_2, \ldots, \lambda_T$, which is also used to parameterize the variance in each diffusion transition distribution, defined as a Gaussian. Consequently, at each timestep $t$, the latent variable $\boldsymbol{x_t}$ is expressed in terms of the latent variable at the previous timestep, $\boldsymbol{x_{t-1}}$, and the residual, $\boldsymbol{\Delta}$, as follows:

$$q(\boldsymbol{x_t}|\boldsymbol{x_{t-1}}, \boldsymbol{\Delta}) = \mathcal{N}(\boldsymbol{x_t}|\boldsymbol{x_{t-1}} - \lambda_t \boldsymbol{\Delta}, \gamma^2 \lambda_t \boldsymbol{I}), \tag{4}$$

where $\gamma \in [0, \infty)$ is a constant hyperparameter introduced to control the strength of the variance, thus allowing interpolation between a deterministic (when $\gamma = 0$) and a stochastic ($\gamma > 0$) forward process. Moreover, each weight $\lambda_t$, used to control the amount of residual to be removed between each diffusion step, is computed in terms of small non-negative constant hyperparameters $\beta_0, \beta_1, \ldots, \beta_T$ as $\lambda_t = \beta_t - \beta_{t-1}$ (see Section 2.6 for details on the $\beta$-schedule).

Furthermore, to avoid a computationally expensive diffusion process, the cumulative forward transitions, $q(\boldsymbol{x_t}|\boldsymbol{x_0}, \boldsymbol{\Delta}) = q(\boldsymbol{x_t}|\boldsymbol{y_0})$, are expressed in closed form by relying on the reparameterization trick (see Appendix A.1):

$$q(\boldsymbol{x_t}|\boldsymbol{x_0}, \boldsymbol{\Delta}) = \mathcal{N}(\boldsymbol{x_t}|\boldsymbol{x_0} - \beta_t \boldsymbol{\Delta}, \gamma^2 \beta_t \boldsymbol{I}). \tag{5}$$

Although this forward process matches ResShift (Yue et al., 2023), the corresponding recursive formulation yields an inefficient reverse process that must iterate over many timesteps (particularly for HQ inverse problems). Therefore, a DDIM-inspired non-Markovian forward process is derived, which preserves the marginal in Eq. (5) while still allowing a Markovian reverse process.

---

[1]To match dimensionalities, $\boldsymbol{y_0}$ is upsampled for SR tasks and its channels are replicated for colorization.

## 2.3 Non-Markovian Forward Process

The forward process is implicitly constructed to ensure consistency with the marginal $q(\boldsymbol{x_t}|\boldsymbol{x_0}, \boldsymbol{\Delta})$ and the reverse process. As a result, each forward transition becomes additionally conditioned on $\boldsymbol{x_0}$ rather than just on the immediate previous timestep, $\boldsymbol{x_{t-1}}$, and the residual, $\boldsymbol{\Delta}$. This introduces explicit dependency on the initial data $\boldsymbol{x_0}$, decoupling the forward process from strict Markovian constraints. Moreover, the forward process is expressed in terms of the forward transition posterior, $q(\boldsymbol{x_{t-1}}|\boldsymbol{x_t}, \boldsymbol{x_0}, \boldsymbol{\Delta})$, further reflecting the non-Markovian behavior and preservation of $q(\boldsymbol{x_t}|\boldsymbol{x_0}, \boldsymbol{\Delta})$. Therefore, although the RDIM forward process is still a distribution over trajectories that start at $\boldsymbol{x_0}$ and end at $\boldsymbol{x_T}$, it is defined as a joint distribution that is factored in reverse[2]:

$$q(\boldsymbol{x_{1:T}}|\boldsymbol{x_0}, \boldsymbol{\Delta}) = q(\boldsymbol{x_T}|\boldsymbol{x_0}, \boldsymbol{\Delta}) \prod_{t=2}^{T} q(\boldsymbol{x_{t-1}}|\boldsymbol{x_t}, \boldsymbol{x_0}, \boldsymbol{\Delta}). \tag{6}$$

The non-Markovian nature of the forward process enables designing a reverse process that can be deterministic and simulated with a reduced number of transitions due to the conditioning on $\boldsymbol{x_0}$. In addition, since the ResShift training objective only depends on the marginal distribution, $q(\boldsymbol{x_t}|\boldsymbol{x_0}, \boldsymbol{\Delta})$, which is preserved, then RDIM optimization (see Section 2.7) will lead to the same training objective as ResShift. Consequently, already trained ResShift models can be leveraged for RDIM sampling without requiring additional retraining.

## 2.4 Reverse Process

The reverse process (reconstruction) intends to revert the forward process, thus sampling back the data, $\boldsymbol{x_0}$. This is achieved by starting from $\boldsymbol{x_T} \sim \mathcal{N}(\boldsymbol{y_0}, \gamma^2\boldsymbol{I})$ and iteratively refining the latent variables $\boldsymbol{x_t}$ until $\boldsymbol{x_0}$ is reached. Accordingly, the reverse process involves computing the forward transition posterior $q(\boldsymbol{x_{t-1}}|\boldsymbol{x_t}, \boldsymbol{x_0}, \boldsymbol{\Delta})$ (reverse transition), defined as a Gaussian distribution:

$$q(\boldsymbol{x_{t-1}}|\boldsymbol{x_t}, \boldsymbol{x_0}, \boldsymbol{\Delta}) = \mathcal{N}(\boldsymbol{x_{t-1}}|\tilde{\boldsymbol{\mu}}_t, \tilde{\sigma}_t^2\boldsymbol{I}), \tag{7}$$

where $\tilde{\boldsymbol{\mu}}_t$ is the mean of the Gaussian distribution and $\tilde{\sigma}_t^2\boldsymbol{I} = \tilde{\boldsymbol{\Sigma}}_t$ is the isotropic covariance matrix. Particularly, the reverse transition is designed to preserve the marginal $q(\boldsymbol{x_t}|\boldsymbol{x_0}, \boldsymbol{\Delta})$ (see Appendix A.2). Considering $\tilde{\sigma}_t^2$ matches the ResShift variance, $\tilde{\lambda}_t = \gamma^2 \frac{\beta_{t-1}}{\beta_t}\lambda_t$, the mean, $\tilde{\boldsymbol{\mu}}_t$, is given as:

$$\tilde{\boldsymbol{\mu}}_t = \begin{cases} \boldsymbol{x_0} - \beta_{t-1}\boldsymbol{\Delta}, & \text{if } \gamma = 0, \\ \boldsymbol{x_0} - \beta_{t-1}\boldsymbol{\Delta} + \sqrt{\gamma^2\beta_{t-1} - \tilde{\sigma}_t^2}\left(\frac{\boldsymbol{x_t} - \boldsymbol{x_0} + \beta_t\boldsymbol{\Delta}}{\sqrt{\gamma^2\beta_t}}\right), & \text{if } \gamma \neq 0, \end{cases} \tag{8}$$

where, for $\gamma = 0$, the reverse process essentially becomes a linear interpolation between the corrupted and original data, which underscores that the RDIM forward process is aligned with a forward model (degradation process) that converts $\boldsymbol{x_0}$ into $\boldsymbol{y_0}$.

Furthermore, fixing $\tilde{\sigma}_t^2$ to the ResShift variance, $\tilde{\lambda}_t$, also results in $\tilde{\boldsymbol{\mu}}_t$ matching the mean of the ResShift reverse transition (see Appendix A.3). Hence, RDIM becomes ResShift for this specific variance, revealing that ResShift is a particular case of RDIM. Subsequently, a constant hyperparameter, $\eta \in [0, 1]$, can be introduced to interpolate between a deterministic ($\eta=0$) and a stochastic ($\eta>0$) reverse process when $\gamma \neq 0$, allowing control over the variability in the RDIM reverse trajectory:

$$\tilde{\boldsymbol{\mu}}_{t|\gamma \neq 0} = \boldsymbol{x_0} - \beta_{t-1}\boldsymbol{\Delta} + \sqrt{\gamma^2\beta_{t-1} - \eta^2\tilde{\lambda}_t}\left(\frac{\boldsymbol{x_t} - \boldsymbol{x_0} + \beta_t\boldsymbol{\Delta}}{\sqrt{\gamma^2\beta_t}}\right), \quad \tilde{\sigma}_{t|\gamma \neq 0}^2 = \eta^2\tilde{\lambda}_t. \tag{9}$$

where, $\eta = 1$ makes the RDIM reverse process identical to ResShift. Meanwhile, setting $\gamma = 0$ converts RDIM into a strictly deterministic model ($\gamma=0 \Rightarrow \tilde{\lambda}_t=0$), avoiding sampling random noise.

However, during inference, $\boldsymbol{x_0}$ and $\boldsymbol{\Delta}$ are unknown, thus sampling from the true reverse transition distribution is not possible. Therefore, a learnable parametric model, $p_\theta(\boldsymbol{x_{t-1}}|\boldsymbol{x_t}, \boldsymbol{y_0})$, defined as a Gaussian distribution, is introduced to approximate the true reverse transition $q(\boldsymbol{x_{t-1}}|\boldsymbol{x_t}, \boldsymbol{x_0}, \boldsymbol{\Delta})$:

$$p_\theta(\boldsymbol{x_{t-1}}|\boldsymbol{x_t}, \boldsymbol{y_0}) = \mathcal{N}\left(\boldsymbol{x_{t-1}}|\boldsymbol{\mu_\theta}\left(\boldsymbol{x_t}, \boldsymbol{y_0}, t\right), \sigma_\theta^2\left(\boldsymbol{x_t}, \boldsymbol{y_0}, t\right)\boldsymbol{I}\right), \tag{10}$$

---

[2]The forward transition, $q(\boldsymbol{x_t}|\boldsymbol{x_{t-1}}, \boldsymbol{x_0}, \boldsymbol{\Delta})$, can be derived via Bayes' rule.

where $\boldsymbol{\mu_\theta}\left(\boldsymbol{x_t}, \boldsymbol{y_0}, t\right)$ is the mean of the Gaussian distribution and $\sigma_\theta^2\left(\boldsymbol{x_t}, \boldsymbol{y_0}, t\right)\boldsymbol{I} = \boldsymbol{\Sigma_\theta}\left(\boldsymbol{x_t}, \boldsymbol{y_0}, t\right)$ is the isotropic covariance matrix. In particular, the variance of the true reverse transition, $\tilde{\sigma}_t^2$, does not have any learnable parameters because it is defined in terms of constant hyperparameters, which are known. Therefore, the variance of $p_\theta(\boldsymbol{x_{t-1}}|\boldsymbol{x_t}, \boldsymbol{y_0})$ can be fixed to equal exactly the variance of $q(\boldsymbol{x_{t-1}}|\boldsymbol{x_t}, \boldsymbol{x_0}, \boldsymbol{\Delta})$:

$$\sigma_\theta^2\left(\boldsymbol{x_t}, \boldsymbol{y_0}, t\right) = \tilde{\sigma}_t^2. \tag{11}$$

Meanwhile, $\boldsymbol{\mu_\theta}\left(\boldsymbol{x_t}, \boldsymbol{y_0}, t\right)$ approximates the mean of the true reverse transition, $\tilde{\boldsymbol{\mu}}_t$. Considering that $\boldsymbol{x_0}$ and $\boldsymbol{\Delta}$ are the only unknown terms and $\boldsymbol{\Delta}$ can be estimated from $\boldsymbol{x_0}$ and $\boldsymbol{y_0}$, then the model solely needs to predict $\boldsymbol{x_0}$ (see Appendix A.4). Accordingly, the mean $\boldsymbol{\mu_\theta}\left(\boldsymbol{x_t}, \boldsymbol{y_0}, t\right)$ is defined as:

$$\boldsymbol{\mu_\theta}\left(\boldsymbol{x_t}, \boldsymbol{y_0}, t\right) = \begin{cases} \hat{\boldsymbol{x}}_{\boldsymbol{0}} - \beta_{t-1}\hat{\boldsymbol{\Delta}}, & \text{if } \gamma = 0, \\ \hat{\boldsymbol{x}}_{\boldsymbol{0}} - \beta_{t-1}\hat{\boldsymbol{\Delta}} + \sqrt{\gamma^2\beta_{t-1} - \eta^2\tilde{\lambda}_t}\left(\frac{\boldsymbol{x_t} - \hat{\boldsymbol{x}}_{\boldsymbol{0}} + \beta_t\hat{\boldsymbol{\Delta}}}{\sqrt{\gamma^2\beta_t}}\right), & \text{if } \gamma \neq 0, \end{cases} \tag{12}$$

where $\hat{\boldsymbol{x}}_{\boldsymbol{0}} = f_\theta(\boldsymbol{x_t}, \boldsymbol{y_0}, t)$ denotes the $\boldsymbol{x_0}$ prediction from a neural network given $\boldsymbol{x_t}$, $\boldsymbol{y_0}$, and timestep $t$. The neural network is parameterized by weights $\theta$ and $\hat{\boldsymbol{\Delta}} = \hat{\boldsymbol{x}}_{\boldsymbol{0}} - \boldsymbol{y_0}$ represents the $\boldsymbol{\Delta}$ estimation. Hence, the whole approximate reverse process is expressed by the following joint distribution:

$$p_\theta(\boldsymbol{x_{0:T}}|\boldsymbol{y_0}) = p(\boldsymbol{x_T}|\boldsymbol{y_0})\prod_{t=1}^{T}p_\theta(\boldsymbol{x_{t-1}}|\boldsymbol{x_t}, \boldsymbol{y_0}). \tag{13}$$

## 2.5 Long-Range Reverse Transition

Particularly, the derived reverse transition structurally matches the reparameterized form of the marginal $q(\boldsymbol{x_{t-1}}|\boldsymbol{x_0}, \boldsymbol{\Delta})$ (see Appendix A.2), which models the cumulative transitions from $\boldsymbol{x_0}$ to $\boldsymbol{x_{t-1}}$ in the forward process. Therefore, the reverse transition formulation aligns with the concept of cumulative transitions, allowing the reverse process to efficiently sample any state at an arbitrary timestep $\tau_{k-1} \in \{0, 1, \ldots, T-1\}$ by skipping intermediate latent variables in the reverse trajectory. Accordingly, the reverse process can be simulated with fewer timesteps, thereby accelerating sampling. Using the reparameterization trick, $\boldsymbol{x_{\tau_{k-1}}} \sim p_\theta\left(\boldsymbol{x_{\tau_{k-1}}}|\boldsymbol{x_{\tau_k}}, \boldsymbol{y_0}\right)$ can be sampled as follows:

$$\boldsymbol{x_{\tau_{k-1}}} = \begin{cases} \hat{\boldsymbol{x}}_{\boldsymbol{0}} - \beta_{\tau_{k-1}}\hat{\boldsymbol{\Delta}}, & \text{if } \gamma = 0, \\ \hat{\boldsymbol{x}}_{\boldsymbol{0}} - \beta_{\tau_{k-1}}\hat{\boldsymbol{\Delta}} + \sqrt{\gamma^2\beta_{\tau_{k-1}} - \eta^2\tilde{\lambda}_{\tau_k}}\hat{\boldsymbol{\epsilon}} + \sqrt{\eta^2\tilde{\lambda}_{\tau_k}}\boldsymbol{z}, & \text{if } \gamma \neq 0, \end{cases} \tag{14}$$

where $(\tau_{k-1}, \tau_k) \in \left\{(t', t) \in \mathbb{N}_0^2 \mid t' + 1 \leq t \leq T\right\}$, $\boldsymbol{z} \sim \mathcal{N}\left(0, \boldsymbol{I}\right)$, and $\hat{\boldsymbol{\epsilon}}$ is expressed by the following relationship when $\gamma \neq 0$ (see Equation (32) in Appendix A.2):

$$\hat{\boldsymbol{\epsilon}} = \frac{\boldsymbol{x_{\tau_k}} - \hat{\boldsymbol{x}}_{\boldsymbol{0}} + \beta_{\tau_k}\hat{\boldsymbol{\Delta}}}{\sqrt{\gamma^2\beta_{\tau_k}}}. \tag{15}$$

Essentially, each iteration of the reverse process involves predicting the original data sample, $\boldsymbol{x_0}$. This estimate is then used to compute the residual $\boldsymbol{\Delta}$ and the noise component $\boldsymbol{\epsilon}$, which together guide the update to the next less-degraded state, $\boldsymbol{x_{\tau_{k-1}}}$. As the reverse process progresses, the model gradually refines its prediction of $\boldsymbol{x_0}$ at each step, leveraging the increasingly accurate intermediate states. This iterative refinement culminates in an accurate prediction of $\boldsymbol{x_0}$. Moreover, the ability of the reverse process to skip intermediate steps not only enables few-step generation but also allows one-step predictions, thus demonstrating the efficiency and flexibility of the RDIM sampling procedure. Here, the number of sampling timesteps along the reverse trajectory, $S \in \{1, 2, \ldots, T\}$, is set arbitrarily. For each case, a uniform schedule is used, as detailed in Appendix C.4.

## 2.6 Residual $\beta$-Schedule

The residual $\beta$-schedule employed is defined by a circular curve (similar to the fourth quadrant $p$-norm shape), ensuring a smooth and adjustable transition between $\boldsymbol{x_0}$ and $\boldsymbol{x_T}$:

$$\beta_t = \frac{t}{T + (p-1)(T-t)}, \tag{16}$$

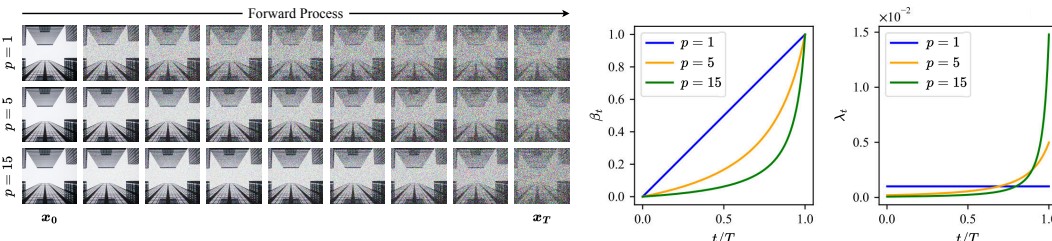

Figure 2: Progression of weights $\beta_t$ and $\lambda_t$ across timesteps and impact of $p$ on the diffusion process.

where $p \in (0, \infty)$ is a parameter that allows controlling the steepness of the curve. As it increases the $\beta$-schedule exhibits a slower initial progression, followed by a rapid increase to larger and more pronounced updates. This design allows for a gentle removal of the residual and injection of noise in the early timesteps of the forward process, which become progressively more aggressive throughout the diffusion trajectory. Figure 2 illustrates the impact of the parameter $p$ on the diffusion process.

Furthermore, this choice for the $\beta$-schedule ensures that $\beta_0 = 0$ and $\beta_T = 1$, such that the residual, $\boldsymbol{\Delta}$, is fully removed from $\boldsymbol{x_0}$ after exactly $T$ timesteps. As a result, the last latent variable, $\boldsymbol{x_T}$, converges to a noisy sample centered at the corrupted data, $\boldsymbol{y_0}$. Additionally, since $\beta_0 = 0$, it follows that when $\gamma \neq 0$ the variance of any reverse transition distribution from $\boldsymbol{x_{\tau_k}}$ to $\boldsymbol{x_0}$ is $\eta^2 \tilde{\lambda}_{\tau_k} = \eta^2 \gamma^2 \frac{\beta_0}{\beta_{\tau_k}} \lambda_{\tau_k} = 0$. Therefore, $p_\theta(\boldsymbol{x_0}|\boldsymbol{x_{\tau_k}}, \boldsymbol{y_0})_{\gamma \neq 0}$ degenerates into a $\delta$-distribution centered at $\hat{\boldsymbol{x}}_0$. Logically, under these conditions, $\eta$ does not have any impact on the last transition of the reverse process.

## 2.7 OPTIMIZATION

At each step of the sampling process, the neural network parameterized by weights $\theta$ yields an estimate of $\boldsymbol{x_0}$. During training, these parameters are learned to assure that the model marginal $p_\theta(\boldsymbol{x_0}|\boldsymbol{y_0})$ fits the true posterior distribution $q(\boldsymbol{x_0}|\boldsymbol{y_0})$ via:

$$q(\boldsymbol{x_0}|\boldsymbol{y_0}) \approx p_\theta(\boldsymbol{x_0}|\boldsymbol{y_0}) = \int p(\boldsymbol{x_T}|\boldsymbol{y_0}) \prod_{t=1}^{T} p_\theta(\boldsymbol{x_{t-1}}|\boldsymbol{x_t}, \boldsymbol{y_0}) \, d\boldsymbol{x_{1:T}}, \qquad (17)$$

which ensures, during inference, that the data, $\boldsymbol{x_0}$, can be sampled back accurately given $\boldsymbol{y_0}$. Accordingly, $p_\theta(\boldsymbol{x_{t-1}}|\boldsymbol{x_t}, \boldsymbol{y_0})$ is required to closely approximate the true forward transition posterior, $q(\boldsymbol{x_{t-1}}|\boldsymbol{x_t}, \boldsymbol{x_0}, \boldsymbol{\Delta})$. This is achieved by minimizing the Kullback–Leibler (KL) divergence between both distributions, while accounting for all timesteps. In fact, this objective can be reduced for simplicity to (see Appendix A.4):

$$\mathcal{L}_{\text{simple}}(\theta) = \mathbb{E}_{\boldsymbol{x_0}, \boldsymbol{\Delta}, t} \left[ \|\boldsymbol{x_0} - \hat{\boldsymbol{x}}_0\|^2 \right]. \qquad (18)$$

Notably, ResShift shares the same training objective as RDIM, further highlighting that ResShift is a particular case of RDIM and that its trained models can be used for RDIM sampling without retraining. The RDIM training and sampling procedures are described in Algorithms 1 and 2, respectively.

---

**Algorithm 1** Training

1: **repeat**
2:     $\boldsymbol{x_0}, \boldsymbol{y_0} \sim q(\boldsymbol{x_0}, \boldsymbol{y_0}) = q(\boldsymbol{x_0})q(\boldsymbol{y_0}|\boldsymbol{x_0})$
3:     $\boldsymbol{\Delta} = \boldsymbol{x_0} - \boldsymbol{y_0}$
4:     $t \sim \mathcal{U}(1, T)$
5:     $\epsilon \sim \mathcal{N}(0, \boldsymbol{I})$
6:     $\boldsymbol{x_t} \sim q(\boldsymbol{x_t}|\boldsymbol{x_0}, \boldsymbol{\Delta})$
7:     $\hat{\boldsymbol{x}}_0 = f_\theta(\boldsymbol{x_t}, \boldsymbol{y_0}, t)$
8:     $\mathcal{L} = \|\boldsymbol{x_0} - \hat{\boldsymbol{x}}_0\|^2$
9:     Take gradient descent step on $\nabla_\theta \mathcal{L}$
10: **until** convergence
11: **return** $f_\theta$

**Algorithm 2** Sampling

1: $\Upsilon = \{\tau_S = T, \tau_{S-1}, \dots, \tau_1, \tau_0 = 0\}$
2: $\boldsymbol{x_T} \sim \mathcal{N}(\boldsymbol{y_0}, \gamma^2 \boldsymbol{I})$
3: **for** $k = S, S-1, \dots, 1$ **do**
4:     $\hat{\boldsymbol{x}}_0 = f_\theta(\boldsymbol{x_{\tau_k}}, \boldsymbol{y_0}, \tau_k)$
5:     $\hat{\boldsymbol{\Delta}} = \hat{\boldsymbol{x}}_0 - \boldsymbol{y_0}$
6:     **if** $\gamma \neq 0$ **then** $\hat{\epsilon} = \frac{\boldsymbol{x_{\tau_k}} - \hat{\boldsymbol{x}}_0 + \beta_{\tau_k} \hat{\boldsymbol{\Delta}}}{\sqrt{\gamma^2 \beta_{\tau_k}}}$
7:     $\boldsymbol{x_{\tau_{k-1}}} \sim p_\theta\left(\boldsymbol{x_{\tau_{k-1}}}|\boldsymbol{x_{\tau_k}}, \boldsymbol{y_0}\right)$
8: **end for**
9: **return** $\boldsymbol{x_0}$

---

## 3 EXPERIMENTS

RDIM is evaluated on image denoising and single image SR using the FMD (Zhang et al., 2019), SIDD (Abdelhamed et al., 2018; 2019), and DIV2K (Agustsson & Timofte, 2017; Timofte et al., 2017) datasets. Several RDIM variants with $\gamma = 3.0$, $\eta = 1.0$, and $p = 5.0$ are considered, differing only in the number of sampling timesteps and loss targets. RDIM-PQ stands for RDIM trained with a perceptual quality (PQ) objective. RDIM-$S$ and RDIM-PQ-$S$ denote sampling with $S$ steps. Particularly, RDIM-1 and RDIM-PQ-1 correspond to single-step deterministic inferences ($S=1$). Their deterministic nature results from the final reverse transition degenerating into a $\delta$-distribution when $\gamma \neq 0$ and $\beta_0 = 0$ (see Sections 2.5 and 2.6). Moreover, RDIM is compared against multiple state-of-the-art methods, including ResShift with $S = T = 100$. Although ResShift is often employed with $S = T = 10$, there is a significant performance improvement when using longer diffusion chains. This effect is evident in the experiments shown in Appendix C.7, where ResShift improves peak signal-to-noise ratio (PSNR) from 39.363 dB for $T = 10$ to 43.599 dB for $T = 100$. Additional qualitative results on image inpainting, colorization, and deblurring are provided on FFHQ (Karras et al., 2019). Experimental details are in Appendix C, including RDIM assessment when varying $S$ (Figure 10).

**Image Denoising.** RDIM is compared against BM3D (Dabov et al., 2007), DnCNN (Zhang et al., 2017), DDPM (Ho et al., 2020), DDIM (Song et al., 2021), and ResShift (Yue et al., 2023). For fairness, all diffusion models use the same network architecture (see Appendix C.2) and diffusion timesteps ($T = 100$). Results are listed in Table 1a. On FMD-Confocal-BPAE-Raw, RDIM-10 achieves the best results in terms of PSNR and structural similarity index measure (SSIM), followed by RDIM-1. On FMD-Confocal-Zebrafish-Raw, ResShift attains the best PSNR score, but is $10\times$ slower than RDIM-10, which obtains comparable PSNR performance and the best SSIM score. On SIDD-Medium, RDIM-1 yields superior results. Diffusion models, which inherently capture richer structures than DnCNN, have their gains diminished on SIDD-Medium due to the use of a small $64\times64$ patch size (which is kept the same across all experiments for consistency). Meanwhile, DnCNN performs full-image processing at inference, giving it a slight unfair advantage. Figure 12, Appendix C.11, presents a qualitative comparison.

**Super-Resolution.** A comparative analysis with $\times2$ and $\times4$ downsampling factors evaluates RDIM against ESRGAN (Wang et al., 2018), DDPM, DDIM, and ResShift. As before, diffusion models were trained under the same conditions, including architecture and diffusion timesteps ($T = 100$). Results are shown in Table 1b. On both DIV2K-Unknown-$\times2$ and DIV2K-Unknown-$\times4$, RDIM-1 performs the best, followed by RDIM-10, highlighting that RDIM consistently surpasses ResShift and DDPM. Figure 13, in Appendix C.11 showcases qualitative results.

An additional analysis is conducted on $\times4$ bicubic downsampled images from the DIV2K dataset, comparing RDIM against DDRM (Kawar et al., 2022), ResShift, IR-SDE (Luo et al., 2023b), DDBM (Zhou et al., 2024), GOUB (Yue et al., 2024), UniDB (Zhu et al., 2025), CTMSR (You et al., 2025), MaRS (Li et al., 2025), DBIM (Zheng et al., 2024), and UniDB++ (Pan et al., 2025). Results are presented in Table 2. RDIM-1 achieves the highest PSNR and SSIM among all methods, with RDIM-PQ-1 following closely. This suggests that RDIM offers an advantage for applications where

Table 1: Comparative analysis of RDIM against relevant state-of-the-art techniques for (a) denoising and (b) SR. Green color highlights the best score overall and Blue color the second best.

(a) Denoising on images from the FMD (BPAE and zebrafish confocal fluorescence microscopy images) and SIDD datasets.

| Denoising Method | S↓ | FMD-BPAE PSNR↑ | FMD-BPAE SSIM↑ | FMD-Zebrafish PSNR↑ | FMD-Zebrafish SSIM↑ | SIDD-Medium PSNR↑ | SIDD-Medium SSIM↑ |
|---|---|---|---|---|---|---|---|
| Noisy | – | 31.596 | 0.812 | 26.732 | 0.603 | 27.797 | 0.515 |
| BM3D | – | 35.862 | 0.933 | 35.289 | 0.918 | 35.880 | 0.906 |
| DnCNN | – | 37.609 | 0.950 | 37.169 | 0.941 | 39.838 | 0.957 |
| DDPM | 100 | 41.775 | 0.981 | 43.214 | 0.974 | 39.329 | 0.945 |
| DDIM-25 | 25 | 35.168 | 0.953 | 39.060 | 0.960 | 28.627 | 0.855 |
| DDIM-50 | 50 | 38.608 | 0.972 | 41.211 | 0.969 | 34.665 | 0.912 |
| ResShift | 100 | 43.599 | 0.984 | **45.167** | 0.976 | 39.663 | 0.949 |
| RDIM-1 | 1 | 43.987 | 0.985 | 44.229 | 0.976 | **40.335** | **0.962** |
| RDIM-10 | 10 | **44.147** | **0.986** | 45.027 | **0.978** | 39.979 | 0.958 |

(b) $\times2$ and $\times4$ SR on images from the DIV2K dataset under unknown degradations.

| SR Method | S↓ | DIV2K-$\times2$ PSNR↑ | DIV2K-$\times2$ SSIM↑ | DIV2K-$\times4$ PSNR↑ | DIV2K-$\times4$ SSIM↑ |
|---|---|---|---|---|---|
| LR (Bicubic) | – | 25.112 | 0.704 | 21.742 | 0.574 |
| ESRGAN | – | 30.017 | 0.857 | 24.957 | 0.690 |
| DDPM | 100 | 31.949 | 0.893 | 26.446 | 0.739 |
| DDIM-25 | 25 | 29.003 | 0.839 | 20.894 | 0.504 |
| DDIM-50 | 50 | 31.150 | 0.879 | 24.949 | 0.687 |
| ResShift | 100 | 32.368 | 0.903 | 26.627 | 0.750 |
| RDIM-1 | 1 | **33.887** | **0.924** | **28.280** | **0.798** |
| RDIM-10 | 10 | 33.019 | 0.914 | 27.266 | 0.770 |

distortion fidelity is critical (e.g., medical imaging). Notably, RDIM-PQ-1 explicitly optimizes for LPIPS and attains the lowest score on this metric while attaining high PSNR and SSIM. Overall, a clear perception–distortion trade-off emerges, as further illustrated in Appendix C.8, where RDIM demonstrates a more favorable balance than state-of-the-art alternatives.

A qualitative comparison in Figure 3 further demonstrates that RDIM yields sharper and more faithful reconstructions, particularly in areas rich in fine textures and structural detail. Other methods introduce noticeable artifacts and deformations. More results are shown in Figures 14 and 15 (Appendix C.11).

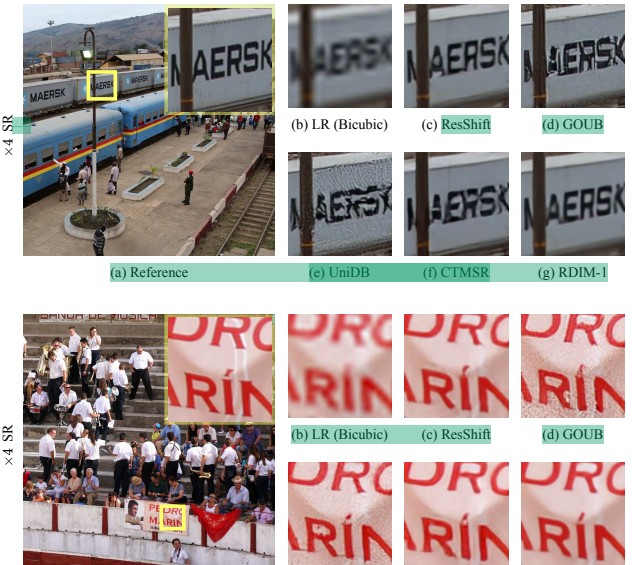

Figure 3: Qualitative ×4 SR results on DIV2K-Bicubic-×4.

Table 2: ×4 SR on bicubic downsampled images from the DIV2K dataset.

| SR Method | S↓ | DIV2K-×4 | | |
|---|---|---|---|---|
| | | PSNR↑ | SSIM↑ | LPIPS↓ |
| DDRM[†] | 100 | 24.350 | 0.592 | 0.364 |
| ResShift | 100 | 27.455 | 0.780 | 0.153 |
| IR-SDE[†] | 100 | 25.900 | 0.657 | 0.231 |
| DDBM[‡] | 100 | 24.210 | 0.581 | 0.384 |
| GOUB-SDE[†] | 100 | 26.890 | 0.748 | 0.220 |
| GOUB-ODE[†] | 100 | 28.500 | 0.807 | 0.328 |
| UniDB-SDE[†] | 100 | 25.460 | 0.686 | 0.179 |
| UniDB-ODE[†] | 100 | 28.640 | 0.807 | 0.323 |
| UniDB++-50[‡] | 50 | 26.610 | 0.754 | 0.159 |
| UniDB++-20[‡] | 20 | 27.380 | 0.777 | 0.179 |
| UniDB++-5[‡] | 5 | 28.400 | 0.805 | 0.235 |
| MaRS-5[‡] | 5 | 27.730 | 0.783 | 0.286 |
| DBIM-5[‡] | 5 | 28.050 | 0.795 | 0.260 |
| CTMSR-1 | 1 | 27.087 | 0.759 | 0.130 |
| RDIM-1 | 1 | **29.180** | **0.824** | 0.257 |
| RDIM-5 | 5 | 28.408 | 0.806 | 0.197 |
| RDIM-10 | 10 | 27.963 | 0.795 | 0.178 |
| RDIM-20 | 20 | 27.636 | 0.786 | 0.166 |
| RDIM-50 | 50 | 27.427 | 0.779 | 0.154 |
| RDIM-PQ-1 | 1 | 29.004 | 0.817 | **0.114** |

[†] Retrieved from Zhu et al. (2025).
[‡] Retrieved from Pan et al. (2025).

**Additional Image Restoration Tasks.** Further evaluation on image inpainting, colorization, and deblurring tasks demonstrates the generalization capabilities of RDIM. Figure 4 presents qualitative results obtained with RDIM-10. Additional details and results are provided in Appendix C.

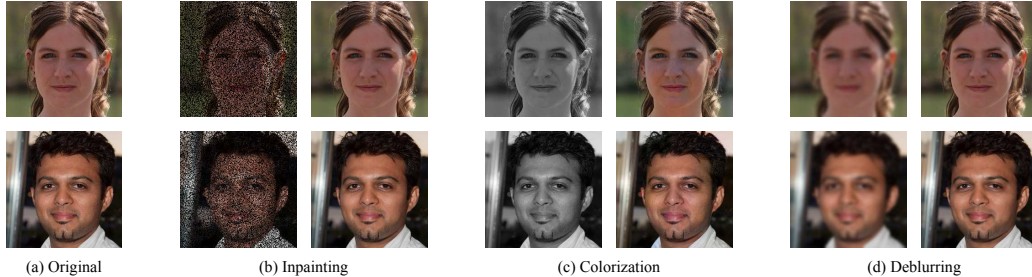

Figure 4: RDIM-10 results in image inpainting, colorization, and deblurring on the FFHQ dataset. In (b), (c) and (d), the left side represents the input image and the right side the output.

**Discussion of results.** RDIM and ResShift consistently outperform DDPM, emphasizing that their diffusion process is more closely aligned with these inverse problems. Moreover, RDIM demonstrates performance comparable to ResShift, often surpassing it, while requiring significantly fewer sampling timesteps. This stands in contrast with the DDIM behavior when applied to a pretrained DDPM. DDIM incurs noticeable degradation when reducing the sampling count from $S = 100$ to $S = 50$, and does not support reliable single-step inference. This suggests that the residual modeling dynamics are intrinsically well aligned with implicit sampling, enabling acceleration factors (up to ×100) that are not achievable by applying DDIM to DDPM. Moreover, since DDPM and ResShift require a reverse process with the same number of timesteps as their forward diffusion process, reducing their diffusion

steps to match the RDIM sampling time would result in a degradation in performance (Shih et al., 2023). This effect is evident in the experiments conducted in Appendix C.7, where ResShift with a reduced number of diffusion steps underperforms compared to its higher-timestep configurations.

Furthermore, FMD-Confocal-Zebrafish-Raw contains noisier images than FMD-Confocal-BPAE-Raw. As shown in Table 1a, RDIM-1 outperforms ResShift on FMD-Confocal-BPAE-Raw, whereas ResShift performs better on FMD-Confocal-Zebrafish-Raw. This suggests that in the presence of stronger degradations a more stochastic approach is advantageous, as variability promotes output diversity. Conversely, when degradations are mild, a more deterministic method ensures consistent and accurate restoration. Therefore, balancing stochasticity is crucial to adapt the method effectively to varying noise levels and degradation strengths. Notably, RDIM-10 achieves comparable results to ResShift in FMD-Confocal-Zebrafish-Raw while requiring only 10 sampling steps instead of 100, rendering inference $10\times$ faster. Further demonstrating its efficiency, RDIM accelerates sampling up to $100\times$ compared to ResShift and DDPM on FMD-Confocal-BPAE-Raw. Additionally, experiments on SIDD highlight that RDIM effectively supports high-resolution (HR) image reconstruction even when operating on relatively small patches (e.g., $64\times64$) compared to the full image size, which here reach resolutions of up to $\approx 5300\times3000$ pixels. Naturally, increasing the patch size will improve performance and could enable restoration of images at even higher resolutions.

In SR under unknown degradations, standard diffusion models and ResShift, often exhibit a tendency to hallucinate details that deviate from the ground truth, particularly when employing long diffusion chains. As illustrated in Figure 13, while iterative refinement encourages the generation of natural-looking textures, it frequently trades off fidelity for perceptual quality, leading to reconstructions that drift away from the original structure (see Appendix C.7 for further evidence). Furthermore, the deterministic RDIM-1 outperforms all methods, suggesting a more deterministic approach to SR is beneficial, as too much stochasticity can introduce unwanted variability in the output and the iterative refinement of long-chain diffusion can become detrimental. A similar trend is observed on the DIV2K-Bicubic-$\times4$ benchmark. As shown in Table 2, RDIM again achieves the highest PSNR and SSIM scores while operating with far fewer sampling steps. It maintains sharper and more faithful textures, as illustrated in Figure 3. These results confirm that the advantages of residual-based implicit sampling carry over to classical SR settings.

## 4  RELATED WORK

**Diffusion Models** (DMs) (Sohl-Dickstein et al., 2015; Song & Ermon, 2019; Song et al., 2020) generate images by iteratively denoising latent variables sampled from a Gaussian prior. For image-to-image tasks, conditioning mechanisms such as classifier guidance (Dhariwal & Nichol, 2021) or classifier-free guidance (Ho & Salimans) enable the generation of target images given source observations (Saharia et al., 2022a; Sasaki et al., 2021; Zhao et al., 2022). However, because DMs start from pure noise, they remain misaligned with inverse problems where the input already contains meaningful structure. Hence, several diffusion-based reconstruction approaches adapt the generative process to low-quality inputs. SR3 (Saharia et al., 2022b) and SRDiff (Li et al., 2022) condition DDPMs on low-resolution inputs, while Whang et al. (2022) use residual-based refinements to improve deblurring. DDRM (Kawar et al., 2022) addresses general linear inverse problems via posterior sampling with a pre-trained DM, and ResShift (Yue et al., 2023) leverages residual modeling between high-resolution and low-resolution images. Despite their effectiveness, these methods still require traversing all diffusion steps sequentially. RDIM generalizes residual modeling while enabling DDIM-style long-range sampling and controllable stochasticity, significantly reducing the number of steps needed for high-quality reconstruction.

**Accelerating DM Sampling** have become an attractive research area, usually focusing on reducing the number of steps to a dozen or fewer. Within the body of work, training-based distillation approaches (Salimans & Ho, 2022; Luhman & Luhman, 2021; Song et al., 2023; Meng et al., 2023; Li et al., 2023b; Luo et al., 2023a; Kim et al., 2024) compress long trajectories into few-step solvers, while training-free methods leverage ordinary differential equations (ODEs), e.g. DDIM (Song et al., 2021), DPM-Solver (Popov et al., 2022; Bao et al., 2022; Lu et al., 2025; Zheng et al., 2023). These methods enable fast sampling but remain primarily designed for unconditional synthesis and do not explicitly align the forward dynamics with the degradation process. In contrast, RDIM targets paired inverse problems by explicitly modeling residuals and allowing few-step implicit updates.

**Flow Models** (FMs) (Albergo et al., 2023; Do et al., 2024; Lipman et al., 2023; Liu et al., 2023b) learn deterministic ODE flows between arbitrary distributions using the flow matching objective (Lipman et al., 2023), which is closely related to DM's score matching (Song et al., 2020). They can be understood as zero-variance limits of diffusion bridges, producing transport maps for unpaired or cross-domain translation. However, the deterministic nature of these flows restricts their capacity to capture uncertainty, an important property for restoration tasks involving strong degradations. RDIM differs by maintaining stochastic residual modeling with controllable variance, which empirically improves robustness and generalization.

**Bridge Models** (BMs) can be categorized into Schrödinger bridges (SB) and diffusion bridges (DB). The former constructs a stochastic process connecting two arbitrary marginal distributions (De Bortoli et al., 2021; Chen et al., 2022; Liu et al., 2023a), while the later conditions a stochastic differential equation (SDE) on fixed endpoints (Heng et al., 2025; Li et al., 2023a; Zhou et al., 2024). Both methods have been exploited for image-to-image translation. In particular, Zhou et al. (2024) (DDBM) learn to simulate the time-reversal of a DB based on a learned score matching. Li et al. (2023a) (BBDM) instead focus on constructing a Brownian bridge. SBALIGN (Somnath et al., 2023) and $\Omega$-Bridge (Liu et al., 2023c) use Doob's $h$-transform to guide trajectories toward prescribed terminal states, and GOUB (Yue et al., 2024) incorporates a mean-reverting Ornstein–Uhlenbeck (OU) bridge to improve stability.

Although BM are powerful, SBs operate in unpaired settings, not enforcing or learning correspondences between samples. DBs impose strict boundary conditioning on endpoints, which can bias trajectories toward smoother transitions, blurring high-frequency details (Kieu et al., 2025). In contrast, RDIM is bridge-like in that its forward process defines a stochastic interpolation between $x_0$ and $y_0$, but it differs fundamentally from SB/DB frameworks: RDIM operates in paired settings, relaxes endpoint constraints, and uses an implicit DDIM-style sampler that supports step skipping.

**Stochastic optimal control** (SOC) has recently been adopted to steer diffusion trajectories. DIS (Berner et al., 2023) formalized the connection between SOC and diffusion, while RB-Modulation (Rout et al., 2024) applied SOC principles for training-free style transfer. UniDB (Zhu et al., 2025) integrates SOC with diffusion bridges using penalty terms to guide forward trajectories toward terminal states, improving perceptual quality. RDIM differs by requiring neither fixed endpoints nor SOC penalties, instead relying on a flexible residual-based forward process that remains aligned with practical degradations.

**Alternative methods** to standard diffusion processes have also been explored. Inversion-by-direct-iteration (IDI) (Delbracio & Milanfar, 2023) replaces the stochastic denoising trajectory with a fixed-point iterative scheme, offering competitive restoration without a diffusion process. DiracDiffusion (Fabian et al., 2024) proposes a deterministic, data-consistent update rule that incrementally reconstructs images using Dirac-like propagation rather than probabilistic diffusion. Residual Denoising Diffusion Models (RDDM) (Liu et al., 2024) incorporate residual learning into the diffusion process to accelerate convergence and reduce the dependency on long sampling chains. Iterative $\alpha$-(de)blending (Heitz et al., 2023) introduces a minimalist deterministic diffusion variant based on recursive blending operations, enabling efficient incremental reconstruction. These approaches share with RDIM the motivation of improving reconstruction fidelity and reducing sampling cost, but differ fundamentally in that RDIM preserves the generative diffusion structure while aligning the forward process with the degradation model and enabling DDIM-style long-range sampling.

## 5 CONCLUSION

RDIMs constitute a diffusion framework tailored for inverse problems that explicitly models the residuals between HQ and LQ images. Aligning the forward process with the actual degradation and leveraging implicit sampling enables RDIMs to produce accurate reconstructions with significantly fewer steps than conventional DDPMs. Furthermore, RDIM achieves superior results compared to DDPM, reducing hallucinations while maintaining fidelity, highlighting that starting the reverse process closer to the LQ images offers a more informed and effective initialization. Experiments on denoising and SR demonstrate consistent improvements over DDPMs and performance comparable to or exceeding ResShift, achieving HQ results with single or few step inference. These results establish RDIMs as an efficient and versatile approach for a wide range of image reconstruction tasks.

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

# A DERIVATIONS

This section presents detailed mathematical derivations to support this work. All intermediate steps and calculations omitted for brevity in the main text are included here for completeness and reference.

## A.1 FORWARD PROCESS CUMULATIVE TRANSITION DISTRIBUTION $q(\boldsymbol{x_t}|\boldsymbol{x_0}, \boldsymbol{\Delta})$

The RDIM forward process is designed to align with a forward model that converts the data, $\boldsymbol{x_0}$, into the corresponding corrupted version, $\boldsymbol{y_0}$. To achieve this, the Gaussian transition distribution in Equation (4) is derived for a Markovian version of the RDIM forward process. However, when generating a latent variable $\boldsymbol{x_t}$ starting from $\boldsymbol{x_0}$, the sequential formulation of the diffusion process can become computationally expensive, particularly as the timestep $t$ increases. To address this problem, the reparameterization trick can be leveraged, allowing the cumulative Gaussian transitions of the forward process to be expressed in closed form. As a result, $\boldsymbol{x_t}$ can be computed at an arbitrary timestep $t$ as a function of $\boldsymbol{x_0}$, the fraction of residual between $\boldsymbol{x_0}$ and $\boldsymbol{y_0}$, $\lambda_t \boldsymbol{\Delta}$ (with $\lambda_t$ determining the amount of residual to be removed between each diffusion step), and optional forward variance parameter $\gamma$:

$$
\begin{aligned}
\boldsymbol{x_t} &= \boldsymbol{x_{t-1}} - \lambda_t \boldsymbol{\Delta} + \sqrt{\gamma^2 \lambda_t} \boldsymbol{\epsilon_t} \\
&= \boldsymbol{x_{t-2}} - \lambda_{t-1} \boldsymbol{\Delta} + \sqrt{\gamma^2 \lambda_{t-1}} \boldsymbol{\epsilon_{t-1}} - \lambda_t \boldsymbol{\Delta} + \sqrt{\gamma^2 \lambda_t} \boldsymbol{\epsilon_t} \\
&= \cdots \\
&= \boldsymbol{x_0} - \boldsymbol{\Delta} \underbrace{(\lambda_1 + \lambda_2 + \cdots + \lambda_t)}_{\bar{\lambda}_t} + \sqrt{\gamma^2 \lambda_1} \boldsymbol{\epsilon_1} + \sqrt{\gamma^2 \lambda_2} \boldsymbol{\epsilon_2} + \cdots + \sqrt{\gamma^2 \lambda_t} \boldsymbol{\epsilon_t}
\end{aligned}
\tag{19}
$$

where $\boldsymbol{\epsilon_1}, \boldsymbol{\epsilon_2}, \ldots, \boldsymbol{\epsilon_t} \sim \mathcal{N}(0, \boldsymbol{I})$. Hence:

$$
\begin{aligned}
\boldsymbol{x_t} &\sim \mathcal{N}\left(\boldsymbol{x_0} - \bar{\lambda}_t \boldsymbol{\Delta}, \gamma^2 (\lambda_1 + \lambda_2 + \cdots + \lambda_t) \boldsymbol{I}\right) \\
&\sim \mathcal{N}\left(\boldsymbol{x_0} - \bar{\lambda}_t \boldsymbol{\Delta}, \gamma^2 \bar{\lambda}_t \boldsymbol{I}\right),
\end{aligned}
\tag{20}
$$

Therefore, the cumulative Gaussian transition in the forward process can be defined as in Equation (5) and, when $\gamma = 0$, it collapses into a Dirac delta function.

**Cumulative sum of weights** $\lambda_t$  Each weight $\lambda_t$, used to control the variance and amount of residual to be removed in each diffusion step, is computed as $\lambda_t = \beta_t - \beta_{t-1}$, with $\beta_t$ representing the transition at forward step $t$ between original and corrupted data in the Markov chain. Consequently, the cumulative sum of weights $\lambda_t$ from the initial timestep $t = 1$ up to timestep $t = \tau$ is given as follows:

$$
\bar{\lambda}_\tau = \sum_{t=1}^{\tau} \lambda_t = \sum_{t=1}^{\tau} (\beta_t - \beta_{t-1}) = \beta_\tau - \beta_0
\tag{21}
$$

**Distribution of the last latent variable** $q(\boldsymbol{x_T}|\boldsymbol{x_0}, \boldsymbol{\Delta}) = q(\boldsymbol{x_T}|\boldsymbol{y_0})$. Given that the RDIM forward process is designed to align with a forward model that converts the data, $\boldsymbol{x_0}$, into the corresponding corrupted version, $\boldsymbol{y_0}$, the residual, $\boldsymbol{\Delta}$, should be fully removed from $\boldsymbol{x_0}$ at the end of the forward process, i.e., after exactly $T$ timesteps. This ensures that the last latent variable, $\boldsymbol{x_T}$, will coincide exactly with the corrupted data, $\boldsymbol{y_0}$, when the forward process is deterministic, and will converge to a noisy sample centered at $\boldsymbol{y_0}$ when the forward process is stochastic. Hence, considering Equation (20), the last latent variable, $\boldsymbol{x_T}$, of the forward process can be sampled as:

$$
\begin{aligned}
\boldsymbol{x_T} &\sim \mathcal{N}\left(\boldsymbol{x_0} - \bar{\lambda}_T \boldsymbol{\Delta}, \gamma^2 \bar{\lambda}_T \boldsymbol{I}\right) \\
&\sim \mathcal{N}\left(\boldsymbol{x_0} - \bar{\lambda}_T\left(\boldsymbol{x_0} - \boldsymbol{y_0}\right), \gamma^2 \bar{\lambda}_T \boldsymbol{I}\right) \\
&\sim \mathcal{N}\left(\boldsymbol{x_0}\left(1 - \bar{\lambda}_T\right) + \bar{\lambda}_T \boldsymbol{y_0}, \gamma^2 \bar{\lambda}_T \boldsymbol{I}\right).
\end{aligned}
\tag{22}
$$

Logically, to ensure the aforementioned condition of centering the distribution $q(\boldsymbol{x_T}|\boldsymbol{x_0}, \boldsymbol{\Delta})$ on the corrupted data, $\boldsymbol{y_0}$, the cumulative sum of weights $\lambda_t$ over the $T$ timesteps must satisfy $\bar{\lambda}_T = 1$. This imposes that $\beta_0 = 0$ and $\beta_T = 1$, since $\bar{\lambda}_T = \beta_T - \beta_0$, as mentioned above. Accordingly:

$$
\boldsymbol{x_T} \sim \mathcal{N}\left(\boldsymbol{y_0}, \gamma^2 \boldsymbol{I}\right).
\tag{23}
$$

This formulation assures that the residual $\boldsymbol{\Delta}$ is fully removed after exactly $T$ timesteps ($\bar{\lambda}_t = 1$ only when $t = T$) and that the distribution $q(\boldsymbol{x_T}|\boldsymbol{x_0}, \boldsymbol{\Delta})$ is centered at the corrupted data, $\boldsymbol{y_0}$. As a result of this deliberate design choice, $q(\boldsymbol{x_T}|\boldsymbol{x_0}, \boldsymbol{\Delta}) = q(\boldsymbol{x_T}|\boldsymbol{y_0})$ holds exactly at $t = T$. In addition, when $\gamma = 0$, the Gaussian collapses into a Dirac delta function centered at $\boldsymbol{y_0}$, thereby the final latent variable, $\boldsymbol{x_T}$, coincides exactly with the corrupted data, i.e., $\boldsymbol{x_T} = \boldsymbol{y_0}$.

Additionally, the $\beta$-schedule defined in Equation (16) is designed to impose $\beta_0 = 0$ and $\beta_T = 1$, thus satisfying the aforementioned requirements. In particular, the cumulative sum of weights $\lambda_t$ is $\bar{\lambda}_t = \beta_t$ when $\beta_0 = 0$ (see Equation (21)). Figure 2 showcases the progression of the weights $\beta_t$ and $\lambda_t$ across timesteps. If $p = 1.0$, the $\beta$-schedule is linear and $\lambda_t$ is constant, resulting in uniform fractions of $\Delta$ removed along the forward process.

Accordingly, under this condition of $\beta_0 = 0$, the cumulative forward transition distribution, $q(\boldsymbol{x_t}|\boldsymbol{x_0}, \boldsymbol{\Delta})$, expressed in Equation (20) can be further simplified to:

$$
\boldsymbol{x_t} \sim \mathcal{N}\left(\boldsymbol{x_0} - \beta_t \boldsymbol{\Delta}, \gamma^2 \beta_t \boldsymbol{I}\right).
\tag{24}
$$

## A.2 Reverse process transition distribution $q(\boldsymbol{x_{t-1}}|\boldsymbol{x_t}, \boldsymbol{x_0}, \boldsymbol{\Delta})$

The reverse process involves computing the reverse transition, which is defined as the Gaussian distribution in Equation (7) and is designed to preserve the marginal $q(\boldsymbol{x_t}|\boldsymbol{x_0}, \boldsymbol{\Delta})$ in Equation (5). Considering that Gaussian distributions exhibit the property that their conditional means are linear combinations of the conditioning variables (see Lemma B.2), then the mean $\tilde{\boldsymbol{\mu}}_t$ of $q(\boldsymbol{x_{t-1}}|\boldsymbol{x_t}, \boldsymbol{x_0}, \boldsymbol{\Delta})$ can be expressed as a linear interpolation between $\boldsymbol{x_t}$, $\boldsymbol{x_0}$, and $\boldsymbol{\Delta}$. Particularly, to match the form of the forward process cumulative transition, $q(\boldsymbol{x_t}|\boldsymbol{x_0}, \boldsymbol{\Delta})$, the mean $\tilde{\boldsymbol{\mu}}_t$ is assumed to be a linear combination between $(\boldsymbol{x_0} - \beta_t \boldsymbol{\Delta})$ and $\boldsymbol{x_t}$:

$$
\tilde{\boldsymbol{\mu}}_t = a\left(\boldsymbol{x_0} - \beta_t \boldsymbol{\Delta}\right) + b\boldsymbol{x_t},
\tag{25}
$$

where $a$ and $b$ are constants.

Following, given $q(\boldsymbol{x_t}|\boldsymbol{x_0}, \boldsymbol{\Delta})$ and the formulation assumed for $q(\boldsymbol{x_{t-1}}|\boldsymbol{x_t}, \boldsymbol{x_0}, \boldsymbol{\Delta})$, then $q(\boldsymbol{x_{t-1}}|\boldsymbol{x_0}, \boldsymbol{\Delta})$ can be defined by leveraging a property of marginal and conditional Gaussians (see Lemma B.1):

$$
\begin{aligned}
q(\boldsymbol{x_{t-1}}|\boldsymbol{x_0}, \boldsymbol{\Delta}) &= \mathcal{N}\left(\boldsymbol{x_{t-1}} \big| b\left(\boldsymbol{x_0} - \beta_t \boldsymbol{\Delta}\right) + a\left(\boldsymbol{x_0} - \beta_t \boldsymbol{\Delta}\right), \tilde{\sigma}_t^2 \boldsymbol{I} + b\gamma^2 \beta_t \boldsymbol{I} b\right) \\
&= \mathcal{N}\left(\boldsymbol{x_{t-1}} \big| \left(\boldsymbol{x_0} - \beta_t \boldsymbol{\Delta}\right)(a + b), \left(\tilde{\sigma}_t^2 + \gamma^2 \beta_t b^2\right) \boldsymbol{I}\right).
\end{aligned}
\tag{26}
$$

Recalling that $q(\boldsymbol{x_t}|\boldsymbol{x_0}, \boldsymbol{\Delta}) = \mathcal{N}\left(\boldsymbol{x_t}|\boldsymbol{x_0} - \beta_t \boldsymbol{\Delta}, \gamma^2 \beta_t \boldsymbol{I}\right)$ is being enforced, the cumulative Gaussian transition to obtain $\boldsymbol{x_{t-1}}$ given $\boldsymbol{x_0}$ and $\boldsymbol{\Delta}$ is also defined as:

$$
q(\boldsymbol{x_{t-1}}|\boldsymbol{x_0}, \boldsymbol{\Delta}) = \mathcal{N}\left(\boldsymbol{x_{t-1}}|\boldsymbol{x_0} - \beta_{t-1} \boldsymbol{\Delta}, \gamma^2 \beta_{t-1} \boldsymbol{I}\right).
\tag{27}
$$

Accordingly, to ensure that the designed reverse transition preserves the marginal $q(\boldsymbol{x_t}|\boldsymbol{x_0}, \boldsymbol{\Delta})$, as guaranteed by Lemma B.1, the following equality must be satisfied:

$$\mathcal{N}\left(\boldsymbol{x_{t-1}}\big|\left(\boldsymbol{x_0} - \beta_t\boldsymbol{\Delta}\right)(a+b), \left(\tilde{\sigma}_t^2 + \gamma^2\beta_t b^2\right)\boldsymbol{I}\right) = \mathcal{N}\left(\boldsymbol{x_{t-1}}\big|\boldsymbol{x_0} - \beta_{t-1}\boldsymbol{\Delta}, \gamma^2\beta_{t-1}\boldsymbol{I}\right), \quad (28)$$

and thus $a$ and $b$ can be computed by solving the following system of equations:

$$\begin{cases} \left(\boldsymbol{x_0} - \beta_t\boldsymbol{\Delta}\right)(a+b) = \boldsymbol{x_0} - \beta_{t-1}\boldsymbol{\Delta} \\ \tilde{\sigma}_t^2 + \gamma^2\beta_t b^2 = \gamma^2\beta_{t-1} \end{cases} \Leftrightarrow \begin{cases} a = 1 + \frac{\lambda_t\boldsymbol{\Delta}}{\boldsymbol{x_0} - \beta_t\boldsymbol{\Delta}} - \sqrt{\frac{\gamma^2\beta_{t-1} - \tilde{\sigma}_t^2}{\gamma^2\beta_t}} \\ b = \sqrt{\frac{\gamma^2\beta_{t-1} - \tilde{\sigma}_t^2}{\gamma^2\beta_t}} \end{cases}. \quad (29)$$

Consequently, the mean of each reverse transition, $\tilde{\boldsymbol{\mu}}_t$, is given as:

$$\begin{aligned} \tilde{\boldsymbol{\mu}}_t &= a\left(\boldsymbol{x_0} - \beta_t\boldsymbol{\Delta}\right) + b\boldsymbol{x_t} \\ &= \left(1 + \frac{\lambda_t\boldsymbol{\Delta}}{\boldsymbol{x_0} - \beta_t\boldsymbol{\Delta}} - \sqrt{\frac{\gamma^2\beta_{t-1} - \tilde{\sigma}_t^2}{\gamma^2\beta_t}}\right)\left(\boldsymbol{x_0} - \beta_t\boldsymbol{\Delta}\right) + \sqrt{\frac{\gamma^2\beta_{t-1} - \tilde{\sigma}_t^2}{\gamma^2\beta_t}}\boldsymbol{x_t} \\ &= \boldsymbol{x_0} - \beta_t\boldsymbol{\Delta} + \lambda_t\boldsymbol{\Delta} - \sqrt{\frac{\gamma^2\beta_{t-1} - \tilde{\sigma}_t^2}{\gamma^2\beta_t}}\left(\boldsymbol{x_0} - \beta_t\boldsymbol{\Delta}\right) + \sqrt{\frac{\gamma^2\beta_{t-1} - \tilde{\sigma}_t^2}{\gamma^2\beta_t}}\boldsymbol{x_t} \\ &= \boldsymbol{x_0} - \beta_t\boldsymbol{\Delta} + \left(\beta_t - \beta_{t-1}\right)\boldsymbol{\Delta} + \sqrt{\frac{\gamma^2\beta_{t-1} - \tilde{\sigma}_t^2}{\gamma^2\beta_t}}\left(\boldsymbol{x_t} - \boldsymbol{x_0} + \beta_t\boldsymbol{\Delta}\right) \\ &= \boldsymbol{x_0} - \beta_{t-1}\boldsymbol{\Delta} + \sqrt{\gamma^2\beta_{t-1} - \tilde{\sigma}_t^2}\left(\frac{\boldsymbol{x_t} - \boldsymbol{x_0} + \beta_t\boldsymbol{\Delta}}{\sqrt{\gamma^2\beta_t}}\right), \end{aligned} \quad (30)$$

where, in particular, singularities can occur for $\gamma = 0$. Therefore, for $\gamma \neq 0$, the mean of the reverse process transition distribution that preserves the marginal $q(\boldsymbol{x_t}|\boldsymbol{x_0}, \boldsymbol{\Delta})$ is given as:

$$\tilde{\boldsymbol{\mu}}_{t|\gamma\neq 0} = \boldsymbol{x_0} - \beta_{t-1}\boldsymbol{\Delta} + \sqrt{\gamma^2\beta_{t-1} - \tilde{\sigma}_t^2}\left(\frac{\boldsymbol{x_t} - \boldsymbol{x_0} + \beta_t\boldsymbol{\Delta}}{\sqrt{\gamma^2\beta_t}}\right). \quad (31)$$

Essentially, the mean, $\tilde{\boldsymbol{\mu}}_t$, is chosen to ensure that $q(\boldsymbol{x_t}|\boldsymbol{x_0}, \boldsymbol{\Delta}) = \mathcal{N}\left(\boldsymbol{x_t}|\boldsymbol{x_0} - \beta_t\boldsymbol{\Delta}, \gamma^2\beta_t\boldsymbol{I}\right)$ is satisfied for all $t \in \{1, 2, \ldots, T\}$. Meanwhile, the variance $\tilde{\sigma}_t^2$ is set equal to the variance of the ResShift reverse transition (see Appendix A.3), thus $\tilde{\sigma}_t^2 = \gamma^2\frac{\beta_{t-1}}{\beta_t}\lambda_t = \tilde{\lambda}_t$.

**Relationship between $\boldsymbol{x_t}$, $\boldsymbol{x_0}$, $\boldsymbol{\Delta}$, and $\boldsymbol{\epsilon}$.** Considering the marginal $q(\boldsymbol{x_t}|\boldsymbol{x_0}, \boldsymbol{\Delta})$ and $\gamma \neq 0$, a relationship between $\boldsymbol{x_t}$, $\boldsymbol{x_0}$, $\boldsymbol{\Delta}$, and $\boldsymbol{\epsilon} \sim \mathcal{N}\left(0, \boldsymbol{I}\right)$ can be derived from the reparameterization trick:

$$\begin{aligned} q(\boldsymbol{x_t}|\boldsymbol{x_0}, \boldsymbol{\Delta}) &= \mathcal{N}\left(\boldsymbol{x_t}|\boldsymbol{x_0} - \beta_t\boldsymbol{\Delta}, \gamma^2\beta_t\boldsymbol{I}\right) \\ &\Rightarrow \boldsymbol{x_t} = \boldsymbol{x_0} - \beta_t\boldsymbol{\Delta} + \sqrt{\gamma^2\beta_t}\boldsymbol{\epsilon} \\ &\Leftrightarrow \boldsymbol{\epsilon} = \frac{\boldsymbol{x_t} - \boldsymbol{x_0} + \beta_t\boldsymbol{\Delta}}{\sqrt{\gamma^2\beta_t}}, \end{aligned} \quad (32)$$

This expression exactly matches the term between parentheses in the mean of the reverse process transition distribution for $\gamma \neq 0$, in Equation (31). Accordingly, the mean can be rewritten as:

$$\tilde{\boldsymbol{\mu}}_{t|\gamma\neq 0} = \boldsymbol{x_0} - \beta_{t-1}\boldsymbol{\Delta} + \sqrt{\gamma^2\beta_{t-1} - \tilde{\sigma}_t^2}\boldsymbol{\epsilon}, \quad (33)$$

which structurally matches the reparameterized form of the marginal $q(\boldsymbol{x_{t-1}}|\boldsymbol{x_0}, \boldsymbol{\Delta})$, exhibiting the same functional form and differing only in the variance term. This highlights that, when $\gamma \neq 0$, the reverse transition is aligned with cumulative transitions and can be leveraged to efficiently sample any state at an arbitrary timestep.

**Reverse transition with $\gamma = 0$.** Particularly, for $\gamma = 0$, the forward process cumulative transition, defined as a Gaussian distribution, degenerates into a Dirac delta function (see also Appendix A.1). Consequently, for $\gamma = 0$, Lemma B.1 is not applicable. In fact, in this case, the forward process effectively becomes a linear interpolation between $x_0$ and $y_0$. Logically, when $\gamma = 0$, it follows that the reverse process simply needs to invert this deterministic process. However, the continuity of the mean, $\tilde{\mu}_t$, should be assured at $\gamma = 0$, i.e., $\tilde{\mu}_{t|\gamma=0} = \lim_{\gamma \to 0} \tilde{\mu}_{t|\gamma \neq 0}$.

Considering Equation (24) in Appendix A.1, it follows $\lim_{\gamma \to 0} x_t = x_0 - \beta_t \Delta$, which implies that $x_t - x_0 + \beta_t \Delta \to 0$ as $\gamma \to 0$. Accordingly, given $\tilde{\sigma}_t^2 = \gamma^2 \frac{\beta_{t-1}}{\beta_t} \lambda_t$, then $\lim_{\gamma \to 0} \tilde{\mu}_{t|\gamma \neq 0} = x_0 - \beta_{t-1} \Delta$. As a result, to ensure the continuity of the mean $\tilde{\mu}_t$ at $\gamma = 0$, the Gaussian transition $q(x_{t-1}|x_t, x_0, \Delta)$ is assumed to collapse into a Dirac delta function centered at $x_0 - \beta_{t-1} \Delta$. Hence, for $\gamma = 0$, the mean of the reverse process transition distribution is defined as:

$$\tilde{\mu}_{t|\gamma=0} = x_0 - \beta_{t-1} \Delta. \tag{34}$$

Notably, this formulation of $\tilde{\mu}_{t|\gamma=0}$ matches the mean of the cumulative forward transition, $q(x_{t-1}|x_0, \Delta)$ (see Appendix A.1), showing that the reverse process, when $\gamma = 0$, reduces to a linear interpolation between $y_0$ and $x_0$ (inverse of the deterministic forward process). Additionally, it aligns with the concept of cumulative transitions, which is paramount for long-range transitions (see Section 2.5). In essence, the mean, $\tilde{\mu}_t$, is expressed as in Equation (8) and is continuous at $\gamma = 0$. Nonetheless, the $\gamma$ constant hyperparameter is immutable in practice, i.e., set only once for each model instance, thereby no discontinuity issues would ever arise due to $\gamma$ (see Appendix A.4).

## A.3 REVERSE TRANSITION WITH $\tilde{\sigma}_t^2 = \gamma^2 \frac{\beta_{t-1}}{\beta_t} \lambda_t$ (RESSHIFT VARIANCE, $\tilde{\lambda}_t$)

In particular, if the reverse process transition variance, $\tilde{\sigma}_t^2$, is set to be the same as in ResShift, $\tilde{\lambda}_t = \gamma^2 \frac{\beta_{t-1}}{\beta_t} \lambda_t$, then the mean, $\tilde{\mu}_{t|\gamma \neq 0}$, reduces to:

$$
\begin{aligned}
\tilde{\mu}_{t|\gamma \neq 0} &= x_0 - \beta_{t-1} \Delta + \sqrt{\gamma^2 \beta_{t-1} - \tilde{\sigma}_t^2} \left( \frac{x_t - x_0 + \beta_t \Delta}{\sqrt{\gamma^2 \beta_t}} \right) \\
&= x_0 - \beta_{t-1} \Delta + \sqrt{\gamma^2 \beta_{t-1} - \gamma^2 \frac{\beta_{t-1}}{\beta_t} \lambda_t} \left( \frac{x_t - x_0 + \beta_t \Delta}{\sqrt{\gamma^2 \beta_t}} \right) \\
&= x_0 - \beta_{t-1} \Delta + \sqrt{\frac{\gamma^4 \beta_t \beta_{t-1} - \gamma^4 \beta_{t-1} (\beta_t - \beta_{t-1})}{\gamma^2 \beta_t}} \left( \frac{x_t - x_0 + \beta_t \Delta}{\sqrt{\gamma^2 \beta_t}} \right) \\
&= x_0 - \beta_{t-1} \Delta + \frac{\sqrt{\gamma^4 \beta_{t-1}^2} (x_t - x_0 + \beta_t \Delta)}{\gamma^2 \beta_t} \\
&= x_0 - \beta_{t-1} \Delta + \frac{\beta_{t-1} x_t - \beta_{t-1} x_0 + \beta_t \beta_{t-1} \Delta}{\beta_t} \\
&= \frac{\beta_{t-1}}{\beta_t} x_t + x_0 \left( 1 - \frac{\beta_{t-1}}{\beta_t} \right) \\
&= \frac{\beta_{t-1}}{\beta_t} x_t + \frac{\lambda_t}{\beta_t} x_0,
\end{aligned} \tag{35}
$$

and thus the distribution $q(x_{t-1}|x_t, x_0, \Delta)_{\gamma \neq 0}$ becomes:

$$q(x_{t-1}|x_t, x_0, \Delta)_{\gamma \neq 0} = \mathcal{N} \left( x_{t-1} \bigg| \frac{\beta_{t-1}}{\beta_t} x_t + \frac{\lambda_t}{\beta_t} x_0, \tilde{\lambda}_t I \right), \tag{36}$$

which is exactly the ResShift reverse transition distribution. In essence, if the RDIM reverse transition variance, $\tilde{\sigma}_t^2$, is set to be the same as in ResShift, then $\tilde{\mu}_t$ will match the mean of the ResShift reverse transition. Accordingly, RDIM reduces to ResShift for this specific variance, revealing that ResShift is a particular case of RDIM.

Alternatively, for $\gamma \neq 0$, if the variance is set to $\tilde{\sigma}_t^2 = 0$, then there are no stochastic terms involved when traversing the reverse trajectory, as $q(x_{t-1}|x_t, x_0, \Delta)_{\gamma \neq 0}$ degenerates into a $\delta$-distribution

and avoids sampling random noise (given Equations (7) and (31)). Consequently, the reverse process becomes deterministic. Therefore, a constant hyperparameter, $\eta \in [0, 1]$, can be introduced to interpolate between a deterministic and stochastic reverse process when $\gamma \neq 0$, thus allowing control over the variability in the reverse trajectory (see Equation (9)). Specifically, when $\eta = 0$, the Gaussian collapses into a Dirac delta function.

**Absence of non-real square roots.** From Equation (9), it follows that to avoid a non-real square root, when $\gamma \neq 0$, the condition $\gamma^2 \beta_{t-1} \geq \eta^2 \tilde{\lambda}_t$ must be satisfied. Considering $\tilde{\lambda}_t = \gamma^2 \frac{\beta_{t-1}}{\beta_t} \lambda_t$, then:

$$\gamma^2 \beta_{t-1} \geq \eta^2 \tilde{\lambda}_t \Leftrightarrow 1 \geq \eta^2 \frac{\lambda_t}{\beta_t} \Leftrightarrow 1 \geq \eta^2 \left(1 - \frac{\beta_{t-1}}{\beta_t}\right). \tag{37}$$

Recalling that $\eta \in [0, 1]$, then $0 \leq \eta^2 \leq 1$. Moreover, since $0 \leq \beta_{t-1} \leq \beta_t$, it follows that $0 \leq \frac{\beta_{t-1}}{\beta_t} \leq 1$, which in turn ensures that the term inside parentheses meets the condition $1 - \frac{\beta_{t-1}}{\beta_t} \leq 1$. Consequently, the product of these two terms is always less than or equal to 1, and thus the Inequality (37) is satisfied for all $\eta \in [0, 1]$ and $t \in \{1, 2, \ldots, T\}$.

### A.4 TRAINING OBJECTIVE

During inference, $\boldsymbol{x_0}$ and $\boldsymbol{\Delta}$ are unknown, thus sampling from the true reverse transition distribution, $q(\boldsymbol{x_{t-1}}|\boldsymbol{x_t}, \boldsymbol{x_0}, \boldsymbol{\Delta})$, is not possible. Therefore, a learnable parametric model, $p_\theta(\boldsymbol{x_{t-1}}|\boldsymbol{x_t}, \boldsymbol{y_0})$, defined as a Gaussian distribution, is introduced to approximate $q(\boldsymbol{x_{t-1}}|\boldsymbol{x_t}, \boldsymbol{x_0}, \boldsymbol{\Delta})$. Particularly, an accurate estimation is required to ensure precise reconstruction of the data, $\boldsymbol{x_0}$, at inference. This approximation is achieved by minimizing the KL divergence between both distributions, while accounting for all timesteps:

$$\theta^* = \arg\min_\theta D_{\text{KL}}(q(\boldsymbol{x_{1:T}}|\boldsymbol{x_0}, \boldsymbol{\Delta}) \| p_\theta(\boldsymbol{x_{1:T}}|\boldsymbol{y_0})), \tag{38}$$

where $\theta^*$ denotes the optimal parameters. In fact, this objective of minimizing the KL divergence in Equation (38) is equivalent to minimizing the negative variational lower bound (VLB) on the conditional log-likelihood. This is the RDIM objective function and it can expanded further:

$$\begin{aligned}
\mathcal{L}(\theta) &= \mathbb{E}_{q(\boldsymbol{x_{1:T}}|\boldsymbol{x_0}, \boldsymbol{\Delta})}\left[\log\left(\frac{q(\boldsymbol{x_{1:T}}|\boldsymbol{x_0}, \boldsymbol{\Delta})}{p_\theta(\boldsymbol{x_{0:T}}|\boldsymbol{y_0})}\right)\right] \\
&= \mathbb{E}_{q(\boldsymbol{x_{1:T}}|\boldsymbol{x_0}, \boldsymbol{\Delta})}\left[\log\left(\frac{q(\boldsymbol{x_T}|\boldsymbol{x_0}, \boldsymbol{\Delta})\prod_{t=2}^T q(\boldsymbol{x_{t-1}}|\boldsymbol{x_t}, \boldsymbol{x_0}, \boldsymbol{\Delta})}{p(\boldsymbol{x_T}|\boldsymbol{y_0})\prod_{t=1}^T p_\theta(\boldsymbol{x_{t-1}}|\boldsymbol{x_t}, \boldsymbol{y_0})}\right)\right] \\
&= \mathbb{E}_{q(\boldsymbol{x_{1:T}}|\boldsymbol{x_0}, \boldsymbol{\Delta})}\left[\log\left(\frac{q(\boldsymbol{x_T}|\boldsymbol{x_0}, \boldsymbol{\Delta})}{p(\boldsymbol{x_T}|\boldsymbol{y_0})}\right)\right. \\
&\qquad \left. + \log\left(\prod_{t=2}^T \frac{q(\boldsymbol{x_{t-1}}|\boldsymbol{x_t}, \boldsymbol{x_0}, \boldsymbol{\Delta})}{p_\theta(\boldsymbol{x_{t-1}}|\boldsymbol{x_t}, \boldsymbol{y_0})}\right) - \log\left(p_\theta(\boldsymbol{x_0}|\boldsymbol{x_1}, \boldsymbol{y_0})\right)\right] \\
&= \mathbb{E}_{q(\boldsymbol{x_T}|\boldsymbol{x_0}, \boldsymbol{\Delta})}\left[\log\left(\frac{q(\boldsymbol{x_T}|\boldsymbol{x_0}, \boldsymbol{\Delta})}{p(\boldsymbol{x_T}|\boldsymbol{y_0})}\right)\right] \\
&\qquad + \mathbb{E}_{q(\boldsymbol{x_{1:T}}|\boldsymbol{x_0}, \boldsymbol{\Delta})}\left[\sum_{t=2}^T \log\left(\frac{q(\boldsymbol{x_{t-1}}|\boldsymbol{x_t}, \boldsymbol{x_0}, \boldsymbol{\Delta})}{p_\theta(\boldsymbol{x_{t-1}}|\boldsymbol{x_t}, \boldsymbol{y_0})}\right)\right] \\
&\qquad - \mathbb{E}_{q(\boldsymbol{x_1}|\boldsymbol{x_0}, \boldsymbol{\Delta})}\left[\log\left(p_\theta(\boldsymbol{x_0}|\boldsymbol{x_1}, \boldsymbol{y_0})\right)\right] \\
&= D_{\text{KL}}(q(\boldsymbol{x_T}|\boldsymbol{x_0}, \boldsymbol{\Delta}) \| p(\boldsymbol{x_T}|\boldsymbol{y_0})) \\
&\qquad + \sum_{t=2}^T \mathbb{E}_{q(\boldsymbol{x_{t-1}}, \boldsymbol{x_t}|\boldsymbol{x_0}, \boldsymbol{\Delta})}\left[\log\left(\frac{q(\boldsymbol{x_{t-1}}|\boldsymbol{x_t}, \boldsymbol{x_0}, \boldsymbol{\Delta})}{p_\theta(\boldsymbol{x_{t-1}}|\boldsymbol{x_t}, \boldsymbol{y_0})}\right)\right] \\
&\qquad - \mathbb{E}_{q(\boldsymbol{x_1}|\boldsymbol{x_0}, \boldsymbol{\Delta})}\left[\log\left(p_\theta(\boldsymbol{x_0}|\boldsymbol{x_1}, \boldsymbol{y_0})\right)\right] \\
&= D_{\text{KL}}(q(\boldsymbol{x_T}|\boldsymbol{x_0}, \boldsymbol{\Delta}) \| p(\boldsymbol{x_T}|\boldsymbol{y_0})) \\
&\qquad + \sum_{t=2}^T \mathbb{E}_{q(\boldsymbol{x_t}|\boldsymbol{x_0}, \boldsymbol{\Delta})}\left[\mathbb{E}_{q(\boldsymbol{x_{t-1}}|\boldsymbol{x_t}, \boldsymbol{x_0}, \boldsymbol{\Delta})}\left[\log\left(\frac{q(\boldsymbol{x_{t-1}}|\boldsymbol{x_t}, \boldsymbol{x_0}, \boldsymbol{\Delta})}{p_\theta(\boldsymbol{x_{t-1}}|\boldsymbol{x_t}, \boldsymbol{y_0})}\right)\right]\right]
\end{aligned} \tag{39}$$

$$-\mathbb{E}_{q(\boldsymbol{x_1}|\boldsymbol{x_0},\boldsymbol{\Delta})}\left[\log\left(p_\theta(\boldsymbol{x_0}|\boldsymbol{x_1},\boldsymbol{y_0})\right)\right]$$

$$=\underbrace{D_{\mathrm{KL}}(q(\boldsymbol{x_T}|\boldsymbol{x_0},\boldsymbol{\Delta})\|p(\boldsymbol{x_T}|\boldsymbol{y_0}))}_{\mathcal{L}_T}$$

$$+\sum_{t=2}^{T}\underbrace{\mathbb{E}_{q(\boldsymbol{x_t}|\boldsymbol{x_0},\boldsymbol{\Delta})}\left[D_{\mathrm{KL}}(q(\boldsymbol{x_{t-1}}|\boldsymbol{x_t},\boldsymbol{x_0},\boldsymbol{\Delta})\|p_\theta(\boldsymbol{x_{t-1}}|\boldsymbol{x_t},\boldsymbol{y_0}))\right]}_{\mathcal{L}_{t-1}}$$

$$\underbrace{-\mathbb{E}_{q(\boldsymbol{x_1}|\boldsymbol{x_0},\boldsymbol{\Delta})}\left[\log\left(p_\theta(\boldsymbol{x_0}|\boldsymbol{x_1},\boldsymbol{y_0})\right)\right]}_{\mathcal{L}_0}=\mathcal{L}_T+\mathcal{L}_{1:T-1}+\mathcal{L}_0.$$

Hence, analogous to DDPMs, the RDIM objective function, $\mathcal{L}(\theta)$, decomposes into $\mathcal{L}_T$ (prior matching term), $\mathcal{L}_{1:T-1}$ (consistency terms), and $\mathcal{L}_0$ (reconstruction term).

**Prior matching term $\mathcal{L}_T$.**   The term $\mathcal{L}_T$ is minimized when the prior, $p(\boldsymbol{x_T}|\boldsymbol{y_0})$, matches the true distribution of the last latent variable, $q(\boldsymbol{x_T}|\boldsymbol{x_0},\boldsymbol{\Delta})=q(\boldsymbol{x_T}|\boldsymbol{y_0})=\mathcal{N}(\boldsymbol{y_0},\gamma^2 I)$. Accordingly, $p(\boldsymbol{x_T}|\boldsymbol{y_0})$ is fixed to such a Gaussian distribution, which is parameterized by constants and involves no learnable parameters. Therefore, $\mathcal{L}_T$ is constant with respect to the model parameters, $\theta$, and is minimized, i.e., $\mathcal{L}_T=0$. Consequently, this term can be excluded from the optimization objective, unlike the terms $\mathcal{L}_{0:T-1}$, which explicitly depend on $\theta$ through the parameterized distribution $p_\theta$.

**Consistency terms $\mathcal{L}_{1:T-1}$.**   The terms $\mathcal{L}_{1:T-1}$ enforce that the learnable parametric model, $p_\theta(\boldsymbol{x_{t-1}}|\boldsymbol{x_t},\boldsymbol{y_0})$, accurately approximates the true reverse transition, $q(\boldsymbol{x_{t-1}}|\boldsymbol{x_t},\boldsymbol{x_0},\boldsymbol{\Delta})$. This fundamentally ensures that the model learns to refine the data at intermediate timesteps, leading to consistency in the reconstruction.

The true reverse transition distribution is known in closed form (see Section 2.4 along with Appendices A.2 and A.3), having mean and variance parameterized as:

$$\tilde{\boldsymbol{\mu}}_{\boldsymbol{t}}=\begin{cases}\boldsymbol{x_0}-\beta_{t-1}\boldsymbol{\Delta}, & \text{if }\gamma=0,\\ \boldsymbol{x_0}-\beta_{t-1}\boldsymbol{\Delta}+\sqrt{\gamma^2\beta_{t-1}-\eta^2\tilde{\lambda}_t}\left(\dfrac{\boldsymbol{x_t}-\boldsymbol{x_0}+\beta_t\boldsymbol{\Delta}}{\sqrt{\gamma^2\beta_t}}\right), & \text{if }\gamma\neq 0,\end{cases} \tag{40}$$

and

$$\tilde{\sigma}_t^2=\begin{cases}\tilde{\lambda}_t, & \text{if }\gamma=0,\\ \eta^2\tilde{\lambda}_t, & \text{if }\gamma\neq 0,\end{cases} \tag{41}$$

where $\tilde{\lambda}_t=\gamma^2\frac{\beta_{t-1}}{\beta_t}\lambda_t$.

Given that $p_\theta(\boldsymbol{x_{t-1}}|\boldsymbol{x_t},\boldsymbol{y_0})$ is defined as a Gaussian distribution with mean $\boldsymbol{\mu_\theta}\left(\boldsymbol{x_t},\boldsymbol{y_0},t\right)$ and variance $\sigma_\theta^2\left(\boldsymbol{x_t},\boldsymbol{y_0},t\right)$, to minimize the KL divergence of each term $\mathcal{L}_{1:T-1}$, the mean and variance of the parametric model should approximate $\tilde{\boldsymbol{\mu}}_{\boldsymbol{t}}$ and $\tilde{\sigma}_t^2$, respectively. Particularly, the variance of $q(\boldsymbol{x_{t-1}}|\boldsymbol{x_t},\boldsymbol{x_0},\boldsymbol{\Delta})$ does not have learnable parameters because it is defined in terms of constant hyperparameters, which are known. Therefore, $\sigma_\theta^2\left(\boldsymbol{x_t},\boldsymbol{y_0},t\right)$ can be fixed to equal exactly $\tilde{\sigma}_t^2$, as expressed in Equation (11). Following, each term $\mathcal{L}_{1:T-1}$ is computed by applying the closed-form expression for the KL divergence between two $d$-dimensional multivariate Gaussian distributions, yielding:

$$D_{\mathrm{KL}}(q(\boldsymbol{x_{t-1}}|\boldsymbol{x_t},\boldsymbol{x_0},\boldsymbol{\Delta})\|p_\theta(\boldsymbol{x_{t-1}}|\boldsymbol{x_t},\boldsymbol{y_0}))$$

$$=\frac{1}{2}\left(\log\left(\frac{|\sigma_\theta^2\left(\boldsymbol{x_t},\boldsymbol{y_0},t\right)\boldsymbol{I}|}{|\tilde{\sigma}_t^2\boldsymbol{I}|}\right)-d+\mathrm{tr}\left(\left(\sigma_\theta^2\left(\boldsymbol{x_t},\boldsymbol{y_0},t\right)\boldsymbol{I}\right)^{-1}\tilde{\sigma}_t^2\boldsymbol{I}\right)\right.$$

$$\left.+\left(\boldsymbol{\mu_\theta}\left(\boldsymbol{x_t},\boldsymbol{y_0},t\right)-\tilde{\boldsymbol{\mu}}_{\boldsymbol{t}}\right)^\top\left(\sigma_\theta^2\left(\boldsymbol{x_t},\boldsymbol{y_0},t\right)\boldsymbol{I}\right)^{-1}\left(\boldsymbol{\mu_\theta}\left(\boldsymbol{x_t},\boldsymbol{y_0},t\right)-\tilde{\boldsymbol{\mu}}_{\boldsymbol{t}}\right)\right)$$

$$=\frac{1}{2}\left(\log\left(\frac{|\tilde{\sigma}_t^2\boldsymbol{I}|}{|\tilde{\sigma}_t^2\boldsymbol{I}|}\right)-d+\mathrm{tr}\left(\left(\tilde{\sigma}_t^2\boldsymbol{I}\right)^{-1}\tilde{\sigma}_t^2\boldsymbol{I}\right)\right.$$

$$\left.+\left(\boldsymbol{\mu_\theta}\left(\boldsymbol{x_t},\boldsymbol{y_0},t\right)-\tilde{\boldsymbol{\mu}}_{\boldsymbol{t}}\right)^\top\left(\tilde{\sigma}_t^2\boldsymbol{I}\right)^{-1}\left(\boldsymbol{\mu_\theta}\left(\boldsymbol{x_t},\boldsymbol{y_0},t\right)-\tilde{\boldsymbol{\mu}}_{\boldsymbol{t}}\right)\right) \tag{42}$$

$$= \frac{1}{2} \left( \left( \frac{1}{\tilde{\sigma}_t^2} \right) \left( \boldsymbol{\mu_\theta} \left( \boldsymbol{x_t}, \boldsymbol{y_0}, t \right) - \tilde{\boldsymbol{\mu}}_t \right)^\top \left( \boldsymbol{\mu_\theta} \left( \boldsymbol{x_t}, \boldsymbol{y_0}, t \right) - \tilde{\boldsymbol{\mu}}_t \right) \right)$$

$$= \frac{1}{2\tilde{\sigma}_t^2} \| \boldsymbol{\mu_\theta} \left( \boldsymbol{x_t}, \boldsymbol{y_0}, t \right) - \tilde{\boldsymbol{\mu}}_t \|^2,$$

where $|\cdot|$ denotes the determinant of a matrix, and $\mathrm{tr}(\cdot)$ is the trace of a matrix. Notably, minimizing the KL divergence effectively reduces to decreasing the difference between the means $\boldsymbol{\mu_\theta} \left( \boldsymbol{x_t}, \boldsymbol{y_0}, t \right)$ and $\tilde{\boldsymbol{\mu}}_t$.

However, this is only valid with $\gamma \neq 0$ and $\eta \neq 0$. In contrast, when either $\gamma = 0$ or $\eta = 0$, the true reverse transition Gaussian collapses into a Dirac delta function:

$$q(\boldsymbol{x_{t-1}}|\boldsymbol{x_t}, \boldsymbol{x_0}, \boldsymbol{\Delta}) = \begin{cases} \delta \left( \boldsymbol{x_{t-1}} - \tilde{\boldsymbol{\mu}}_{t|\gamma=0} \right), & \text{if } \gamma = 0, & \text{(A)} \\ \delta \left( \boldsymbol{x_{t-1}} - \tilde{\boldsymbol{\mu}}_{t|\gamma \neq 0} \right), & \text{if } \gamma \neq 0 \text{ and } \eta = 0, & \text{(B)} \\ \mathcal{N}(\boldsymbol{x_{t-1}}|\tilde{\boldsymbol{\mu}}_{t|\gamma \neq 0}, \tilde{\sigma}_{t|\gamma \neq 0}^2 \boldsymbol{I}), & \text{if } \gamma \neq 0 \text{ and } \eta \neq 0, & \text{(C)} \end{cases} \tag{43}$$

where A, B, and C correspond to the cases of $\gamma = 0$, $(\gamma \neq 0$ and $\eta = 0)$, and $(\gamma \neq 0$ and $\eta \neq 0)$, respectively.

The KL divergence between two Dirac delta functions is not defined in the conventional sense due to their singular nature, but it can be analyzed through limiting behavior. Two delta functions centered at different points have infinite divergence, thereby the KL divergence of each term $\mathcal{L}_{1:T-1}$ tends to infinity, when $\gamma = 0$ or $\eta = 0$, unless $\boldsymbol{\mu_\theta} \left( \boldsymbol{x_t}, \boldsymbol{y_0}, t \right) = \tilde{\boldsymbol{\mu}}_t$:

$$D_{\mathrm{KL}}(q(\boldsymbol{x_{t-1}}|\boldsymbol{x_t}, \boldsymbol{x_0}, \boldsymbol{\Delta}) \| p_\theta(\boldsymbol{x_{t-1}}|\boldsymbol{x_t}, \boldsymbol{y_0}))$$

$$= \begin{cases} 0, & \begin{aligned} &\text{if (A and } \boldsymbol{\mu}_{\boldsymbol{\theta}|\gamma=0} \left( \boldsymbol{x_t}, \boldsymbol{y_0}, t \right) = \tilde{\boldsymbol{\mu}}_{t|\gamma=0}) \\ &\text{or (B and } \boldsymbol{\mu}_{\boldsymbol{\theta}|\gamma \neq 0} \left( \boldsymbol{x_t}, \boldsymbol{y_0}, t \right) = \tilde{\boldsymbol{\mu}}_{t|\gamma \neq 0}), \end{aligned} \\ \\ \infty, & \begin{aligned} &\text{if (A and } \boldsymbol{\mu}_{\boldsymbol{\theta}|\gamma=0} \left( \boldsymbol{x_t}, \boldsymbol{y_0}, t \right) \neq \tilde{\boldsymbol{\mu}}_{t|\gamma=0}) \\ &\text{or (B and } \boldsymbol{\mu}_{\boldsymbol{\theta}|\gamma \neq 0} \left( \boldsymbol{x_t}, \boldsymbol{y_0}, t \right) \neq \tilde{\boldsymbol{\mu}}_{t|\gamma \neq 0}), \end{aligned} \\ \\ \frac{1}{2\tilde{\sigma}_{t|\gamma \neq 0}^2} \| \boldsymbol{\mu}_{\boldsymbol{\theta}|\gamma \neq 0} \left( \boldsymbol{x_t}, \boldsymbol{y_0}, t \right) - \tilde{\boldsymbol{\mu}}_{t|\gamma \neq 0} \|^2, & \text{if C.} \end{cases} \tag{44}$$

For the Dirac delta cases, where either $\gamma = 0$ or $\eta = 0$, to avoid an infinite loss, the only choice is to force $\boldsymbol{\mu_\theta} \left( \boldsymbol{x_t}, \boldsymbol{y_0}, t \right) = \tilde{\boldsymbol{\mu}}_t$. However, directly optimizing under such a hard constraint is infeasible in practice, as it provides no gradient information unless the condition is already satisfied. To circumvent this, a relaxed proxy objective is adopted, mirroring the approach used in the Gaussian case. Specifically, it minimizes half of the squared Euclidean distance between $\boldsymbol{\mu_\theta} \left( \boldsymbol{x_t}, \boldsymbol{y_0}, t \right)$ and $\tilde{\boldsymbol{\mu}}_t$. This mean-matching proxy loss serves as a differentiable surrogate that naturally encourages the model to align the means and can be interpreted as the limiting case of the KL divergence when the variance tends to zero. Consequently, the reduction of the KL divergence to mean matching holds for all scenarios of $\gamma$ and $\eta$.

Moreover, considering the formulation of $\tilde{\boldsymbol{\mu}}_t$ given in Equation (40) and since at every timestep, $t$, in the reverse process, only the exact values of $\boldsymbol{x_0}$ and $\boldsymbol{\Delta}$ are unknown, then $\boldsymbol{\mu_\theta} \left( \boldsymbol{x_t}, \boldsymbol{y_0}, t \right)$ can be defined as in Equation (12). In this definition of $\boldsymbol{\mu_\theta} \left( \boldsymbol{x_t}, \boldsymbol{y_0}, t \right)$, the only components dependent on the parameters $\theta$ are $\hat{\boldsymbol{x}}_0$ and $\hat{\boldsymbol{\Delta}}$. Since, $\boldsymbol{\Delta}$ can be estimated from $\boldsymbol{x_0}$ and $\boldsymbol{y_0}$, then the model solely needs to predict $\boldsymbol{x_0}$. Hence, $\hat{\boldsymbol{x}}_0 = f_\theta(\boldsymbol{x_t}, \boldsymbol{y_0}, t)$ denotes the $\boldsymbol{x_0}$ prediction from a neural network given $\boldsymbol{x_t}$, $\boldsymbol{y_0}$, and timestep $t$. Meanwhile, $\hat{\boldsymbol{\Delta}} = \hat{\boldsymbol{x}}_0 - \boldsymbol{y_0}$ represents the $\boldsymbol{\Delta}$ estimation, computed from the $\boldsymbol{x_0}$ prediction and the known $\boldsymbol{y_0}$. The remaining components are fixed hyperparameters and $\boldsymbol{x_t}$, which are known for every reverse transition from $\boldsymbol{x_t}$ at any timestep, $t$. In essence, the approximate reverse transition, $p_\theta(\boldsymbol{x_{t-1}}|\boldsymbol{x_t}, \boldsymbol{y_0})$, is modeled as a Gaussian whose mean is computed using a neural network that predicts $\boldsymbol{x_0}$. Accordingly, the KL divergence of each term $\mathcal{L}_{1:T-1}$ can be further expanded as:

$$D_{\mathrm{KL}}(q(\boldsymbol{x_{t-1}}|\boldsymbol{x_t}, \boldsymbol{x_0}, \boldsymbol{\Delta}) \| p_\theta(\boldsymbol{x_{t-1}}|\boldsymbol{x_t}, \boldsymbol{y_0}))$$

$$= \begin{cases} \frac{1}{2} \| \boldsymbol{\mu}_{\boldsymbol{\theta}|\gamma=0} \left( \boldsymbol{x_t}, \boldsymbol{y_0}, t \right) - \tilde{\boldsymbol{\mu}}_{t|\gamma=0} \|^2, & \text{if A,} \\ \frac{1}{2} \| \boldsymbol{\mu}_{\boldsymbol{\theta}|\gamma \neq 0} \left( \boldsymbol{x_t}, \boldsymbol{y_0}, t \right) - \tilde{\boldsymbol{\mu}}_{t|\gamma \neq 0} \|^2, & \text{if B,} \\ \frac{1}{2\tilde{\sigma}_{t|\gamma \neq 0}^2} \| \boldsymbol{\mu}_{\boldsymbol{\theta}|\gamma \neq 0} \left( \boldsymbol{x_t}, \boldsymbol{y_0}, t \right) - \tilde{\boldsymbol{\mu}}_{t|\gamma \neq 0} \|^2, & \text{if C,} \end{cases}$$

$$
= \begin{cases}
\frac{1}{2}\|\boldsymbol{x_0} - \beta_{t-1}\boldsymbol{\Delta} - (\hat{\boldsymbol{x}}_{\boldsymbol{0}} - \beta_{t-1}\hat{\boldsymbol{\Delta}})\|^2, & \text{if A,} \\[2ex]
\frac{1}{2}\left\| \boldsymbol{x_0} - \beta_{t-1}\boldsymbol{\Delta} + \sqrt{\gamma^2\beta_{t-1} - \eta^2\tilde{\lambda}_t}\left(\frac{\boldsymbol{x_t} - \boldsymbol{x_0} + \beta_t\boldsymbol{\Delta}}{\sqrt{\gamma^2\beta_t}}\right) \right. \\
\quad \left. - \left(\hat{\boldsymbol{x}}_{\boldsymbol{0}} - \beta_{t-1}\hat{\boldsymbol{\Delta}} + \sqrt{\gamma^2\beta_{t-1} - \eta^2\tilde{\lambda}_t}\left(\frac{\boldsymbol{x_t} - \hat{\boldsymbol{x}}_{\boldsymbol{0}} + \beta_t\hat{\boldsymbol{\Delta}}}{\sqrt{\gamma^2\beta_t}}\right)\right) \right\|^2, & \text{if B,} \\[3ex]
\frac{1}{2\eta^2\tilde{\lambda}_t}\left\| \boldsymbol{x_0} - \beta_{t-1}\boldsymbol{\Delta} + \sqrt{\gamma^2\beta_{t-1} - \eta^2\tilde{\lambda}_t}\left(\frac{\boldsymbol{x_t} - \boldsymbol{x_0} + \beta_t\boldsymbol{\Delta}}{\sqrt{\gamma^2\beta_t}}\right) \right. \\
\quad \left. - \left(\hat{\boldsymbol{x}}_{\boldsymbol{0}} - \beta_{t-1}\hat{\boldsymbol{\Delta}} + \sqrt{\gamma^2\beta_{t-1} - \eta^2\tilde{\lambda}_t}\left(\frac{\boldsymbol{x_t} - \hat{\boldsymbol{x}}_{\boldsymbol{0}} + \beta_t\hat{\boldsymbol{\Delta}}}{\sqrt{\gamma^2\beta_t}}\right)\right) \right\|^2, & \text{if C,}
\end{cases}
$$

$$
= \begin{cases}
\frac{1}{2}\|\boldsymbol{x_0} - \hat{\boldsymbol{x}}_{\boldsymbol{0}} - \beta_{t-1}(\boldsymbol{x_0} - \hat{\boldsymbol{x}}_{\boldsymbol{0}})\|^2, & \text{if A,} \\[2ex]
\frac{1}{2}\left\| \boldsymbol{x_0} - \hat{\boldsymbol{x}}_{\boldsymbol{0}} - \beta_{t-1}(\boldsymbol{x_0} - \hat{\boldsymbol{x}}_{\boldsymbol{0}}) \right. \\
\quad \left. + \sqrt{\frac{\beta_{t-1}}{\beta_t}}(\hat{\boldsymbol{x}}_{\boldsymbol{0}} - \boldsymbol{x_0} + \beta_t(\boldsymbol{x_0} - \hat{\boldsymbol{x}}_{\boldsymbol{0}})) \right\|^2, & \text{if B,} \\[3ex]
\frac{1}{2\eta^2\tilde{\lambda}_t}\left\| \boldsymbol{x_0} - \hat{\boldsymbol{x}}_{\boldsymbol{0}} - \beta_{t-1}(\boldsymbol{x_0} - \hat{\boldsymbol{x}}_{\boldsymbol{0}}) \right. \\
\quad \left. + \sqrt{\frac{\gamma^2\beta_{t-1} - \eta^2\tilde{\lambda}_t}{\gamma^2\beta_t}}(\hat{\boldsymbol{x}}_{\boldsymbol{0}} - \boldsymbol{x_0} + \beta_t(\boldsymbol{x_0} - \hat{\boldsymbol{x}}_{\boldsymbol{0}})) \right\|^2, & \text{if C,}
\end{cases} \tag{45}
$$

$$
= \begin{cases}
\frac{1}{2}\|(\boldsymbol{x_0} - \hat{\boldsymbol{x}}_{\boldsymbol{0}})(1 - \beta_{t-1})\|^2, & \text{if A,} \\[2ex]
\frac{1}{2}\left\| (\boldsymbol{x_0} - \hat{\boldsymbol{x}}_{\boldsymbol{0}})(1 - \beta_{t-1}) \right. \\
\quad \left. + \sqrt{\frac{\beta_{t-1}}{\beta_t}}(\hat{\boldsymbol{x}}_{\boldsymbol{0}} - \boldsymbol{x_0} + \beta_t(\boldsymbol{x_0} - \hat{\boldsymbol{x}}_{\boldsymbol{0}})) \right\|^2, & \text{if B,} \\[3ex]
\frac{1}{2\eta^2\tilde{\lambda}_t}\left\| (\boldsymbol{x_0} - \hat{\boldsymbol{x}}_{\boldsymbol{0}})(1 - \beta_{t-1}) \right. \\
\quad \left. + \sqrt{\frac{\gamma^2\beta_{t-1} - \eta^2\tilde{\lambda}_t}{\gamma^2\beta_t}}(\hat{\boldsymbol{x}}_{\boldsymbol{0}} - \boldsymbol{x_0} + \beta_t(\boldsymbol{x_0} - \hat{\boldsymbol{x}}_{\boldsymbol{0}})) \right\|^2, & \text{if C,}
\end{cases}
$$

$$
= \begin{cases}
\frac{1 - \beta_{t-1}}{2}\|\boldsymbol{x_0} - \hat{\boldsymbol{x}}_{\boldsymbol{0}}\|^2, & \text{if A,} \\[2ex]
\frac{1}{2}\left\| (\boldsymbol{x_0} - \hat{\boldsymbol{x}}_{\boldsymbol{0}})(1 - \beta_{t-1}) - \sqrt{\frac{\beta_{t-1}}{\beta_t}}(\boldsymbol{x_0} - \hat{\boldsymbol{x}}_{\boldsymbol{0}})(1 - \beta_t) \right\|^2, & \text{if B,} \\[2ex]
\frac{1}{2\eta^2\tilde{\lambda}_t}\left\| (\boldsymbol{x_0} - \hat{\boldsymbol{x}}_{\boldsymbol{0}})(1 - \beta_{t-1}) - \sqrt{\frac{\gamma^2\beta_{t-1} - \eta^2\tilde{\lambda}_t}{\gamma^2\beta_t}}(\boldsymbol{x_0} - \hat{\boldsymbol{x}}_{\boldsymbol{0}})(1 - \beta_t) \right\|^2, & \text{if C,}
\end{cases}
$$

$$
= \begin{cases}
\frac{1 - \beta_{t-1}}{2}\|\boldsymbol{x_0} - \hat{\boldsymbol{x}}_{\boldsymbol{0}}\|^2, & \text{if A,} \\[2ex]
\frac{1}{2}\left\| (\boldsymbol{x_0} - \hat{\boldsymbol{x}}_{\boldsymbol{0}})\left(1 - \beta_{t-1} - \sqrt{\frac{\beta_{t-1}}{\beta_t}}(1 - \beta_t)\right) \right\|^2, & \text{if B,} \\[2ex]
\frac{1}{2\eta^2\tilde{\lambda}_t}\left\| (\boldsymbol{x_0} - \hat{\boldsymbol{x}}_{\boldsymbol{0}})\left(1 - \beta_{t-1} - \sqrt{\frac{\gamma^2\beta_{t-1} - \eta^2\tilde{\lambda}_t}{\gamma^2\beta_t}}(1 - \beta_t)\right) \right\|^2, & \text{if C,}
\end{cases}
$$

$$
= \begin{cases}
\frac{1 - \beta_{t-1}}{2}\|\boldsymbol{x_0} - \hat{\boldsymbol{x}}_{\boldsymbol{0}}\|^2, & \text{if } \gamma = 0, \\[2ex]
\frac{1 - \beta_{t-1} - \sqrt{\frac{\beta_{t-1}}{\beta_t}(1 - \beta_t)}}{2}\|\boldsymbol{x_0} - \hat{\boldsymbol{x}}_{\boldsymbol{0}}\|^2, & \text{if } \gamma \neq 0 \text{ and } \eta = 0, \\[2ex]
\frac{1 - \beta_{t-1} - \sqrt{\frac{\gamma^2\beta_{t-1} - \eta^2\tilde{\lambda}_t}{\gamma^2\beta_t}}(1 - \beta_t)}{2\eta^2\tilde{\lambda}_t}\|\boldsymbol{x_0} - \hat{\boldsymbol{x}}_{\boldsymbol{0}}\|^2, & \text{if } \gamma \neq 0 \text{ and } \eta \neq 0,
\end{cases}
$$

$$
= \omega_t(\gamma, \eta, t)\|\boldsymbol{x_0} - \hat{\boldsymbol{x}}_{\boldsymbol{0}}\|^2.
$$

Therefore, irrespective of the specific values of $\gamma$ and $\eta$, each consistency term $\mathcal{L}_{1:T-1}$ ultimately reduces to the expectation of a weighted squared Euclidean distance between the original data $\boldsymbol{x_0}$ and its prediction, where the expectation is taken over $q(\boldsymbol{x_t}|\boldsymbol{x_0}, \boldsymbol{\Delta})$:

$$
\mathcal{L}_{t-1} = \mathbb{E}_{q(\boldsymbol{x_t}|\boldsymbol{x_0}, \boldsymbol{\Delta})}\left[\omega_t(\gamma, \eta, t)\|\boldsymbol{x_0} - \hat{\boldsymbol{x}}_{\boldsymbol{0}}\|^2\right], \tag{46}
$$

with weights $\omega_t(\cdot)$ defined as a function of $\gamma$, $\eta$, and $t$. Essentially, approximating $\hat{x}_0$ to the original data effectively ensures that $\mu_\theta(x_t, y_0, t)$ converges to $\tilde{\mu}_t$. As a result, $p_\theta(x_{t-1}|x_t, y_0)$ accurately models $q(x_{t-1}|x_t, x_0, \Delta)$, which is the primary purpose of the consistency terms $\mathcal{L}_{1:T-1}$.

In particular, due to the relationship between $x_0$ and $\epsilon$ given in Equation (32), the objective derived in Equation (45) could be converted to predicting noise $\epsilon$ similar to DDPMs (Ho et al., 2020). However, this reformulation of the objective would not be possible with a deterministic forward process ($\gamma = 0$), as it works only for cases where noise was added during the forward process ($\gamma \neq 0$). Hence, having the neural network predict $x_0$ directly is preferred for broader applicability and improved generalizability.

Notably, the mean is continuous at $\gamma = 0$ (see Appendix A.2), thus there are no problems during gradient computation, such as taking gradients where a function is not differentiable. Nonetheless, for each specific value of $\gamma$, the mean is continuous and the $\gamma$ constant hyperparameter is immutable, i.e., set only once for each model instance, thereby no discontinuity issues would ever arise due to $\gamma$.

**Reconstruction term $\mathcal{L}_0$.** The $\mathcal{L}_0$ term is essentially the expectation of the negative log-likelihood (NLL) of the original data, $x_0$, conditioned on the first latent variable, $x_1$, and the corrupted version, $y_0$, where the expectation is taken over $x_1 \sim q(x_1|x_0, \Delta)$. In essence, it quantifies how well the model can reconstruct $x_0$ given $x_1$ and $y_0$. Since minimizing the NLL encourages the model to output high-probability (accurate) reconstructions, it can be interpreted as a reconstruction loss. Conceptually, this term acts as a final quality check, ensuring that after practically all the diffusion degradation is removed[3] iteratively, the model can accurately reconstruct the original clean data, $x_0$, from the almost degradation-free input, $x_1$. It assures that the model not only learns to refine the data at intermediate timesteps, but also produces outputs consistent with the underlying real data distribution conditioned on $y_0$. As a result, it contributes to aligning the model marginal $p_\theta(x_0|y_0)$ with the true posterior distribution $q(x_0|y_0)$ as given in Equation (17). Nonetheless, similar to DDPMs, this term is omitted in practice, since it is implicitly included in a simplified training objective.

**Simplified objective function.** Since the term $\mathcal{L}_T$ can be excluded from the optimization objective, the loss function in Equation (39) becomes:

$$\mathcal{L}(\theta) = \cancel{\mathcal{L}_T} + \mathcal{L}_{1:T-1} + \mathcal{L}_0 = \sum_{t=2}^{T} \mathcal{L}_{t-1} + \mathcal{L}_0$$
$$= \sum_{t=2}^{T} \mathbb{E}_{q(x_t|x_0, \Delta)} \left[ D_{\mathrm{KL}}(q(x_{t-1}|x_t, x_0, \Delta) \| p_\theta(x_{t-1}|x_t, y_0)) \right] + \mathcal{L}_0. \tag{47}$$

Following, the term $\mathcal{L}_0$ can be omitted, as it is implicitly included by extending the sum to encompass all timesteps, $t \in \{1, 2, \ldots, T\}$, thereby accounting for the transition from $x_1$ to $x_0$:

$$\mathcal{L}(\theta) = \sum_{t=1}^{T} \mathbb{E}_{q(x_t|x_0, \Delta)} \left[ D_{\mathrm{KL}}(q(x_{t-1}|x_t, x_0, \Delta) \| p_\theta(x_{t-1}|x_t, y_0)) \right], \tag{48}$$

and given Equation (45), then:

$$\mathcal{L}(\theta) = \sum_{t=1}^{T} \mathbb{E}_{q(x_t|x_0, \Delta)} \left[ D_{\mathrm{KL}}(q(x_{t-1}|x_t, x_0, \Delta) \| p_\theta(x_{t-1}|x_t, y_0)) \right]$$
$$= \sum_{t=1}^{T} \mathbb{E}_{q(x_t|x_0, \Delta)} \left[ \omega_t(\gamma, \eta, t) \| x_0 - \hat{x}_0 \|^2 \right]. \tag{49}$$

Considering $\tilde{\lambda}_t = \gamma^2 \frac{\beta_{t-1}}{\beta_t} \lambda_t$, the weights $\omega_t$ only depend on the predefined $\gamma$, $\eta$, and $\beta$-schedule constant hyperparameters. In many practical implementations, such as DDPMs, this weighting is

---

[3]The forward process progressively incorporates degradation and removes $\Delta$. The reverse process removes degradation and reintroduces $\Delta$.

often omitted for all timesteps, finding that this still produces excellent results (Ho et al., 2020; Yue et al., 2023). Therefore, the loss function can be further simplified by excluding the scaling:

$$\mathcal{L}(\theta) = \sum_{t=1}^{T} \mathbb{E}_{q(\boldsymbol{x_t}|\boldsymbol{x_0},\boldsymbol{\Delta})} \left[ \|\boldsymbol{x_0} - \hat{\boldsymbol{x}}_0\|^2 \right], \tag{50}$$

and since evaluating the full sum over all time steps is computationally expensive, a single time step can be sampled per training example. This yields an unbiased estimator of the full objective and significantly improves training efficiency:

$$\mathcal{L}_{\text{simple}}(\theta) = \mathbb{E}_{\boldsymbol{x_0},\boldsymbol{\Delta},t} \left[ \|\boldsymbol{x_0} - \hat{\boldsymbol{x}}_0\|^2 \right], \tag{51}$$

where $\boldsymbol{x_t} \sim q(\boldsymbol{x_t}|\boldsymbol{x_0},\boldsymbol{\Delta})$, $t \sim \mathcal{U}(1,T)$, and the case $t = 1$ corresponds to $\mathcal{L}_0$. Consequently, the objective function of RDIMs simplifies to a squared Euclidean distance between the original data and its prediction. Notably, RDIM and ResShift lead to the same training objective, further highlighting that ResShift is a particular case of RDIM. This follows from the objective depending only on the marginal distribution $q(\boldsymbol{x_t}|\boldsymbol{x_0},\boldsymbol{\Delta})$, which both models share. It does not strictly require the forward process to be a Markov chain.

# B  LEMMAS

This section presents lemmas that support this work. These lemmas provide foundational results and properties that support the main arguments and proofs.

**Lemma B.1 (Bishop & Nasrabadi (2006))** *Given a marginal Gaussian distribution for random variable $\boldsymbol{x}$ and a conditional Gaussian distribution for random variable $\boldsymbol{y}$ given $\boldsymbol{x}$ in the form:*

$$\begin{aligned} p(\boldsymbol{x}) &= \mathcal{N}\left(\boldsymbol{x}|\boldsymbol{\mu_x},\boldsymbol{\Sigma_x}\right), \\ p(\boldsymbol{y}|\boldsymbol{x}) &= \mathcal{N}\left(\boldsymbol{y}|\boldsymbol{C}\boldsymbol{x} + \boldsymbol{c},\boldsymbol{\Sigma_{y|x}}\right), \end{aligned} \tag{52}$$

*where $\boldsymbol{\mu_x}$, $\boldsymbol{C}$, and $\boldsymbol{c}$ are parameters governing the means, while $\boldsymbol{\Sigma_x}$ and $\boldsymbol{\Sigma_{y|x}}$ denote covariance matrices. Then the marginal distribution of $\boldsymbol{y}$ and the conditional distribution of $\boldsymbol{x}$ given $\boldsymbol{y}$ are in the form:*

$$\begin{aligned} p(\boldsymbol{y}) &= \mathcal{N}\left(\boldsymbol{y}|\boldsymbol{C}\boldsymbol{\mu_x} + \boldsymbol{c}, \boldsymbol{\Sigma_{y|x}} + \boldsymbol{C}\boldsymbol{\Sigma_x}\boldsymbol{C}^\top\right), \\ p(\boldsymbol{x}|\boldsymbol{y}) &= \mathcal{N}\left(\boldsymbol{x}\middle|\boldsymbol{\Sigma_{x|y}}\left(\boldsymbol{C}^\top\boldsymbol{\Sigma_{y|x}^{-1}}\left(\boldsymbol{y} - \boldsymbol{c}\right) + \boldsymbol{\Sigma_x^{-1}}\boldsymbol{\mu_x}\right), \boldsymbol{\Sigma_{x|y}}\right), \end{aligned} \tag{53}$$

*with $\boldsymbol{\Sigma_{x|y}}$ representing the conditional covariance matrix of $\boldsymbol{x}$ given $\boldsymbol{y}$, defined as:*

$$\boldsymbol{\Sigma_{x|y}} = \left(\boldsymbol{\Sigma_x^{-1}} + \boldsymbol{C}^\top\boldsymbol{\Sigma_{y|x}^{-1}}\boldsymbol{C}\right)^{-1}. \tag{54}$$

**Lemma B.2 (Bishop & Nasrabadi (2006))** *Given a joint Gaussian distribution over random variables $\boldsymbol{x}$ and $\boldsymbol{y}$ of the form:*

$$p\left(\begin{bmatrix}\boldsymbol{x} \\ \boldsymbol{y}\end{bmatrix}\right) = \mathcal{N}\left(\begin{bmatrix}\boldsymbol{\mu_x} \\ \boldsymbol{\mu_y}\end{bmatrix}, \begin{bmatrix}\boldsymbol{\Sigma_{xx}} & \boldsymbol{\Sigma_{xy}} \\ \boldsymbol{\Sigma_{yx}} & \boldsymbol{\Sigma_{yy}}\end{bmatrix}\right), \tag{55}$$

*where $\boldsymbol{\mu_x}$ and $\boldsymbol{\mu_y}$ are the mean vectors of $\boldsymbol{x}$ and $\boldsymbol{y}$, respectively, while $\boldsymbol{\Sigma_{xx}}$, $\boldsymbol{\Sigma_{xy}}$, $\boldsymbol{\Sigma_{yx}}$, and $\boldsymbol{\Sigma_{yy}}$ denote covariance matrices. Then the conditional distribution of $\boldsymbol{x}$ given $\boldsymbol{y}$ is Gaussian:*

$$p(\boldsymbol{x}|\boldsymbol{y}) = \mathcal{N}\left(\boldsymbol{x}\middle|\boldsymbol{\mu_{x|y}},\boldsymbol{\Sigma_{x|y}}\right), \tag{56}$$

*with the conditional mean and covariance given by:*

$$\begin{aligned} \boldsymbol{\mu_{x|y}} &= \boldsymbol{\mu_x} + \boldsymbol{\Sigma_{xy}}\boldsymbol{\Sigma_{yy}^{-1}}(\boldsymbol{y} - \boldsymbol{\mu_y}), \\ \boldsymbol{\Sigma_{x|y}} &= \boldsymbol{\Sigma_{xx}} - \boldsymbol{\Sigma_{xy}}\boldsymbol{\Sigma_{yy}^{-1}}\boldsymbol{\Sigma_{yx}}, \end{aligned} \tag{57}$$

*where the expressions follow from the Schur complement. This result shows that the conditional mean of $\boldsymbol{x}$ given $\boldsymbol{y}$ is a linear function of $\boldsymbol{y}$.*

## C   Experimental Details and Additional Results

This section presents experimental details and additional results that complement those discussed in the main text.

### C.1   Datasets

Experiments were performed across eight subsets, derived from four public data collections, namely (i) Fluorescence Microscopy Denoising (FMD) dataset (Zhang et al., 2019), (ii) DIVerse 2K Resolution High Quality Images (DIV2K) dataset (Agustsson & Timofte, 2017; Timofte et al., 2017), (iii) Smartphone Image Denoising Dataset (SIDD) (Abdelhamed et al., 2018; 2019), and (iv) Flickr–Faces-HQ (FFHQ) dataset (Karras et al., 2019).

The FMD dataset is specifically designed for Poisson-Gaussian denoising tasks and consists of 12,000 real images acquired from representative biological samples, including bovine pulmonary artery endothelial (BPAE) cells, zebrafish embryos, and mouse brain tissues, using confocal, two-photon, and wide-field modalities. The dataset contains images with multiple noise levels, resulting in several subsets, but only the strongest noise level (labeled raw in Zhang et al. (2019)) subsets are considered, thus prioritizing the most challenging conditions. Solely confocal images were used and mouse images are excluded. Accordingly, the two FMD dataset partitions used are Confocal-BPAE-Raw with 1,000 noisy-clean image pairs and Confocal-Zebrafish-Raw with 1,000 pairs. Moreover, each subset was randomly partitioned into training, test and validation splits, corresponding to 80%, 10%, and 10% of the data, respectively.

DIV2K is a publicly available benchmark dataset originally introduced for the NTIRE 2017 Challenge on Single Image Super-Resolution. It is specifically designed for SR tasks and comprises a collection of HR images along with their corresponding low-resolution (LR) counterparts. Each HR image in the dataset is paired with several downscaled versions, generated through different degradation operations and scaling factors of 2, 3, and 4. Particularly, three subsets of DIV2K with unknown and bicubic degradation operators are used, namely DIV2K-Unknown-$\times$2, DIV2K-Unknown-$\times$4, and DIV2K-Bicubic-$\times$4. Each subset includes 1,000 LR-HR image pairs and is divided into 800 images used for training, 100 for validation, and 100 for testing. The validation split will be employed to evaluate the performance of the models as the testing split is not available.

The SIDD dataset is specifically designed for image denoising tasks, particularly focusing on real-world noisy images captured with smartphone cameras. The dataset consists of $\approx 30{,}000$ noisy images with their corresponding clean ground truth, from 10 scenes under different lighting conditions and using five representative smartphone cameras, hence spanning a wide range of image types and noise levels. Only images from the SIDD-Medium subset are used, comprising 320 noisy-clean image pairs. Ultimately, SIDD-Medium was randomly partitioned into training, test and validation splits, corresponding to 80%, 10%, and 10% of the data, respectively.

The FFHQ dataset consists of 70,000 HQ human face images, originally created as a benchmark for generative adversarial networks (GANs). It contains faces with considerable variation in terms of age, ethnicity, and image background. In this work, it is used for image inpainting, colorization, and deblurring. For computational efficiency, images were downsampled to a quarter of the original resolution using bicubic interpolation. Subsequently, corrupted-original image pairs were generated, resulting in three task-specific subsets, namely FFHQ-Inpainting, FFHQ-Colorization, and FFHQ-Deblurring. For image inpainting, pixels in the original images are randomly masked and set to zero with probability $p_{\text{mask}} = 0.5$. For colorization, grayscale inputs are obtained by converting the original RGB images to luminance. For deblurring, synthetic blurred images are generated from ground truth images by applying a Gaussian blur with kernel size $15 \times 15$ and standard deviation $\sigma = 3.0$. Each subset was randomly partitioned into training, validation, and test splits corresponding to 80%, 10%, and 10% of the data, respectively.

### C.2   Network Architecture

RDIM employs a U-Net-based architecture to predict $\hat{x}_0$ at each iteration of the reverse process. As illustrated in Figure 5, the network is composed of encoder, bottleneck, and decoder blocks, with skip connections linking encoder and decoder blocks at matching spatial resolutions. For

SR tasks, an upsample block transforms $\boldsymbol{y_0}$ to match the dimensionality (number of channels and resolution) expected by the network. For other tasks, this layer simplifies to a projection layer. At each iteration, the network is conditioned on a timestep embedding, which is computed with sinusoidal positional encoding and transformed through a small multilayer perceptron (MLP) consisting of a fully connected layer, a Swish activation, and a second fully connected layer. This embedding encodes the current diffusion step, providing information about the position within the reverse process.

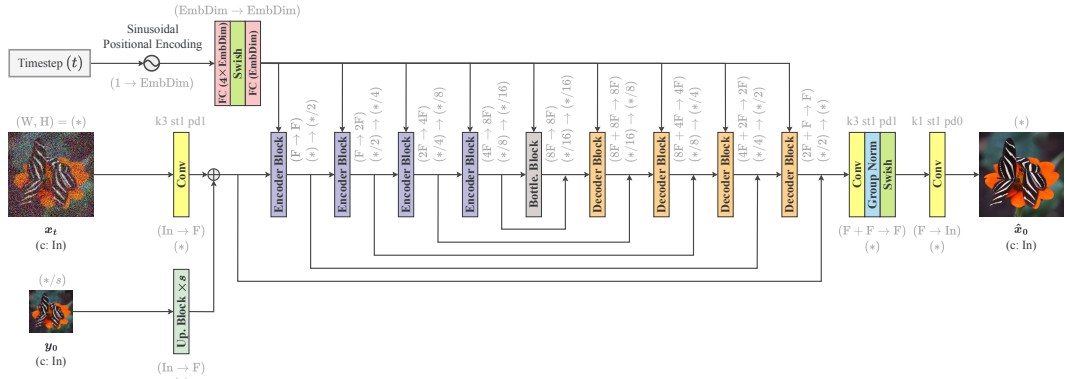

Figure 5: U-Net-based network. In the convolutional layers, the parameters $k$, $st$, and $pd$ represent the kernel size, stride, and padding, respectively. Additionally, $(*/s)$ denotes $(W/s, H/s)$, where $s$ is a scale factor ($s > 1$ for SR tasks and $s = 1$ otherwise).

Figure 6 shows the core blocks of the network. Each encoder block consists of multiple residual blocks, each optionally followed by a self-attention block, and concludes with a downsample block to reduce spatial resolution. Bottleneck blocks operate at the lowest spatial resolution and consist of multiple residual blocks interleaved with self-attention blocks. Decoder blocks consist of multiple residual blocks, each optionally followed by a self-attention block, and conclude with an upsample block to increase spatial resolution. Notably, all residual blocks incorporate the timestep embedding. Self-attention blocks are included only at the two lowest spatial resolution levels of the encoder and decoder blocks due to computational constraints at higher resolutions.

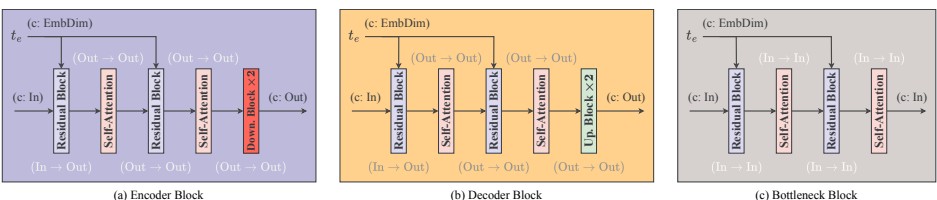

Figure 6: Core blocks of the U-Net-based network. (a) Encoder Block, (b) Decoder Block, and (c) Bottleneck Block.

The building blocks of the network are illustrated in Figure 7. Each residual block applies two convolutional layers with group normalization and Swish activation. They also contain a projection layer for the timestep embedding, composed of a Swish activation followed by a fully connected layer. Moreover, if the number of input channels (In) does not match the number of output channels (Out), an additional convolutional layer is included in the skip connection to project the input to the expected number of channels (Out), ensuring that the element-wise addition is well-defined. Self-attention blocks model long-range dependencies and incorporate group normalization both before and after the attention mechanism, operating over flattened spatial dimensions. Upsample and downsample blocks perform spatial resizing. Upsample blocks first perform bilinear interpolation (trilinear in case of 3D settings) to increase spatial resolution, followed by a convolutional layer, while downsample blocks perform convolution with stride greater than 1 (st > 1) to reduce spatial resolution. In the current implementation, activations are omitted, although the generalized block design can optionally include them.

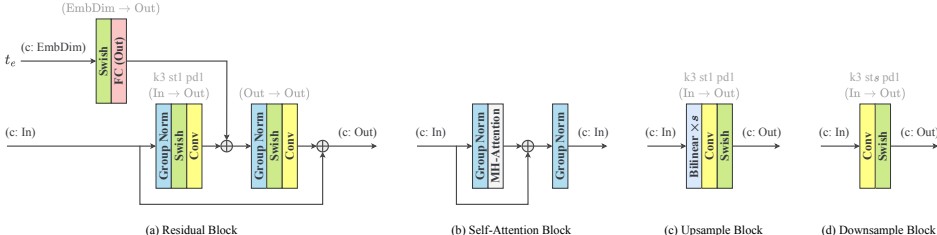

Figure 7: Building blocks. (a) Residual Block, (b) Self-Attention Block, (c) Upsample Block, and (d) Downsample Block.

### C.3 IMPLEMENTATION DETAILS

RDIM is implemented in PyTorch 2.5.1 (Paszke et al., 2019) and trained using the Adam optimizer (Kingma & Ba, 2014) with $\beta_1 = 0.9$ and $\beta_2 = 0.999$. The learning rate was initialized at $1.0 \times 10^{-4}$ and decayed following a cosine annealing schedule with minimum value $\eta_{\min} = 1.0 \times 10^{-9}$. Additionally, RDIM-PQ was trained using a combination of mean squared error (MSE) and SSIM losses, with the total loss defined as $\mathcal{L} = \mathcal{L}_{\text{MSE}} + \alpha \mathcal{L}_{\text{SSIM}}$, where $\alpha = 5.0 \times 10^{-2}$. All experiments were conducted with a batch size of $64$ and an effective patch resolution of $64 \times 64$ (except for DIV2K-Bicubic-$\times 4$, where a larger resolution of $128 \times 128$ was used). For SR, this corresponds to LR patch sizes of $32 \times 32$ and $16 \times 16$ for $\times 2$ and $\times 4$ scale factors, respectively (scaled proportionally for DIV2K-Bicubic-$\times 4$). The implementation is available at https://anonymous.4open.science/r/RDIM/.

DDPM, DDIM, ResShift, and RDIM were trained with the same number of diffusion timesteps ($T = 50$ for FFHQ and $T = 100$ for experiments on FMD, SIDD, and DIV2K) and network architecture with 128 base channels (detailed in Appendix C.2). The only difference lies in the diffusion framework employed. ResShift is a specific case of RDIM, thus a single network was trained for both. For DDPM, following SR3 (Saharia et al., 2022b), the model learns to approximate a reverse process, starting from pure Gaussian noise and iteratively denoising $x_t$ toward the HQ image, $x_0$, by predicting noise at each step, while conditioned on the LQ input, $y_0$. DDIM employed the network trained in the DDPM framework. Training was conducted for 4,000,000 iterations on FMD-Confocal datasets and DIV2K-Unknown subsets, 280,000 iterations on the DIV2K-Bicubic-$\times 4$ subset, 640,000 iterations on SIDD, and 4,375,000 iterations on FFHQ. For SR tasks in RDIM and ResShift, the LR input, $y_0$, is upsampled to the target HR resolution using bilinear interpolation, ensuring compatibility with the resolution employed in the diffusion framework (i.e., the size of $x_0, x_1, \ldots, x_T$).

All other techniques used in the comparative analysis of Section 3 strictly followed the reference papers and the official source codes. BM3D was applied with noise standard deviations of 10 for FMD-Confocal-BPAE-Raw, 30 for Confocal-Zebrafish-Raw, and 50 for SIDD-Medium. DnCNN was trained for 2,500,000 iterations on FMD-Confocal and SIDD datasets. ESRGAN was trained for a total of 1,400,000 iterations, with 1,000,000 iterations used to train a PSNR-oriented model that serves as initialization for the adversarial model, which was optimized for the remaining 400,000 iterations. Ultimately, GOUB and UniDB were trained for 900,000 iterations on DIV2K-Bicubic-$\times 4$, while CTMSR was trained for 500,000 iterations.

### C.4 UNIFORM SAMPLING TIMESTEP SCHEDULE

At inference, RDIM intends to reconstruct the original data, $x_0$, starting from the degraded final latent variable, $x_T$. Unlike DDPMs and ResShift, where the sampling process requires iterating over all diffusion timesteps, $T$, the RDIM reverse process can be simulated with fewer timesteps. This results from the formulation of the RDIM reverse transition, which allows skipping intermediate timesteps during sampling (see Section 2.5). Accordingly, this flexibility motivates the selection of a subset, $\Upsilon$, of $S < T$ sampling timesteps to traverse the reverse trajectory.

A simple yet effective approach is to adopt a linear sampling schedule, where the selected timesteps are uniformly spaced. Geometric schedules with denser allocation toward earlier or later stages of the

reverse process were empirically evaluated, but they underperformed against a uniform alternative or yielded marginal improvements. As a result, the following uniform scheduler is devised:

$$\Upsilon = \left\{ \tau_k = \left\lfloor \frac{k}{S} \cdot T \right\rfloor \;\middle|\; k \in \{0, 1, \ldots, S\} \right\}, \tag{58}$$

where, during sampling, $\Upsilon$ is iterated from $\tau_S = T$ to $\tau_1$, resulting in the order of sampling points $\tau_S \to \tau_{S-1} \to \cdots \to \tau_1$. Reverse transitions occur exclusively at these selected timesteps, from each $\tau_t$ to $\tau_{t-1}$, with all intermediate timesteps being skipped. The exception is the target timestep $\tau_0 = 0$, which marks the end of the reverse trajectory and does not produce a further transition. Moreover, all adjacent sampling timestep pairs, $(\tau_{k-1}, \tau_k)$, satisfy the following condition:

$$(\tau_{k-1}, \tau_k) \in \left\{ (t', t) \in \mathbb{N}_0^2 \mid t' + 1 \leq t \leq T \right\}. \tag{59}$$

In essence, only the latent variables associated with these timesteps are sampled, enabling a more efficient inference process. Figure 8 illustrates the sampling points (where reverse transitions occur) along the reverse trajectory, contextualized with the corresponding values of the $\beta$-schedule.

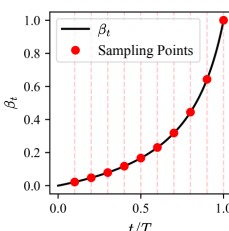

Figure 8: Residual $\beta$-schedule ($p = 5.0$) overlaid with red markers indicating the $\beta_t$ value at each sampling point. The schedule adopted selects timesteps uniformly spaced. In this illustration, the forward process involves $T = 100$ timesteps and the number of sampling steps, where reverse transitions occur, is $S = 10$.

### C.5 IMPACT OF $\beta$-SCHEDULE PARAMETER $p$

Experiments were conducted to determine an appropriate value for the $\beta$-schedule parameter $p$ (see Section 2.6). In these experiments, RDIM employs $T = 10$ diffusion timesteps and a forward variance hyperparameter $\gamma = 9.0$. During inference, the reverse variance hyperparameter is set to $\eta = 1.0$ and multiple reverse trajectory lengths were evaluated. All other implementation details follow those described in Appendix C.3.

Table 3: Impact of parameter $p$, which controls the steepness of the curve in the $\beta$-schedule. All RDIM configurations were trained using a forward process with $T = 10$ timesteps and variance hyperparameter $\gamma = 9.0$. During inference, the reverse process variance hyperparameter is fixed to $\eta = 1.0$. Orange color rows highlight ResShift scenarios, corresponding to particular cases where RDIM reduces to ResShift under the conditions $\eta = 1.0$ and $S = T$.

| $p$ | $S$ | FMD-Confocal-BPAE-Raw | | |
|---|---|---|---|---|
| | | PSNR↑ | SSIM↑ | LPIPS↓ |
| | 1 | 40.0836 | 0.9678 | 0.0205 |
| 1.0 | 5 | 40.0772 | 0.9678 | 0.0205 |
| | 10 | 40.0565 | 0.9677 | 0.0205 |
| | 1 | **40.1100** | **0.9681** | **0.0202** |
| 5.0 | 5 | 40.0436 | 0.9679 | **0.0202** |
| | 10 | 39.9524 | 0.9674 | **0.0202** |
| | 1 | 39.4014 | 0.9639 | 0.0238 |
| 15.0 | 5 | 39.1177 | 0.9624 | 0.0236 |
| | 10 | 38.8951 | 0.9609 | 0.0236 |

Table 3 presents the results for $p = 1$, $p = 5$, and $p = 15$. It follows that on denoising images from the FMD-Confocal-BPAE-Raw dataset, RDIM with $T = 10$ and $\gamma = 9.0$ achieves the best overall performance when $p = 5.0$. Moreover, irrespective of the steepness of the $\beta$-schedule, skipping timesteps and using fewer reverse timesteps ($S < T$) consistently yields superior results in terms of PSNR and SSIM compared to iterating through all diffusion steps ($S = T$). Particularly, Figure 2 in Section 2.6 illustrates the $\beta$-schedule curves and the effect on the diffusion process corresponding to these parameter values.

## C.6 IMPACT OF VARIANCE PARAMETER $\gamma$

The diffusion process variance is controlled with a constant hyperparameter $\gamma \in [0, \infty)$, which allows interpolation between a deterministic ($\gamma = 0 \Rightarrow$ Gaussian collapses into a $\delta$-distribution) and a stochastic ($\gamma > 0$) forward process. Figure 9 illustrates the impact of $\gamma$ on the forward process.

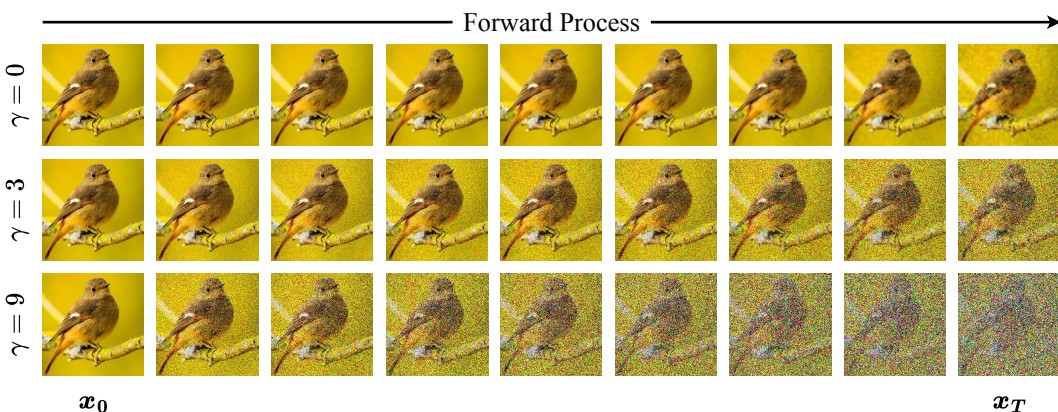

Figure 9: Impact of $\gamma$ on the diffusion process with $\beta$-schedule parameter fixed to $p = 5.0$.

## C.7 COMPARATIVE ANALYSIS OF MULTIPLE RDIM CONFIGURATIONS IN IMAGE DENOISING

To identify the best RDIM configuration, several setups were compared on denoising of BPAE confocal images from the benchmark dataset FMD. The experiments explore the impact of the diffusion chain length ($T$), the number of sampling timesteps ($S$), and the variance controlled by the constant hyperparameters $\gamma$ and $\eta$. Setups with $T = 100$ followed the implementation details described in Appendix C.3. For configurations with a different number of diffusion timesteps, the number of iterations (and consequently the training time) was adjusted linearly in proportion to the number of diffusion steps, $T$. This ensures that each timestep undergoes a similar number of weight updates across all configurations, thereby preventing imbalanced training between timesteps in configurations with different chain lengths. All other implementation details follow those described in Appendix C.3.

Table 4 summarizes the results. The RDIM configuration with $\gamma = 3.0$, $T = 100$, and $S = 10$ achieves the best performance in terms of PSNR and SSIM. Overall, the results suggest that increasing the number of diffusion timesteps improves denoising performance. Meanwhile, reducing the number of sampling timesteps ($S < T$) often yields better results. In contrast, learned perceptual image patch similarity (LPIPS) scores show that fewer sampling timesteps lead to worse perceptual quality. This highlights a trade-off between content fidelity (measured by PSNR and SSIM) and perceptual realism (measured by LPIPS). Iterative refinement enhances fine-grained details and promotes the recovery of natural textures. Logically, more sampling timesteps allow greater refinement, producing highly realistic outputs. However, results may diverge slightly from the ground truth in terms of pixel-wise similarity, resulting in lower PSNR and SSIM.

Moreover, results further indicate that controlled stochasticity in the forward process is beneficial. Setting $\gamma = 0.0$ leads to poor results, indicating that some variance is necessary. Conversely, $\gamma = 3.0$ and $\gamma = 9.0$ achieve significantly superior performance. Particularly, $\gamma = 9.0$ outperforms $\gamma = 3.0$ for configurations with few diffusion timesteps, but its relative performance gains diminish as $T$

Table 4: Denoising performance comparison of several RDIM configurations on BPAE confocal images from the benchmark dataset FMD. It exhibits the impact of the constant hyperparameter $\gamma$ that controls the variance in the forward process and the impact of the number of diffusion timesteps during training ($T$) and inference ($S$). The constant hyperparameter that controls the variance in the reverse process is fixed to $\eta = 1.0$. Orange color rows highlight ResShift scenarios, corresponding to cases where RDIM reduces to ResShift ($\eta = 1.0$ and $S = T$).

| | | FMD-Confocal-BPAE-Raw | | | | | | | | |
|---|---|---|---|---|---|---|---|---|---|---|
| $T$ | $S$ | $\gamma = 0.0$ | | | $\gamma = 3.0$ | | | $\gamma = 9.0$ | | |
| | | PSNR↑ | SSIM↑ | LPIPS↓ | PSNR↑ | SSIM↑ | LPIPS↓ | PSNR↑ | SSIM↑ | LPIPS↓ |
| 10 | 1 | 38.3703 | 0.9575 | 0.0296 | 40.0998 | 0.9686 | 0.0196 | 40.1100 | 0.9681 | 0.0202 |
| | 10 | 38.3487 | 0.9572 | 0.0299 | 39.3632 | 0.9644 | 0.0187 | 39.9524 | 0.9674 | 0.0202 |
| 50 | 1 | 38.4181 | 0.9578 | 0.0293 | 42.5354 | 0.9803 | 0.0093 | 43.1161 | 0.9821 | 0.0079 |
| | 10 | 38.4162 | 0.9578 | 0.0295 | 42.4566 | 0.9802 | 0.0079 | 43.2029 | 0.9824 | 0.0077 |
| | 50 | 38.3198 | 0.9564 | 0.0299 | 42.0566 | 0.9785 | 0.0071 | 43.1970 | 0.9824 | 0.0074 |
| 100 | 1 | 38.3758 | 0.9575 | 0.0295 | 43.9872 | 0.9851 | 0.0056 | 43.3040 | 0.9828 | 0.0075 |
| | 10 | 38.3752 | 0.9575 | 0.0295 | **44.1468** | **0.9855** | 0.0047 | 43.3743 | 0.9830 | 0.0073 |
| | 100 | 38.1775 | 0.9548 | 0.0298 | 43.5990 | 0.9837 | **0.0042** | 43.2484 | 0.9826 | 0.0068 |

increases, whereas $\gamma = 3.0$ continues to improve with longer diffusion chains, ultimately surpassing $\gamma = 9.0$ for larger $T$. These observations underline the importance of carefully balancing variance.

Figure 10 showcases PSNR, SSIM, and LPIPS scores for different numbers of sampling timesteps, using the configuration with $T = 100$ and $\gamma = 3.0$, which obtained the best results in denoising BPAE confocal images from the FMD dataset. The number of sampling timesteps evaluated includes a single-step prediction and then ranges from 10 to 100 in increments of 10. It follows that PSNR and SSIM performances peak around $S = 10$, while LPIPS achieves the best scores between $S = 90$ and $S = 100$, showing that more sampling timesteps result in higher perceptual quality but reduced reconstruction fidelity.

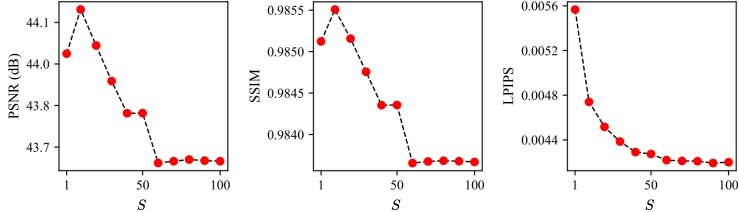

Figure 10: PSNR, SSIM, and LPIPS performance as a function of $S$ (number of sampling timesteps) on denoising of BPAE confocal images from the FMD dataset. RDIM is trained with $T = 100$ and $\gamma = 3.0$. The constant hyperparameter that controls the variance in the reverse process is fixed to $\eta = 1.0$. The number of sampling timesteps evaluated includes a single-step prediction and then ranges from 10 to 100 in increments of 10.

Table 5 demonstrates the effect of varying the constant hyperparameter $\eta$, which controls the variance in the reverse process. Looking at Table 5, the parameter $\eta$ manifests marginal impact on denoising of BPAE confocal images from the FMD dataset. Additionally, when $\gamma = 0$, the parameter $\eta$ does not affect performance, as $\eta$ is absent in the reparameterized form of $p_\theta(\boldsymbol{x_{t-1}}|\boldsymbol{x_t}, \boldsymbol{y_0})_{\gamma=0}$ (see Equations (11) and (12)).

Table 5: Impact of the constant hyperparameter $\eta$, which controls the variance in the reverse process, on denoising BPAE confocal images from the benchmark dataset FMD.

| | | | FMD-Confocal-BPAE-Raw | | | | | | | | |
|---|---|---|---|---|---|---|---|---|---|---|---|
| $T$ | $S$ | $\eta$ | $\gamma = 0.0$ | | | $\gamma = 3.0$ | | | $\gamma = 9.0$ | | |
| | | | PSNR↑ | SSIM↑ | LPIPS↓ | PSNR↑ | SSIM↑ | LPIPS↓ | PSNR↑ | SSIM↑ | LPIPS↓ |
| | | 0.0 | 38.3752 | 0.9575 | 0.0295 | 44.1429 | 0.9855 | **0.0047** | 43.3732 | 0.9830 | 0.0073 |
| 100 | 10 | 0.5 | 38.3752 | 0.9575 | 0.0295 | 44.1440 | 0.9855 | **0.0047** | 43.3738 | 0.9830 | 0.0073 |
| | | 1.0 | 38.3752 | 0.9575 | 0.0295 | **44.1468** | **0.9855** | **0.0047** | 43.3743 | 0.9830 | 0.0073 |

## C.8 COMPARING THE PERCEPTION-DISTORTION TRADE-OFF AGAINST RECENT WORK

To provide a more complete assessment of reconstruction quality, we follow the framework established in Blau & Michaeli (2018). Traditional distortion metrics such as PSNR strongly penalize any deviation from the exact ground truth, often driving models toward overly smooth or conservative solutions. In contrast, perceptual metrics such as LPIPS capture human-aligned similarity in deep feature space and reward reconstructions that preserve realistic texture and structure, even at the cost of introducing plausible high-frequency hallucinations. While such hallucinated details can be undesirable in certain reconstruction domains (e.g., in medical imaging), they offer a useful lens for quantifying perceptual fidelity. Since different diffusion-based frameworks are optimized with varying objectives, plotting PSNR against LPIPS provides a principled way to visualize and measure their position along the perception–distortion trade-off.

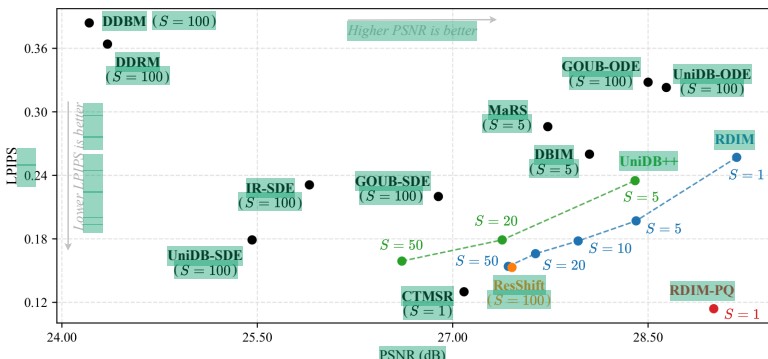

Figure 11: Comparison of the perception-distortion trade-off between the proposed RDIM and the state-of-the-art on the DIV2K dataset for $4\times$ SR.

Figure 11 compares the proposed method against existing techniques. For RDIM and UniDB++, we exploit the methods native fast-sampling capabilities to extract multiple operating points along the perception–distortion curve. The resulting comparison shows that the proposed RDIM achieves a substantially improved perception–distortion profile against all other methods. In particular, when comparing with the most recent UniDB++ framework, which relies on a diffusion bridge-based on Doob's $h$-transform (Pan et al., 2025), RDIM delivers significant PSNR gains for similar LPIPS values.

Moreover, CTMSR achieves a competitive LPIPS score as it explicitly optimizes this perceptual metric, effectively trading distortion for improved visual quality. Naturally, this comes at the expense of reduced PSNR. In contrast, RDIM-PQ-1 attains even lower perceptual scores while simultaneously yielding substantially lower distortion. This property is particularly advantageous for applications requiring high fidelity to the original signal, such as medical imaging, scientific microscopy, satellite and aerial sensing, and downstream vision tasks where hallucinated details could compromise reliability.

## C.9 QUANTITATIVE RESULTS IN ADDITIONAL IMAGE RESTORATION TASKS

In addition to the qualitative examples in Figure 4, quantitative results for inpainting, colorization, and deblurring are reported in Table 6. Across all three tasks, RDIM maintains the same trends observed in denoising and SR. RDIM-1 consistently achieves the highest PSNR and SSIM, while RDIM-10 provides competitive performance with slightly better perceptual quality (LPIPS). RDIM surpasses ResShift in every metric except a few LPIPS cases, indicating that RDIM produces sharper and more faithful reconstructions even in challenging restoration settings.

Table 6: Performance on the FFHQ dataset for image inpainting, colorization, and deblurring.

| Method | FFHQ-Inpainting | | | FFHQ-Colorization | | | FFHQ-Deblurring | | |
|---|---|---|---|---|---|---|---|---|---|
| | PSNR↑ | SSIM↑ | LPIPS↓ | PSNR↑ | SSIM↑ | LPIPS↓ | PSNR↑ | SSIM↑ | LPIPS↓ |
| Corrupted | 9.034 | 0.136 | 1.165 | 20.806[†] | 0.926[†] | 0.223[†] | 24.691 | 0.686 | 0.486 |
| ResShift | 31.721 | 0.922 | **0.022** | 25.320 | 0.948 | 0.103 | 28.331 | 0.812 | **0.089** |
| RDIM-1 | **33.514** | **0.941** | 0.029 | **25.727** | **0.950** | **0.094** | **29.995** | **0.847** | 0.170 |
| RDIM-10 | 32.330 | 0.931 | 0.025 | 25.565 | 0.949 | 0.099 | 28.870 | 0.826 | 0.111 |

[†] To compute colorization scores on corrupted images, the existing channel is replicated.

Notably, these experiments use a forward diffusion process with $T = 50$ steps, which is half the length employed in the denoising and SR experiments. As a result, performances reported in Table 6 still have margin for improvement when adopting longer diffusion chain lengths (see Appendix C.7).

## C.10 PERFORMANCE WHEN THE FORWARD MODEL IS MISMATCHED

To evaluate model robustness when the forward model is mismatched (i.e., when the testing data contain unseen degradations), signal-dependent Poisson noise is simulated and applied to the LR images of the DIV2K-Bicubic-$\times 4$ validation set. A fixed peak photon count of $\lambda_{peak} = 1.0 \times 10^3$ is used. This corresponds to a moderate imaging scenario (e.g., indoor lighting or mid-ISO conditions). Notably, the models were trained on the standard DIV2K-Bicubic-$\times 4$ dataset without incorporating Poisson noise.

Table 7: Evaluation on Poisson-corrupted DIV2K-Bicubic-$\times 4$ images after training on the original dataset without adding Poisson noise. Values in parentheses indicate the performance drop relative to the evaluation on the original noise-free DIV2K-Bicubic-$\times 4$ dataset (see Table 2). Red color indicates the worst performance drop overall and Green color the best.

| Method | Poisson-corrupted DIV2K-Bicubic-$\times 4$ | | |
|---|---|---|---|
| | PSNR↑ | SSIM↑ | LPIPS↓ |
| ResShift | 23.610 (-3.845) | 0.521 (-0.259) | 0.428 (+0.275) |
| GOUB-SDE | 20.165 (-6.725) | 0.335 (-0.413) | 0.664 (+0.444) |
| UniDB-SDE | 19.284 (-6.176) | 0.314 (-0.372) | 0.697 (+0.518) |
| CTMSR-1 | 22.987 (-4.100) | 0.466 (-0.293) | 0.495 (+0.365) |
| RDIM-1 | 26.260 (-2.920) | 0.673 (-0.151) | 0.519 (+0.262) |
| RDIM-10 | 24.363 (-3.600) | 0.564 (-0.231) | 0.415 (+0.237) |

As expected, performance decreases for all methods when evaluated on unseen Poisson-corrupted images, reflecting the sensitivity of supervised reconstruction to mismatched forward degradations. Particularly, GOUB and UniDB suffer the largest drops in PSNR, SSIM, and LPIPS. This can be attributed to their bridge formulation, which relies on fixed endpoint distributions and tightly couples the reconstruction process to the forward degradation operator. When this endpoint shifts due to unseen Poisson noise, the learned bridge becomes misaligned, causing the reverse dynamics to deviate from the correct posterior and leading to substantially larger reconstruction errors. Meanwhile, CTMSR-1 is comparatively less affected, but its LPIPS score still deteriorates substantially, which is particularly striking given that it directly optimizes for perceptual quality. This underscores the difficulty of maintaining perceptual fidelity under unseen Poisson noise.

In contrast, RDIM demonstrates superior robustness with RDIM-1 exhibiting the smallest drop in PSNR ($-2.920$ dB) and SSIM ($-0.151$). Moreover, RDIM-10 obtains the least increase in LPIPS

(+0.237), followed by RDIM-1 comparatively modest increase (+0.262). Unlike bridge-based methods with fixed endpoint constraints, RDIM does not rely on a strictly specified degradation endpoint, and its controllable variance forward process allows the last latent variable in the forward process, $x_T$, to remain near the corrupted observation, $y_0$ (i.e., the LQ image), without being tied to it. This flexibility helps the model remain consistent even when the degradation shifts. In addition, the few-step reconstruction enabled by the implicit sampling strategy reduces error accumulation, which is particularly beneficial under forward model mismatch. Overall, these results indicate that RDIM not only achieves state-of-the-art performance under matched conditions but also retains faithful reconstructions under moderate deviations from the training degradation model, highlighting its practical robustness for real-world image restoration scenarios.

## C.11 ADDITIONAL QUALITATIVE RESULTS

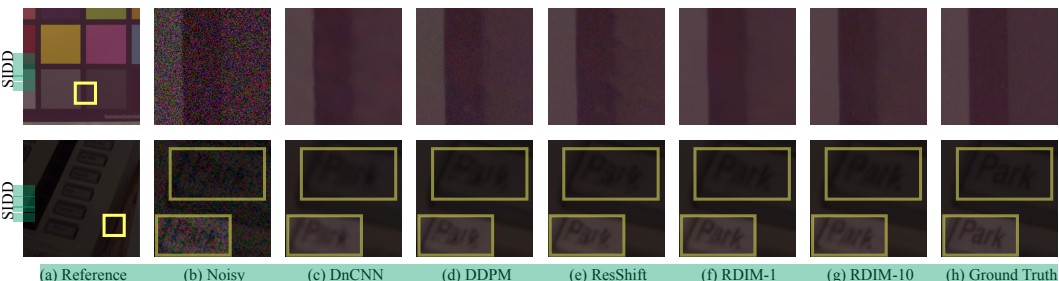

Figure 12: Qualitative denoising analysis on cropped regions from the SIDD dataset. Since SIDD contains noisy images captured under challenging lighting conditions, brightness-adjusted crops of the marked regions are shown in the bottom row for enhanced visualization.

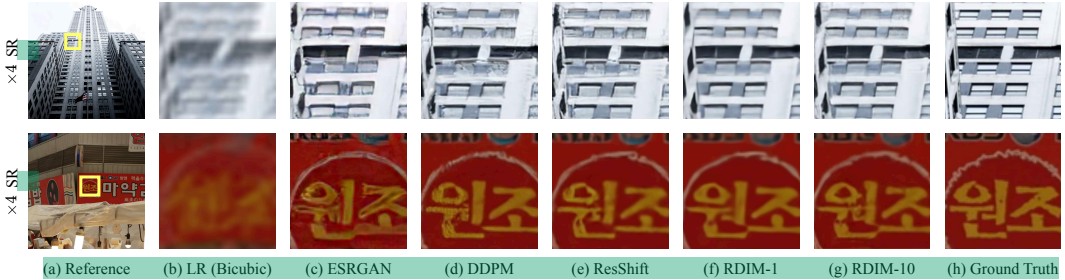

Figure 13: SR qualitative comparison on cropped regions from the DIV2K subset with unknown degradation.

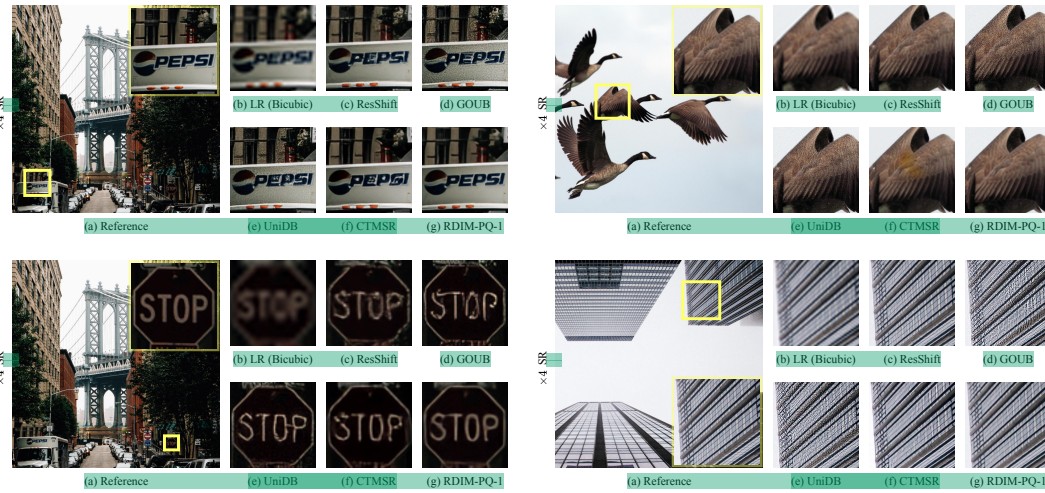

Figure 14: SR qualitative comparison on cropped regions from the DIV2K subset with bicubic downsampled images.

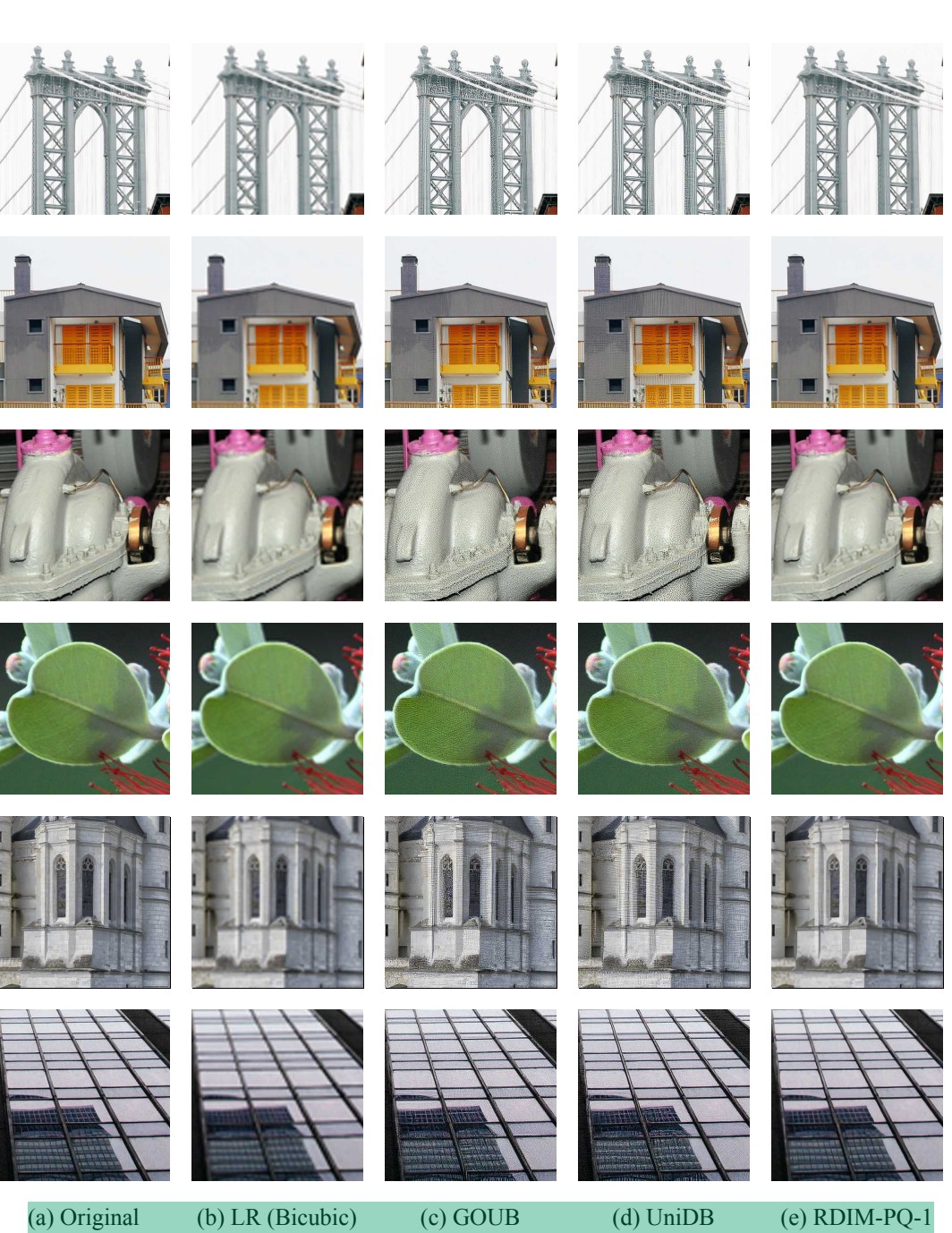

(a) Original          (b) LR (Bicubic)          (c) GOUB          (d) UniDB          (e) RDIM-PQ-1

Figure 15: Qualitative comparison of RDIM-PQ-1 on ×4 SR. Cropped regions from the DIV2K subset, with bicubic downsampled images, suggest RDIM achieves greater structural and texture fidelity than bridge-based models.

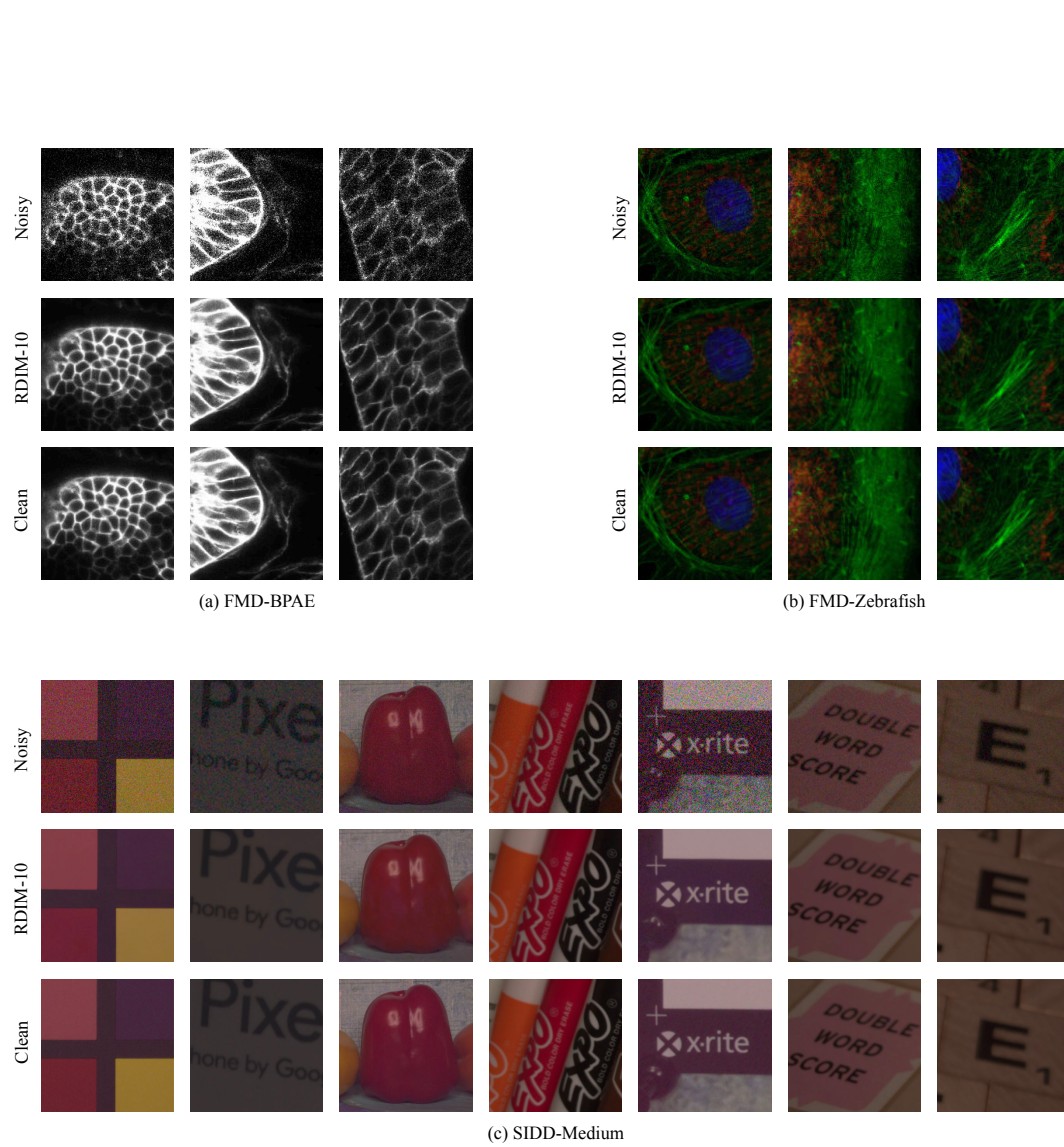

(a) FMD-BPAE

(b) FMD-Zebrafish

(c) SIDD-Medium

Figure 16: Denoising results of RDIM-10 on images from the FMD and SIDD datasets. For improved visualization, only cropped regions are shown. RDIM is trained with $T = 100$ and $\gamma = 3.0$. Inference is conducted with $S = 10$ and $\eta = 1.0$.

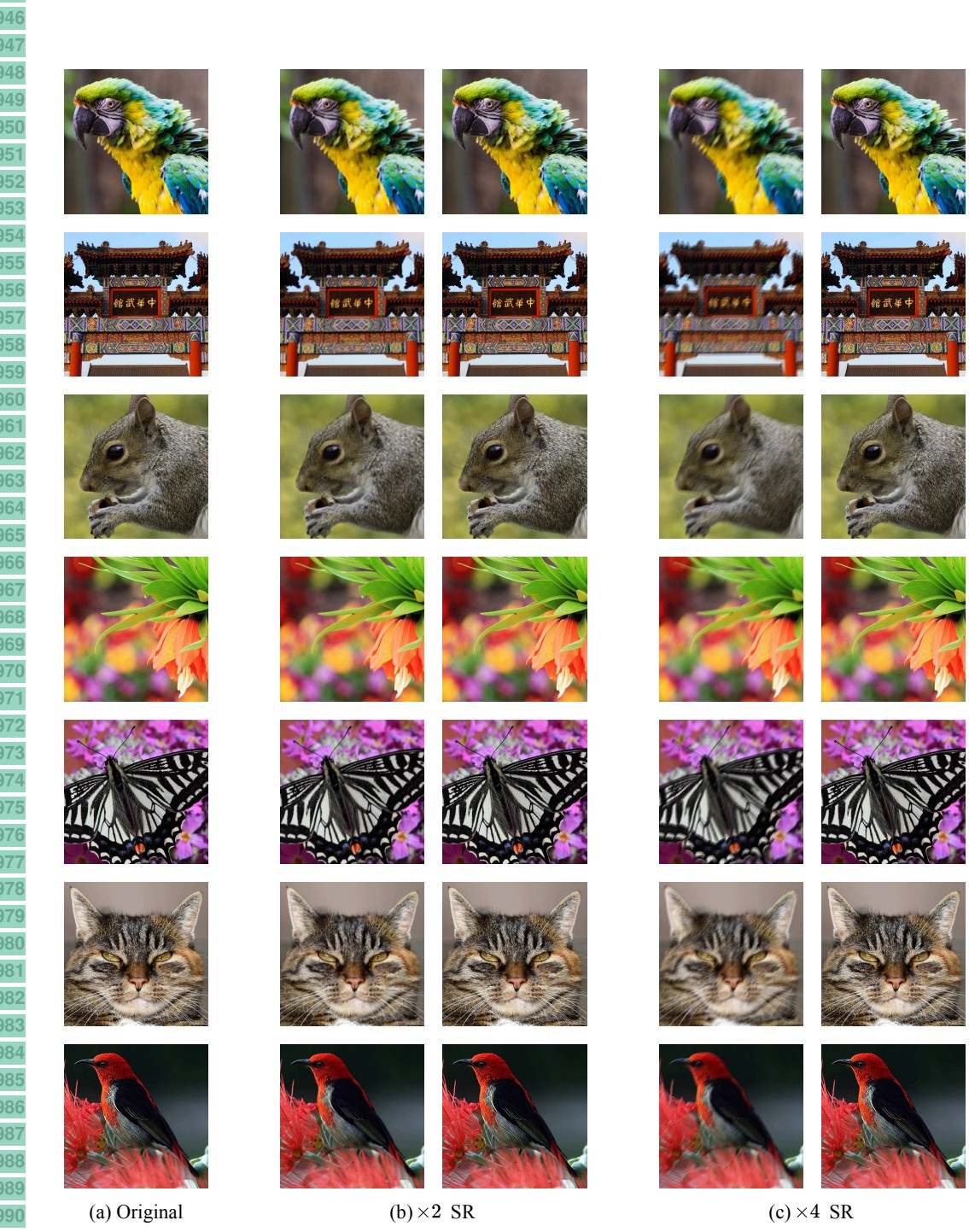

(a) Original           (b) ×2 SR           (c) ×4 SR

Figure 17: ×2 and ×4 SR results of RDIM-10 on images from the DIV2K dataset under unknown degradations. RDIM is trained with $T = 100$ and $\gamma = 3.0$. Inference is conducted with $S = 10$ and $\eta = 1.0$. In (b) and (c), the left side represents the input image and the right side the output.

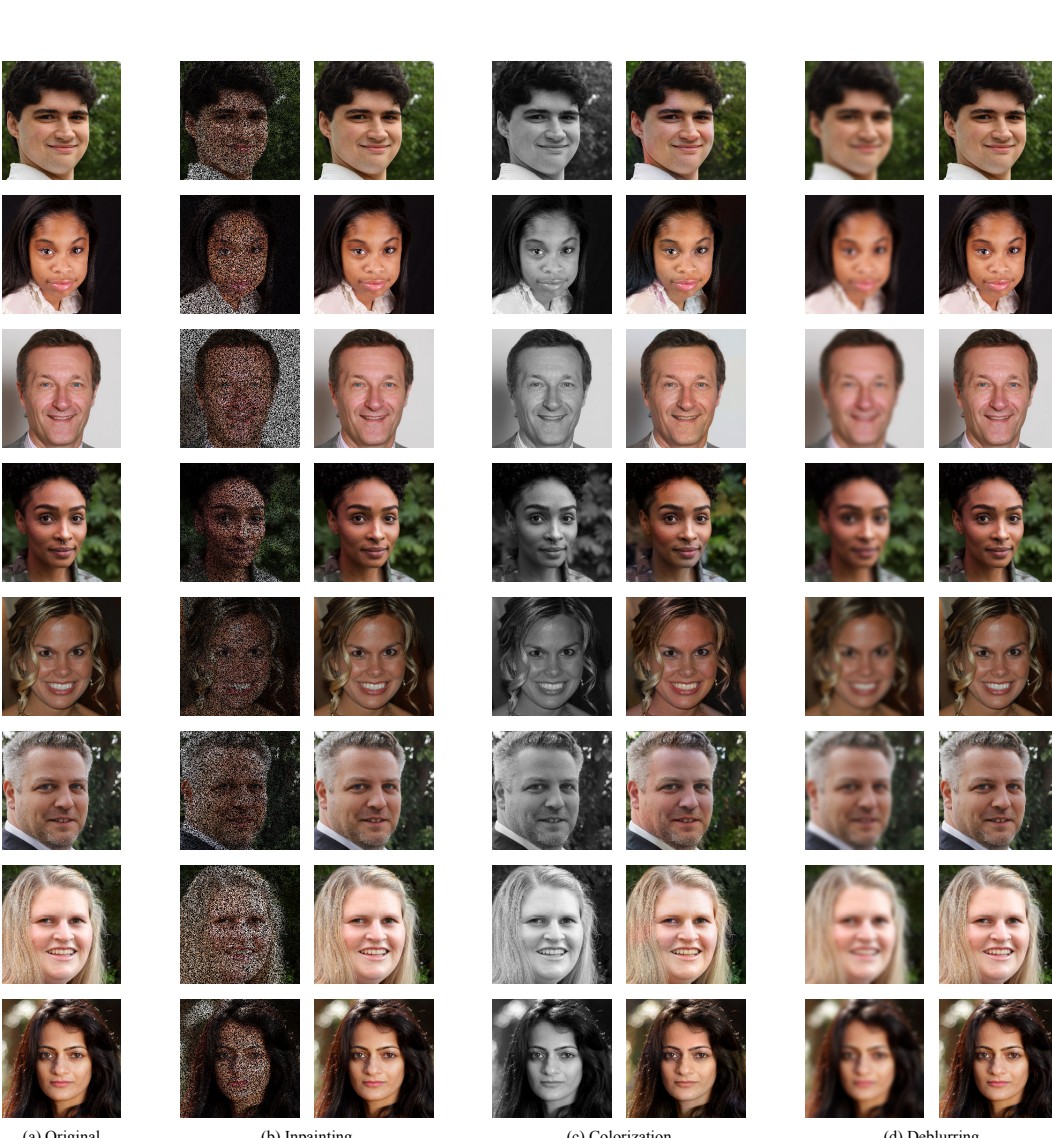

(a) Original      (b) Inpainting      (c) Colorization      (d) Deblurring

Figure 18: Image inpainting, colorization, and deblurring results of RDIM-10 on images from the FFHQ dataset. For inpainting, pixels in the original images are randomly masked and set to zero with probability $p_{\text{mask}} = 0.5$. For colorization, grayscale inputs are obtained by converting the original RGB images to luminance. For deblurring, synthetic blurred images are generated from ground truth images by applying a Gaussian blur with kernel size $15 \times 15$ and standard deviation $\sigma = 3.0$. RDIM is trained with $T = 50$ and $\gamma = 3.0$. Inference is conducted with $S = 10$ and $\eta = 1.0$. In (b), (c) and (d), the left side represents the input image and the right side the output.

