# OpenReview forum: "Residual Diffusion Implicit Models"
_ICLR.cc/2026/Conference — Submitted to ICLR 2026_

### Official Review · Reviewer_hRiF · 2025-10-19

**Soundness:** 2
**Presentation:** 2
**Contribution:** 2
**Rating:** 2
**Confidence:** 5

**Summary:**

This paper proposes *Residual Diffusion Implicit Models (RDIM)*, which aligns the *forward* process with real degradation (by removing the residual between HQ and LQ) and adopts a DDIM-style non-Markovian long-range skip sampling for the *reverse* process. Formally, the method strictly subsumes ResShift as a special case, making RDIM a generalization of it. Experiments report that RDIM-1 (1 step) and RDIM-10 (10 steps) outperform or are comparable to ResShift/DDPM while being faster.

**Strengths:**

1. The paper proves that ResShift is a special case of RDIM, enabling existing ResShift models to be directly used for RDIM sampling without retraining.
2. The forward process is aligned with degradation and includes a controllable variance mechanism.
3. Experiments demonstrate strong performance with significant acceleration.

**Weaknesses:**

**Lack of novelty:**

1. One of the authors’ contributions is generalizing ResShift, but from today’s perspective, RDIM is already outdated. The authors lack an adequate survey of recent works.
   The core formulation of the paper is: $q(x_t|x_0,\Delta)=\mathcal{N}(x_0 - \beta_t \Delta, \gamma^2 \beta_t I)$. However, expanding it gives: $q(x_t|x_0,\Delta)=\mathcal{N}((1-\beta_t)x_0+\beta_t y_0, \gamma^2 \beta_t I)$, which can also be written as: $q(x_t|x_0,y_0)=\mathcal{N}((1-\beta_t)x_0+\beta_t y_0, \gamma^2 \beta_t I)$. This is clearly a simple case of stochastic interpolation [1] (diffusion bridge) or essentially time varying Ornstein–Uhlenbeck (OU) process. The sampling formulas and training losses mentioned later are straightforward corollaries under this framework. The acceleration formula that follows is also a rather trivial conclusion of DDIM.

**Lack of baseline comparisons:**

2. Based on the above, the authors did not compare with several advanced diffusion-bridge-type models. For example, **GOUB[3] (ICML 2024)** achieves 28.5 PSNR on DIV2K ×4, and **UniDB [7](ICML 2025)** reaches 28.64 — both higher than RDIM, and belonging to the same family of methods. Other models such as **BBDM**[4], **I2SB**[5], , **DDBM**[6] and more advanced methods, should also be discussed in the *Related Work* section.

**Writing:**

3. The authors first introduce the goal as $q(x_T |x_0,\Delta) = q(x_T |y_0)$, but immediately follow it with the proof in Appendix 1. At that point, the forward process has not yet been defined, but the appendix proof directly uses the later-defined forward process. This creates a logical inconsistency and can confuse readers when reading sequentially.

**Overall:**
From today’s perspective, RDIM lacks innovation. I have some suggestions:

1. Conduct a deeper survey of diffusion-bridge-related works (2024–2025). The authors need to distinguish RDIM from existing models and establish a more advanced motivation.
2. Compare RDIM with more diffusion-bridge models through deeper experiments to validate its theoretical advantages.

# Reference:

[1] Michael S. Albergo, et al. "Stochastic Interpolants: A Unifying Framework for Flows and Diffusions." (arXiv 2023)

[2] Luo, Ziwei, et al. "Image restoration with mean-reverting stochastic differential equations." (ICML 2023).

[3] Yue, Conghan, et al. "Image restoration through generalized ornstein-uhlenbeck bridge." (ICML 2024).

[4] Li, Bo, et al. "Bbdm: Image-to-image translation with brownian bridge diffusion models." (CVPR 2023).

[5] Liu, Guan-Horng, et al. "I $^ 2$ SB: Image-to-Image Schr\" odinger Bridge." (ICML 2023).

[6] Zhou, Linqi, et al. "Denoising diffusion bridge models." (ICLR 2024).

[7] Zhu, Kaizhen, et al. "UniDB: A Unified Diffusion Bridge Framework via Stochastic Optimal Control." (ICML 2025).

**Questions:**

See weaknesses.

---

> ### Author Response · Authors · 2025-11-21
>
> We sincerely thank the reviewer for the detailed and thoughtful feedback. We have carefully revised the manuscript to address the points raised regarding novelty, related work, experimental coverage, and presentation. We address your comments below.
>
> **Novelty and relation to diffusion-bridge literature.** We agree that RDIM is connected to recent diffusion-bridge and stochastic-interpolant formulations. We have substantially rewritten the “Related Work” section (Sec. 4) to give a clearer and more comprehensive survey of bridge models (SB/DB/Brownian bridges/OU bridges), including all works mentioned by the reviewer (your ref [1-7]). RDIM differs from standard bridge models in crucial aspects. Namely, most bridge methods (e.g., Schrödinger bridges, generalized OU bridges, I2SB, BBDM) align distribution marginals, but do not preserve correspondences between individual samples at the two endpoints (as explicitly noted in SB literature and in Kieu et al., 2025 - revised manuscript). In inverse problems such as denoising and super-resolution, maintaining a stable mapping between the LQ observation and the HQ reconstruction is essential. RDIM is explicitly designed to preserve these correspondences by modeling the residual $x_0-y_0$ and keeping the diffusion trajectory anchored to the actual degradation model. Furthermore, RDIM operates directly on the residual domain, which tightly links the diffusion evolution to the real degradation operator. In contrast, bridge methods typically operate in ambient space and enforce endpoint constraints without exploiting residual structure. Finally, RDIM derives a closed-form implicit sampler that preserves the forward marginals, enabling effective single-step or few-step reconstruction. While recent works derive diffusion-bridge formulations supporting accelerated sampling (e.g., UniDB++), they lack a mechanism to modulate stochasticity at the endpoints during training. Controlling stochasticity improves both generalization and robustness (Appendix C.7). We clarified these distinctions in the revised manuscript to better highlight the motivation and contribution of RDIM beyond existing bridge-based approaches.
>
> **Expanded comparisons with diffusion-bridge baselines.** Following the reviewer’s recommendation, we significantly expanded the experimental section to include the most relevant bridge and consistency models, as well as fast samplers (CTMSR, MaRS, DDBM, DBIM, GOUB, UniDB, UniDB++) and classical baselines (DDRM, IR-SDE). All newly obtained results are included in Table 2. Despite the strength of recent bridge-based approaches, RDIM-1 achieves the highest PSNR/SSIM across all baselines. Its performance is only surpassed by CTMSR and only on LPIPS, as it directly optimises for this metric. Nonetheless, the RDIM-PQ (variant optimizing perceptual quality) surpasses CTMSR on LPIPS, attaining only a slight degradation on PSNR and SSIM vs the standard proposed RDIM, but still far better than all the state of the art. Qualitative examples also show that RDIM produces sharper and more faithful reconstructions (Figs. 3,11,14,15), especially on fine textures and detailed structures, where competing methods exhibit noticeable blurring, deformations and hallucinations. Notably, while GOUB and UniDB are strong performers on DIV2K ×4 SR (as the reviewer noted), RDIM exceeds them in both fidelity (Table 2) and perception-distortion trade-off (Appendix C.8), while offering significantly faster sampling. For fairness, we note that GOUB and UniDB are evaluated on bicubic degradation, while in Table 1 we use unknown, which is a harder problem. Hence, Table 2 is evaluated on bicubic degradation.
>
> **Clarifying the contribution: beyond a simple generalization of ResShift.** We recognize that RDIM subsumes ResShift as a special case, and we view this as a strength: existing ResShift models can be reused for RDIM without retraining. However, RDIM is not a simple combination of ResShift and DDIM. Its contributions include: (i) residual-aligned diffusion forward process, preserving sample-level correspondences across HQ/LQ domains; (ii) a non-Markovian reverse process derived to preserve the forward marginals, enabling theoretically grounded skip sampling and stable few-step reconstruction; (iii) controllable stochasticity $(\gamma,\eta)$ to interpolate between deterministic and stochastic reconstructions. These components collectively enable RDIM to achieve high reconstruction fidelity with very few steps, which neither ResShift nor existing diffusion-bridge methods achieve reliably.
>
> **Addressing the writing and logical presentation issue.** We thank the reviewer for pointing out the logical inconsistency between the main text and Appendix A. The revised manuscript now introduces the forward process first, and only after referencing the proof related to the forward process cumulative transition distribution.

---

> > ### Author Response · Authors · 2025-11-25
> >
> > At the request of one of the reviewers, we have updated the paper by highlighting changes using pdfdiff, and also took the opportunity to include the missing results regarding CTMSR.

---

### Official Review · Reviewer_2RLd · 2025-10-28

**Soundness:** 3
**Presentation:** 3
**Contribution:** 2
**Rating:** 4
**Confidence:** 4

**Summary:**

In this work, authors propose residual diffusion impicit models (RDIMs), a generalized framework to model the residuals between high-quality and low-quality images. In particular, the forward process of the model is aligned with the degradation model, which enables reconstructions that are faster and more accurate. Inspired by DDIMs, during the reverse process, some of the intermediate timesteps are skipped to accelerate the reconstruction. Akin to DDIM sampling, authors introduce a controllable variance mechanism to control the stochasticity of the sampling. Authors demonstrate the effectiveness of RDIMs on denoising and super-resolution benchmarks and further qualitative evidence on image inpainting, colorization, and deblurring tasks.

**Strengths:**

* The paper is written well and is easy to read.
* The ablation studies are conducted extensively, solidying the design choices.
* The authors demonstrate their method on various degradation models (SR, denoising, deblurring, colorization, inpainting, etc.)
* In terms of reproducibility, authors did an excellent job in disclosing dataset and implementation details.

**Weaknesses:**

* The work is not contextualized well. There are several important works that investigate aligning the forward process of the model with degradation (see [1-4] below, note that [3] is cited but not discussed). These should be discussed and compared with RDIM. Whenever appropriate, it would be good to include some of these as part of baseline results. The only competitive baseline method is ResShift which shares too many similarities with RDIM already.
* On a related note, I find the technical contributions of the paper to be limited. In its current version, it reads as a simple extension of ResShift combining it with (essence of) DDIM sampling.
* The central claim that RDIM matching or exceeding ResShift performance while requiring fewer samples is due to distortion metrics (PSNR and SSIM). Due to perception-distortion trade-off [5] (also noted by authors in Appendix C.7), it is possible that in terms of perceptual quality, ResShift (or other models that perform many steps) can outperform few/single-step solvers (also observed in [1,2] and also ResShift). If this is the case, limitations/benefits of RDIM against ResShift should be disclosed and discussed better.
* Please see questions below as well.

**Questions:**

***Questions and Suggestions:***
* In terms of evaluation metrics, providing perceptual quality scores such as LPIPS and FID would be good to complement the distortion metrics (PSNR/SSIM) and provide a more comprehensive view. As authors also comment on this in Appendix C.7, perception-distortion trade-off is a well known phenomenon [5].
* As mentioned in the weaknesses section, could the authors comment on similarities and differences between RDIM and existing work [1,2,3,4]? I would like to note that for RDDM [3], the authors cite this work saying "... requiring to traverse all diffusion timesteps" but their conclusion mentions that "RDDM achieves SOTA performance in no more than five sampling steps".
* Besides comparing with ResShift, which can be considered as an ablation study due to RDIM generalizing this work, I think comparison with more competitive baselines (which are dedicated diffusion based solver) would help clarify the strengths/weaknesses of the method. Some of the good candidates are DPS (Diffusion Posterior Sampling), CCDF (Come-Close-Diffuse-Faster), DDRM, Resample [6], etc.
* In the introduction, authors cite DDPM as a powerful class of models for image reconstruction (line 36-37). I think this is slightly misleading. Such models were introduced as a means for generative modeling/image synthesis. Although it is natural to consider them as useful priors, it took non-trivial effort in the community to actually use such priors and guide the generation process tailored towards arbitrary (linear/non-linear) reconstruction problems. For readers not familiar with the literature, I would recommend rephrasing this section for clarity.

***
***References:***

[1] Delbracio, Mauricio, and Peyman Milanfar. "Inversion by direct iteration: An alternative to denoising diffusion for image restoration." arXiv preprint arXiv:2303.11435 (2023).

[2] Fabian, Zalan, Berk Tinaz, and Mahdi Soltanolkotabi. "Diracdiffusion: Denoising and incremental reconstruction with assured data-consistency." Proceedings of machine learning research 235 (2024): 12754.

[3] Liu, Jiawei, et al. "Residual denoising diffusion models." Proceedings of the IEEE/CVF Conference on Computer Vision and Pattern Recognition. 2024.

[4] Heitz, Eric, Laurent Belcour, and Thomas Chambon. "Iterative α-(de) blending: A minimalist deterministic diffusion model." ACM SIGGRAPH 2023 Conference Proceedings. 2023.

[5] Blau, Yochai, and Tomer Michaeli. "The perception-distortion tradeoff." Proceedings of the IEEE conference on computer vision and pattern recognition. 2018.

[6] Song, Bowen, et al. "Solving inverse problems with latent diffusion models via hard data consistency." arXiv preprint arXiv:2307.08123 (2023).

---

> ### Author Response · Authors · 2025-11-21
>
> We sincerely thank the reviewer for the constructive and detailed feedback. We have revised the manuscript to improve contextualization, clarify contributions, and expand the empirical evaluation. Below, we address all your concerns.
>
> **Contextualization and comparison to prior work.** We significantly expanded the “Related Work” section (Sec. 4) to discuss the methods highlighted by the reviewer and to contextualize the proposed RDIM model within the most recent literature. RDIM shares the core idea of aligning the forward process with degradation, but differs in two key points: (i) residual-based diffusion formulation, and (ii) non-markovian reverse process with marginal preservation. More specifically, RDIM explicitly models residuals between HQ and LQ images, aligning the diffusion trajectory with the ground-truth degradation while preserving sample-level correspondences. This contrasts with direct inversion schemes [your ref #1], deterministic blending [4], or incremental reconstruction [2], which do not exploit residual diffusion modeling. Further, RDIM derives an implicit DDIM-style reverse process that preserves the forward marginal distribution, enabling few- or single-step reconstructions while maintaining fidelity. This differs from RDDM [3] and related approaches, which rely on multi-step denoising; although RDDM does accelerate sampling, it still performs several diffusion steps and does not model residuals in a marginal-preserving way.
>
> **Expanded and more competitive experimental comparisons.** While direct experimental comparisons with all the suggested methods would be ideal, for some methods, the source code is either not publicly available or not directly compatible with our datasets and tasks. Nonetheless, to provide meaningful baselines, we now compare RDIM against 9 new models (CTMSR, MaRS, DDBM, DBIM, GOUB, UniDB, UniDB++, DDRM, IR-SDE). The experimental results (Table 2, Fig. 3) show that RDIM-1 consistently achieves the highest PSNR/SSIM, outperforming all competing methods, highlighting its extreme efficiency. Moreover, the method Inversion by Direct Iteration (Delbracio & Milanfar, 2023), suggested by the reviewer, was not used for comparison in the paper because we couldn’t access the type of degradation reported (unknown or bicubic). However, on DIV2K ×4 SR, (Delbracio & Milanfar, 2023) report a PSNR of 26.45, while RDIM achieves 28.28 PSNR and 29.18 PSNR for unknown and bicubic degradation, respectively.
>
> **Perception-distortion trade-off and perceptual metrics (LPIPS).** Following the reviewer’s suggestion, we added a perception-distortion analysis (Appendix C.8), including LPIPS vs PSNR curves (Fig. 11) for RDIM and all competitive models. Results show that: (i) RDIM achieves a superior perception–distortion profile overall; (ii) multi-step models (e.g., ResShift) may obtain lower LPIPS due to heavier texture hallucination, but this decreases structural fidelity (a behavior well-documented in [5]); (iii) RDIM requires significantly fewer steps (S=50) to match the perceptual quality of ResShift at S=100, offering a significant acceleration for a similar perception-distortion operating point. We initially chose PSNR/SSIM as primary metrics because hallucination-free fidelity is critical in inverse problems (e.g., microscopy, medical imaging). Nonetheless, LPIPS-based comparisons are now included for completeness. Furthermore, since RDIM operates in image space, integrating a perceptual loss is straightforward. Thus, RDIM-PQ was trained with a perceptual quality (PQ) objective, which attained the best LPIPS score against SoA (see Table 2, Fig. 14 and 15).
>
> **Clarifying the contribution beyond ResShift.** While RDIM generalizes ResShift, it is not a simple combination with DDIM sampling. The essential novelties are: (i) residual-aligned diffusion forward process, preserving sample-level correspondences across HQ/LQ domains; (ii) a non-Markovian reverse process derived to preserve the forward marginals, enabling theoretically grounded skip sampling and stable few-step reconstruction; (iii) controllable stochasticity ($\gamma,\eta$) to interpolate between deterministic and stochastic reconstructions, (iv) evidence that reducing sampling steps accelerates inference and yields more faithful reconstructions by limiting hallucinations that arise in long diffusion chains. These innovations jointly enable RDIM to achieve high fidelity with 1-10 steps, a regime where ResShift and other diffusion models degrade substantially. We tried to make this more clear in the revised manuscript.
>
> **Clarification about DDPM in the introduction.** We agree that DDPMs were originally introduced for generative modeling and image synthesis, and that their application to image reconstruction and inverse problems required significant subsequent developments by the community. We have modified the paragraph in the introduction accordingly.

---

> > ### Author Response · Authors · 2025-11-25
> >
> > At the request of one of the reviewers, we have updated the paper by highlighting changes using pdfdiff, and also took the opportunity to include the missing results regarding CTMSR.

---

### Official Review · Reviewer_RD6i · 2025-11-01

**Soundness:** 2
**Presentation:** 2
**Contribution:** 2
**Rating:** 4
**Confidence:** 4

**Summary:**

This paper introduces Residual Diffusion Implicit Models (RDIMs), a diffusion framework tailored for inverse problems such as image denoising and super-resolution. Unlike standard DDPMs that start from pure Gaussian noise, RDIMs explicitly model the residuals between high-quality and low-quality images, aligning the forward process with the actual degradation. The reverse process leverages implicit sampling to skip intermediate steps, enabling few-step or even single-step reconstructions. Additionally, RDIMs incorporate a controllable variance mechanism to balance deterministic and stochastic sampling. Experiments on denoising and super-resolution benchmarks demonstrate that RDIMs outperform DDPMs and match or surpass ResShift, while reducing inference steps by up to 100×.

**Strengths:**

1. This paper aligns the forward process with real degradations, addressing the mismatch of standard diffusion models in inverse problems.
2. This paper shows that ResShift is a special case of RDIM, highlighting the generality of the framework.

**Weaknesses:**

1. Experiments mainly compare against DDPM and ResShift; missing comparisons with other recent efficient diffusion approaches (e.g., consistency models, rectified flow).
2. This paper focuses on denoising and super-resolution; validation on other inverse problems (e.g., deblurring, inpainting, compressed sensing) is limited.

**Questions:**

1. How does RDIM ensure stability and avoid artifacts when using extremely few steps (e.g., single-step reconstruction)?
2. Can pretrained ResShift models be directly reused for RDIM across different tasks without retraining, or are task-specific adjustments required?

---

> ### Author Response · Authors · 2025-11-21
>
> We sincerely thank the reviewer for the constructive comments and helpful suggestions. We revised the manuscript to improve clarity, completeness, and empirical validation. Below, we address all concerns.
>
> **Expanded comparisons with recent efficient diffusion approaches.** We have enriched our experimental section to include additional comparisons beyond DDPM and ResShift. In particular, we now compare RDIM against: (i) consistency-based models, CTMSR (ICCV’25), which directly represent the family of consistency and few-step distillation approaches; (ii) fast samplers, MaRS (ICLR’25), which provide analytic ODE/SDE solutions for mean-reverting diffusions; (iii) diffusion-bridge and accelerated restoration methods, including DDBM (ICLR’24), DBIM (ICLR’25), GOUB (ICML’24), UniDB (ICML’25), and UniDB++ (2025); (iv) other classical baselines, such as DDRM (NeurIPS’22) and IR-SDE (ICML’23). To ensure fairness across all models, we introduced new x4 SR experiments on DIV2K bicubic downsampling using 128x128 patches. Results are reported in Table 2, with visual comparisons in Fig. 3 and perception-distortion trade-off curves in Appendix C.8. Across all settings, RDIM consistently achieves the strongest fidelity (PSNR/SSIM) while requiring extremely few sampling steps. Notably, RDIM-1 (single step) outperforms all competing models in PSNR and SSIM while maintaining competitive perceptual quality. In addition, we significantly expanded the “Related Work” section (Sec. 4) to better contextualize the proposed RDIM model with respect to consistency models, rectified flows, and diffusion bridges.
>
> **Validation on other inverse problems.** Our work focuses on denoising and super-resolution; however, RDIM is already visually evaluated on inpainting, deblurring, and colorization (Fig. 4). To strengthen this part, we added quantitative results in Appendix C.9, complementing the qualitative results in Fig. 4 and Fig. 18. Nonetheless, the extended experiments demonstrate that RDIM generalizes effectively across multiple problems. Compressed sensing is not included due to scope and time constraints.
>
> **Reusing pretrained ResShift models.** ResShift models can be reused directly when the task and degradation model match, because both frameworks share the same forward marginals and training objective. RDIM simply applies a different reverse process and therefore does not require retraining. When the degradation operator differs, task-specific retraining is necessary, as expected in supervised reconstruction methods. However, to illustrate robustness under mismatch, we added a new experiment (Appendix C.10 and table AC.2 in official comments to AC): training on DIV2K x4 bicubic SR and testing on Poisson-corrupted DIV2K x4 SR.  RDIM shows the smallest PSNR drop among all compared methods (similar conclusion to SSIM and LPIPS): RDIM-1 decreases 2.92 dB, ResShift decreases 3.85 dB, GOUB-SDE decreases 6.73 dB, and UniDB-SDE decreases 6.18 dB. These results demonstrate that RDIM is more resilient to degradation mismatch than competing approaches. In fact, RDIM-1 PSNR absolute value (26.26 dB) with model mismatched is still better than several related works without mismatch (no added noise), e.g., DDRM, IR-SDE, DDBM (see table AC.1/official comment to AC).
>
> **Stability and artifact avoidance in extremely few-step sampling.** RDIM achieves stable behavior in the few-step regime because the model predicts $x_0$ directly, in contrast to DDPM-style noise prediction. This reduces cumulative error propagation and limits the hallucinations often introduced by long diffusion chains. Moreover, RDIM matches or surpasses ResShift while using significantly fewer sampling timesteps. In contrast, DDIM applied to a pretrained DDPM degrades noticeably when reducing steps (e.g., S = 100 to S = 50) and fails entirely for single-step inference. This superior performance of RDIM arises from the $x_0$ prediction dynamics being intrinsically aligned with implicit sampling, allowing the network to correct residual errors directly in image space and maintain faithful reconstructions even with extremely short sampling chains. Empirically, we observe that very few steps tend to produce slightly smoother images, but more faithful reconstructions (high PSNR, stable structure), and increasing the number of steps increases high-frequency details (improving LPIPS) but also increases the risk of artifacts. To clarify this, we added a perception-distortion analysis (Appendix C.8). The curves show that RDIM achieves a better fidelity-perception trade-off compared to existing methods (see Fig. 11) and that RDIM-1 and -5 are particularly robust against artifact formation. Also, because RDIM operates in image space, it is straightforward to integrate perceptual losses, as demonstrated with the RDIM-PQ-1 model, which attains the best LPIPS score (0.114) overall at a small cost of a small PSNR decrease (still better than the state of the art) - see Table 2 in the paper.

---

> ### Author Response · Authors · 2025-12-03
> **Our responses to Reviewer RD6i**
>
> We sincerely thank the Reviewer for recognizing the efficiency of our approach for inverse problems and the theoretical significance of our contribution.
>
> **Practical applicability of RDIM.** We understand that text-to-image editing tasks are a fundamental problem of current days. In this context, we value the contribution of FLUX.1. Yet, there are other important image-to-image problems which are not yet fully solved and remain an open challenge. This is shown by the volume of work that has been published lately (e.g., CTMSR, ICCV’25; MaRS, ICLR’25; DDBM, ICLR’24; DBIM, ICLR’25; GOUB, ICML’24; UniDB, ICML’25; UniDB++, 2025; DDRM, NeurIPS’22; IR‑SDE, ICML’23).
> In particular, RDIM is positioned in this context (image-to-image), and we show that it achieves substantially higher reconstruction quality than all prior efficient diffusion and bridge-based approaches while requiring only **1–5 inference steps**, in contrast to methods that rely on long sampling trajectories. This provides a direct advantage in real-world deployment scenarios where speed and computational efficiency are critical.
>
> Furthermore, we emphasize that the proposed method achieves top performance while performing patched inference (32x32 low resolution patches to 128x128 high resolution patches). In contrast, many state of the art (SoA) methods use full-size image inference, requiring significantly more resources to attain comparable or even worse performance.
>
> We also note that RDIM has been evaluated on two real-world restoration datasets (FMD and SIDD) included in the paper. Notably, the FMD dataset contains mixed Gaussian–Poisson noise, directly showing a real-application use-case with a mixture of noise sources. Across both datasets, RDIM demonstrates the best performance, highlighting its practical relevance beyond synthetic benchmarks.
>
> **Latent-space vs. image-space diffusion.** We appreciate the Reviewer’s connection to recent latent-space editing models such as FLUX.1/FLUX.2. We agree that latent-space inference is attractive for semantic editing. However, we would like to note that:
>
> - **RDIM is not tied to image space.** The framework can operate in the latent space without any conceptual changes, exactly as DDPMs were adapted for latent diffusion. Our choice of image space follows standard restoration practice and is motivated by stability and fidelity rather than by theoretical limitations.
> - **Image-space processing can be beneficial.** Operating in image space makes integrating perceptual losses straightforward.
> - **Restoration tasks differ fundamentally from semantic editing.**  In inverse problems, strict reconstruction fidelity and non-hallucination are crucial. Latent-space models such as FLUX are optimized for creative editing, not accurate inversion of physical degradations. For restoration, operating in image space remains advantageous.
> - **Efficiency comparison favors RDIM.** Even latent-space ODE solvers typically require 20–50 steps. RDIM achieves competitive or superior fidelity in 1–5 steps, making it highly efficient even relative to latent methods.
>
> **Robustness under model mismatch.** RDIM’s robustness under model mismatch arises from the fact that its residual-prediction formulation allows avoiding long diffusion trajectories (reducing error accumulation) and does not require the terminal forward state to coincide exactly with the observed low-quality input (a key limitation of many bridge-based formulations), making it less sensitive to discrepancies in the degradation operator, which helps preserve reconstruction quality under mismatch. These characteristics help stabilize performance when the true degradation differs from the nominal one. *An additional observation is that RDIM-1 maintains a PSNR of 26.26 dB even under mismatched degradation, which is higher than the matched-degradation performance reported by several methods such as DDRM, IR-SDE, DDBM, and UniDB-SDE, all of which remain below 26 dB (see Table 2).* This reinforces that the method preserves reconstruction fidelity even when the assumed and actual degradations are not aligned.

---

### Official Review · Reviewer_BJ2i · 2025-11-01

**Soundness:** 3
**Presentation:** 3
**Contribution:** 3
**Rating:** 4
**Confidence:** 4

**Summary:**

This paper proposes Residual Diffusion Implicit Models (RDIM), shifting the diffusion process from the traditional paradigm of ‘diffusing a clean image into pure noise and then reconstructing from the noise’ to directly modelling the residual $\Delta = x_0 − y_0$ between the clean image $x_0$ and the observed low-quality image $y_0$. Specifically, the forward (degradation) process progressively removes residuals while allowing controllable variance injection; the inverse (reconstruction) process recovers $x_0$ from the final latent variable $x_T$ aligned with the observation (rather than pure noise). Inverse sampling employs implicit sampling akin to DDIM, enabling skipping of intermediate steps to achieve restoration in fewer steps or even a single step. Experiments demonstrate significant reductions in sampling steps compared to standard methods like DDPM/ResShift for denoising and super-resolution tasks, while achieving superior PSNR/SSIM metrics.

**Strengths:**

1. The idea is very interesting, i.e. changing the diffusion objective from ‘noise’ to “residual” theoretically aligns better with inverse problems (where observations already contain substantial information), reducing unnecessary randomness and redundant ‘recovery from noise’ steps.

2. Employing implicit sampling (DDIM-style) with skip sampling enables the inverse process to be performed in $S<<T$ steps. This should reduce inference latency in practical applications, especially for natural image restoration.

3. Experimental results on natural image restorations (e.g. denoising, super-resolution) are good.

**Weaknesses:**

Overall, the paper builds upon and extends ResShift, representing a further advancement of the residual/residual shift concept when combined with DDIM-style implicit sampling. Similar approaches, such as replacing noise space with observation-aligned latent space or directly modelling residuals, have partial precedents in the literature and essentially overlap with certain conditional diffusion/restoration models, such as DDRM (Denoising Diffusion Restoration Models) and 'Residual denoising diffusion models'.

1. Line 127. 'Subsequently, the Markovian formulation is relaxed to derive a non-Markovian process that preserves the same
marginal distributions. The author does not provide a complete mathematical proof or rigorous conditions. In particular, when stepwise sampling is employed, the claim that the target marginal distribution is preserved (or that the inverse process remains valid) needs further theoretical justification.

2. The results are not particularly promising. The experiments lack a systematic comparison with recent works on fast or accelerated diffusion models, e.g. Consistency Models (Song et al., ICML'23).

3. Furthermore, the paper does not sufficiently demonstrate performance degradation when the forward model is mismatched.

4. The performance of RDIM needs to be demonstrated on real or more challenging applications, such as medical image reconstruction (e.g. accelerated MRI or sparse-view CT).

**Questions:**

Is the forward degradation model used for training and testing consistent with the assumptions made during inference? If the degradation operator known during training differs from that encountered during testing, how robust is the method?

In the context of 'single-step reconstruction', the constraints and generalisability (i.e. feasibility across tasks/noise intensities) must be explicitly stated.

Regarding real inverse problems (e.g. sparse-view CT), where the observations (sinograms) are generally corrupted by a mixture of Gaussian and Poisson processes, will RDIM still work?

---

> ### Author Response · Authors · 2025-11-21
>
> We sincerely thank the reviewer for the constructive and insightful feedback. We have revised the manuscript to clarify the methodology, expand comparisons, and strengthen empirical validation. We address your comments below.
>
> **Relation to prior work.** We agree that RDIM is conceptually connected to residual modeling and observation-aligned diffusion. Our contribution unifies these ideas under a principled residual marginal-preserving formulation that (i) explicitly aligns the forward diffusion with the true degradation process, (ii) generalizes ResShift as a special case, and (iii) enables DDIM-style implicit sampling that preserves the forward marginals, allowing extreme acceleration. We carefully expanded the “Related Work” section (Sec. 4) to contextualize the proposed RDIM model and highlighted the distinctions relative to DDRM, residual-DDPMs, and bridge models.
>
> **Broader comparisons and experimental strengthening.** We expanded our evaluation to include a wide range of recent accelerated and consistency-based methods. Table 2 and Fig. 3 now compare RDIM against recent consistency-based and few-step methods (CTMSR, ICCV’25; MaRS, ICLR’25), as well as diffusion bridge models (DDBM, ICLR’24; DBIM, ICLR’25; GOUB, ICML’24; UniDB, ICML’25; UniDB++, 2025), and other baselines such as DDRM (NeurIPS’22) and IR‑SDE (ICML’23). RDIM consistently achieves the best PSNR and SSIM, even with extremely few sampling steps. In particular, RDIM-1, which uses a single sampling step, surpasses these methods in PSNR and SSIM while using the same or fewer sampling steps, highlighting its extreme efficiency. Its performance is only surpassed by CTMSR and only on LPIPS, as it directly optimises for this metric. Nonetheless, the RDIM-PQ (variant optimizing perceptual quality) surpasses CTMSR on LPIPS, attaining only a slight degradation on PSNR and SSIM vs the standard proposed RDIM, but still far better than all the state of the art. Qualitative examples also show that RDIM produces sharper and more faithful reconstructions (Figs. 3,11,14,15), especially on fine textures and detailed structures, where competing methods exhibit noticeable blurring, deformations and hallucinations.
>
> **Non-Markovian relaxation and marginal preservation.** The non-Markovian forward process conditions on $x_0$, breaking the memoryless property, while the reverse transition is designed to preserve the target marginal $q(x_t|x_0,\Delta)$. Because Gaussian conditionals are linear in their conditioning variables, we can solve an affine system whose solution yields a reverse transition that exactly matches the desired marginal. We reformulated both the main text and appendices to take into account the comments by all the reviewers. In particular, we now explicitly reference Lemma B.1 in the derivation, making the conditions for marginal preservation explicit and mathematically grounded.
>
> **Robustness, generalizability, and applicability to real inverse problems.** We agree that evaluating on MRI/CT is an important direction. While this falls outside the current scope, we note that RDIM is already evaluated on two real-world restoration datasets (FMD and SIDD) included in the paper. Notably, the FMD dataset contains mixed Gaussian–Poisson noise, directly addressing the question posed by the reviewer. Moreover, RDIM’s residual-driven formulation applies directly to broader inverse problems (inpainting, colorization, etc.). Nonetheless, we acknowledge that extending to MRI/CT is important future work. Further, to explicitly evaluate robustness under model mismatch, we added a robustness study (Appendix C.10 and table AC.2 in official comment to AC) evaluating RDIM trained on bicubic ×4 downsampling and tested on images that are additionally corrupted with Poisson noise. As expected, this train/test mismatch reduces performance. However, RDIM exhibits less performance deterioration than competing methods. For example, RDIM-1 drops from 29.180 to 26.26 dB (-2.92 dB), which is significantly smaller than ResShift (-3.845 dB), GOUB-SDE (-6.725 dB), and UniDB-SDE (-6.176 dB). This indicates strong practical robustness compared to existing residual- or bridge-based diffusion models.

---

> > ### Author Response · Authors · 2025-11-25
> >
> > At the request of one of the reviewers, we have updated the paper by highlighting changes using pdfdiff, and also took the opportunity to include the missing results regarding CTMSR.

---

### Author Response · Authors · 2025-12-03
**Official Comment to the Area Chair (AC) by Authors**

Thank you for taking the time to handle our submission. Below we address the main/common issues raised by the reviewers, which we believe contributed to the initial scores. We summarize our overall responses here, with detailed point-by-point replies provided in each individual reviewer response.

In line with open-science practices, we provide public access to our source code for reproducibility and testing at: https://anonymous.4open.science/r/RDIM.

### **Clarification of contributions**

We recall the contributions (revised in the paper):
- a residual-aligned diffusion forward process, preserving sample-level correspondences across HQ/LQ domains;
- a non-Markovian reverse process derived to preserve the forward marginals, enabling theoretically grounded implicit sampling and stable few-step reconstruction;
- controllable stochasticity ($\gamma$,$\eta$) on both forward and reverse processes, allowing interpolation between deterministic and stochastic reconstructions;
- demonstrated SoA results in image reconstruction (denoising, super-resolution), on real data, even with one-step inference;
- evidence that reducing sampling steps accelerates inference and yields more faithful reconstructions by limiting hallucinations that arise in long diffusion chains.

### **Related work/Comparison with state of the art (SoA)**

Reviewers were particularly concerned with a detailed SoA comparison. We address this by:
1. fully revising the SoA section of the paper
2. including an in-depth comparison with the SoA, including 9 new methods taken from reviewer’s suggestions (new on the paper: tbl 2, figs. 3,11,14,15). Key new results below:

» **Table AC.1: x4-SR on DIV2K-Bicubic**
|  **Method**  | **#steps** | **PSNR ↑** | **SSIM ↑** | **LPIPS ↓** |
|---------|:-------:|:----------:|:----------:|:------------:|
| DDRM (NeurIPS’22)    | 100     | 24.350     | 0.592      | 0.364        |
| ResShift (NeurIPS’23) | 100     | 27.455     | 0.780      | 0.153        |
| IR-SDE (ICML’23)     | 100     | 25.900     | 0.657      | 0.231        |
| DDBM (ICLR’24)       | 100     | 24.210     | 0.581      | 0.384        |
| GOUB-SDE (ICML’24)   | 100     | 26.890     | 0.748      | 0.220        |
| GOUB-ODE (ICML’24)   | 100     | 28.500     | **0.807**  | 0.328        |
| UniDB-SDE (ICML’25)  | 100     | 25.460     | 0.686      | 0.179        |
| UniDB-ODE (ICML’25)  | 100     | **28.640** | **0.807**  | 0.323        |
| UniDB++-50 (2025)    | 50      | 26.610     | 0.754      | 0.159        |
| UniDB++-20 (2025)    | 20      | 27.380     | 0.777      | 0.179        |
| UniDB++-5 (2025)     | 5       | 28.400     | 0.805      | 0.235        |
| MaRS-5 (ICLR’25)     | 5       | 27.730     | 0.783      | 0.286        |
| DBIM-5 (ICLR’24)     | 5       | 28.050     | 0.795      | 0.260        |
| CTMSR (ICCV’25)      | **1**   | 27.087     | 0.759      | **0.130**    |
| » **Proposed** | | | | |
| RDIM-1               | **1**   | **29.180** | **0.824**  | 0.257        |
| RDIM-PQ-1        | **1**   | 29.004     | 0.817      | **0.114**    |

**In conclusion, RDIM delivers:**
- better reconstruction with a fraction of the sampling cost (#steps).
- better perception–distortion trade-off than SoA (new Appendix C.8).
- top performance with patched inference (32x32 degraded patches to 128x128 reconstructed ones), while most SoA uses full-image inference (computationally heavier).

### **Robustness under training/inference mismatch**

**New SR results in Appendix C.10** with:
- training on standard bicubic degradation;
- inference with bicubic degraded images plus Poisson noise (model mismatch).
Results below (Δ = difference between model mismatch, this experiment, and model match, the above table):

» **Table AC.2: x4-SR on DIV2K-Bicubic with model mismatch**
| **Method** | **PSNR ↑** | **ΔPSNR** | **SSIM ↑** | **ΔSSIM** | **LPIPS ↓** | **ΔLPIPS** |
|--------|:------:|:-----:|:-------:|:------:|:-------:|:-------:|
| GOUB-SDE   | 20.165 | -6.725 | 0.335 | -0.413 | 0.664 | +0.444 |
| UniDB-SDE  | 19.284 | -6.176 | 0.314 | -0.372 | 0.697 | +0.518 |
| CTMSR-1    | 22.987 | -4.100 | 0.466 | -0.293 | 0.495 | +0.365 |
| **RDIM Series** | | | | | | |
| RDIM-1     | **26.260** | **-2.920** | **0.673** | **-0.151** | 0.519 | +0.262 |
| RDIM-10    | 24.363 | -3.600 | 0.564 | -0.231 | **0.415** | **+0.237** |

**Conclusion:**
- Under model mismatch, RDIM achieves top performance and smallest Δ difference.

### **Other comments**

- RDIM is evaluated on real datasets (FMD, SIDD), showing top performance. FMD dataset contains mixed Gaussian–Poisson noise, showing that RDIM can handle mixtures of noise sources (tbl 1).
- RDIM allows direct use of perceptual losses. RDIM-PQ-1 (trained with a perceptual quality (PQ) objective) attains best LPIPS in x4-SR, while achieving competitive PSNR and SSIM. The perception–distortion (Appendix C.8) shows superior fidelity–perception trade-off, consistent with the *qualitative results in Figs 14,15*.

---

### Meta-Review · Area_Chair_FTuA · 2026-01-05

**Summary:**

**Summary**: This paper introduces Residual Diffusion Implicit Models (RDIM), a diffusion framework designed for image inverse problems such as denoising and super-resolution. The core proposal involves modeling the residual between high-quality (HQ) and low-quality (LQ) images, aligning the forward process with the degradation mechanism, and employing a non-Markovian implicit sampling scheme for accelerated reconstruction. Experiments on denoising and super-resolution tasks are presented, aiming to demonstrate RDIM's efficiency and performance.

**Strengths**: Overall, reviewers acknowledged that the paper is generally well-written and easy to follow. Some reviewers noted RDIM's reported performance benefits on the evaluated tasks and the extensiveness of the ablation studies.

**Weaknesses**: Reviewers raised significant and numerous concerns, primarily regarding the technical novelty, experimental scope, theoretical foundation, and presentation. These critical issues remained largely unaddressed or unconvincingly resolved during the rebuttal period, leading to a consensus for rejection:
*   **Limited Technical Novelty and Unaddressed Related Work:** Multiple reviewers (R_BJ2i, R_2RLd, R_hRiF) critically questioned the novelty of RDIM. They highlighted that the core concepts, such as residual modeling and observation-aligned latent spaces, bear strong similarities and have significant precedents in prior works like ResShift (which RDIM generalizes), DDRM, and RDDM ("Residual denoising diffusion models"). Additionally, reviewers consistently noted a lack of an adequate survey of these recent and highly relevant works, particularly in distinguishing RDIM's contributions.

*   **Weak Theoretical Justification:** Reviewer R_BJ2i specifically highlighted a lack of "complete mathematical proof or rigorous conditions" for the non-Markovian reverse process to preserve marginal distributions.

*   **Insufficient Experimental Comparisons and Scope:** A pervasive concern across all reviewers (R_BJ2i, R_RD6i, R_2RLd, R_hRiF) was the initial lack of comprehensive comparison with state-of-the-art methods. The paper failed to adequately benchmark against a wide array of recent accelerated diffusion models, consistency models, and various diffusion bridge methods. Reviewers requested more comprehensive analysis on recent SOTA methods, performance on challenging real-world applications, and the inclusion of perceptual metrics (LPIPS, FID). While the authors added extra results for tasks like inpainting and deblurring, these were considered trivial as they only compared against ResShift and not more recent state-of-the-art methods, thus not sufficiently addressing the demand for competitive baselines.

*   **Incomplete Robustness Analysis:** Reviewer R_BJ2i noted that the paper did not sufficiently demonstrate performance degradation when the forward model is mismatched, which is a critical aspect for assessing real-world applicability and reliability.

*   **Clarity and Presentation Issues:** Reviewer R_hRiF identified a "logical inconsistency" in the paper's structure. Reviewer R_2RLd also suggested rephrasing the introduction's characterization of DDPMs, which were characterized as powerful models for reconstruction in the original manuscript.

**Decision**: The paper received consistent negative ratings (4, 4, 4, 2). The fundamental weakness of the submission lies in its lack of technical novelty. Multiple reviewers noted that the central ideas of RDIM are highly similar to previous works such as DDRM, RDDM, and other bridge diffusion models. Specifically, while the authors claim that their method starts from pure noise and progressively denoises latent variables, and that the RDIM forward process is “explicitly designed to align with the degradation mechanism by progressively removing the residuals between the clean and corrupted signals while optionally injecting a controllable amount of noise,” this approach closely parallels the ideas in RDDM. For reference, RDDM states: “Our residual diffusion represents directional diffusion from the target image to the degraded input image and explicitly guides the reverse generation process for image restoration, while noise diffusion represents random perturbations in the diffusion process.” During the rebuttal, the authors attempted to clarify the differences between RDIM and ResShift, but did not address the much more significant similarities to RDDM and DDRM. This omission greatly undermines the claimed contributions of the paper. Additionally, although the authors provided extra results on tasks such as inpainting and deblurring, these experiments were trivial, as they only compared RDIM to ResShift, rather than to more recent state-of-the-art methods.  Given these issues, especially the unresolved concerns about novelty and lack of comprehensive comparisons with the latest methods, I recommend rejection in its current form.

**Reviewer Concerns:**

The authors made efforts to address reviewer feedback by clarifying differences with ResShift, expanding experimental results, and making some improvements to the manuscript. However, the most significant concerns—especially regarding the technical novelty of the work and the comprehensiveness of experimental validation—remain largely unresolved.

**Concerns Addressed or Partially Addressed**

- **Clarity and Presentation:**
  Reviewers acknowledged that the paper is generally well-written and easy to follow. The ablation studies were also noted as extensive and informative.

- **Distinction from ResShift:**
  The authors attempted to clarify the differences between RDIM and ResShift in their rebuttal.

- **Additional Image Restoration Tasks:**
  The authors added results on inpainting and deblurring. However, these new experiments only compared against ResShift and not more recent state-of-the-art (SOTA) methods, thus failing to fully address reviewer expectations for competitive baselines.

- **Expanded Experimental Comparisons:**
  The rebuttal included broader comparisons for image super-resolution, adding the LPIPS perceptual metric and additional SOTA methods. Reviewers suggested similar comprehensive comparisons for denoising and other restoration tasks, as well as including additional perceptual metrics such as FID.

- **Related Work and Wording:**
  The authors added more discussion of recent related works and rephrased potentially misleading claims about DDPMs in the introduction.

**Outstanding Concerns**

- **Limited Technical Novelty and Overlap with Prior Work:**
  The core concepts of RDIM (e.g., residual modeling and observation-aligned latent spaces) are highly similar to prior works such as DDRM and RDDM. The rebuttal did not sufficiently clarify distinctions between RDIM and these closely related approaches, substantially diminishing the perceived contribution of the paper.

- **Insufficient Experimental Scope:**
  Despite some improvements, the experimental evaluation remains lacking. Comprehensive comparisons with recent SOTA methods on tasks beyond super-resolution (such as denoising and other image restoration tasks) are still absent. Suggested metrics like FID were not incorporated.

**Reviewer Scores:**

Reviewer BJ2i (Rating: 4 -> maybe keep the same), Reviewer RD6i (Rating: 4 -> maybe keep the same), Reviewer 2RLd (Rating: 4 -> maybe keep the same), Reviewer hRiF (Rating: 2 -> maybe slightly improved)

---

### Decision · Program_Chairs · 2026-01-26

Reject